# Imitation Learning as Return Distribution Matching

**Filippo Lazzati**
Politecnico di Milano
Milan, Italy
`filippo.lazzati@polimi.it`

**Alberto Maria Metelli***
Politecnico di Milano
Milan, Italy
`albertomaria.metelli@polimi.it`

## Abstract

We study the problem of training a risk-sensitive reinforcement learning (RL) agent through imitation learning (IL). Unlike standard IL, our goal is not only to train an agent that matches the expert's expected return (i.e., its *average performance*) but also its *risk attitude* (i.e., other features of the return distribution, such as variance). We propose a general formulation of the risk-sensitive IL problem in which the objective is to match the expert's return distribution in Wasserstein distance. We focus on the tabular setting and assume the expert's reward is *known*. After demonstrating the limited expressivity of Markovian policies for this task, we introduce an efficient and sufficiently expressive subclass of non-Markovian policies tailored to it. Building on this subclass, we develop two provably efficient algorithms—**RS-BC** and **RS-KT**—for solving the problem when the transition model is unknown and known, respectively. We show that **RS-KT** achieves substantially lower sample complexity than **RS-BC** by exploiting dynamics information. We further demonstrate the sample efficiency of return distribution matching in the setting where the expert's reward is *unknown* by designing an oracle-based variant of **RS-KT**. Finally, we complement our theoretical analysis of **RS-KT** and **RS-BC** with numerical simulations, highlighting both their sample efficiency and the advantages of non-Markovian policies over standard sample-efficient IL algorithms.

## 1 Introduction

Imitation Learning (IL) (Abbeel & Ng, 2004; Osa et al., 2018) is the problem of training an agent to behave by mimicking demonstrations from an expert. By removing the need for designing a reward function for the task—which is often a difficult challenge (Hadfield-Menell et al., 2017)—IL has been successfully applied in diverse domains, including robotics (Argall et al., 2009), autonomous driving (Le Mero et al., 2022), finance (Goluža et al., 2023), and LLMs (Zhao et al., 2025).

Most existing IL algorithms—including `BC` (Behavioral Cloning) (Pomerleau, 1988), `GAIL` (Ho & Ermon, 2016), and others (Ziebart, 2010; Reddy et al., 2020; Garg et al., 2021)—focus on finding the *Markovian* policy that best matches the expert's *occupancy measure*. This focus is motivated by two observations. First, matching occupancy measures guarantees that the *expected return* of our policy is close to the expert's, regardless of the expert's *unknown* reward (Abbeel & Ng, 2004). Second, Markovian policies are sufficiently expressive. Indeed, for any arbitrary policy, there exists a Markovian policy with the same occupancy measure (Puterman, 1994).

By focusing solely on the occupancy measure—which captures the expected value of the return distribution—standard IL algorithms are inherently *risk-neutral*, ignoring other characteristics of the return distribution such as the variance (Mannor & Tsitsiklis, 2011). However, expert demonstrations often come from humans who, in domains like finance (Föllmer & Schied, 2016) or autonomous driving (Bernhard et al., 2019), exhibit *risk-sensitive* behavior under stochasticity. In these settings, the key aspect of the demonstrations is the expert's *risk attitude*, encoded in the shape of the return distribution (Bellemare et al., 2023), but overlooked by standard IL methods.

---

*Corresponding author.

To address this, Santara et al. (2018); Lacotte et al. (2019) proposed extending occupancy measure matching to risk-sensitive settings by additionally matching the Conditional Value at Risk (CVaR) (Rockafellar & Uryasev, 2000) at a chosen level $\alpha$ of the expert's return distribution. Intuitively, by seeking the *Markovian* policy that best matches both the expectation and the CVaR at level $\alpha$, these algorithms attempt to imitate not only the expert's average performance but also its tail behavior.

However, this approach faces two main limitations: $(i)$ matching only the expectation and the CVaR at level $\alpha$ captures a narrow slice of the expert's full return distribution and thus provides a weak imitation of risk attitude, and $(ii)$ Markovian policies are not expressive enough to capture all relevant risk-sensitive behaviors (Bellemare et al., 2023), leading to misspecification error. To overcome these challenges, we reformulate risk-sensitive IL as matching the expert's *entire* return distribution, and we design algorithms that perform policy search efficiently in the space of non-Markovian policies.

**Contributions.**    Our main contributions are as follows:

- We formulate IL as the problem of matching the expert's return distribution in Wasserstein distance. We motivate this setting and demonstrate the importance of non-Markovian policies (Section 3).

- We introduce an efficient and expressive subclass of non-Markovian policies for the tabular setting with a *known* expert reward, and use it to develop two provably efficient algorithms, **RS-BC** and **RS-KT**, for the cases where the transition model is unknown and known, respectively (Section 4).

- We show that in the tabular setting with an *unknown* expert reward but a known transition model, sample efficiency can still be achieved by devising an oracle-based variant of **RS-KT** (Section 5).

- Finally, we conduct numerical simulations to empirically evaluate **RS-BC** and **RS-KT**, comparing them in particular against standard provably efficient IL algorithms (Section 6).

All proofs are provided in Appendix B–D, and additional related work is discussed in Appendix A.

## 2    PRELIMINARIES

**Notation.**    Given a natural number $n \in \mathbb{N}$, we define $[\![n]\!] := \{1, 2, \ldots, n\}$. Given a real number $x \in \mathbb{R}$, we let $\lfloor x \rfloor := \max_{m \in \mathbb{Z} : m \leqslant x} m$ be the floor function. Given two sets $\mathcal{X}, \mathcal{Y}$, we denote by $\Delta^{\mathcal{X}}$ and $\Delta^{\mathcal{X}}_{\mathcal{Y}}$, respectively, the set of probability measures on $\mathcal{X}$ and the set of functions from $\mathcal{Y}$ to $\Delta^{\mathcal{X}}$. Given probabilities $p, q \in \Delta^{\mathcal{X}}$ on a finite support $\mathcal{X} \subset \mathbb{R}$, with cumulative distributions $F_p$ and $F_q$, the (1-)Wasserstein distance is $\mathcal{W}(p, q) := \int_{\mathbb{R}} |F_p(x) - F_q(x)| dx$ (Villani, 2008), and the total variation distance is $\mathrm{TV}(p, q) := \frac{1}{2} \sum_{x \in \mathcal{X}} |p(x) - q(x)|$. The CVaR at level $\alpha \in (0, 1)$ of $p$ is $\mathrm{CVaR}_\alpha(p) := \frac{1}{\alpha} \int_0^\alpha F_p^{-1}(u) du$, where $F_p^{-1}(u) := \inf_{z \in \mathbb{R} : F_p(z) \geqslant u} z$ (Rockafellar & Uryasev, 2000).

**Markov Decision Processes (MDPs).**    A tabular finite-horizon episodic Markov Decision Process without reward (MDP\R) (Puterman, 1994; Abbeel & Ng, 2004) is a tuple $\mathcal{M} := (\mathcal{S}, \mathcal{A}, H, s_0, p)$, where $\mathcal{S}$ is the finite state space ($S := |\mathcal{S}|$), $\mathcal{A}$ is the finite action space ($A := |\mathcal{A}|$), $H \in \mathbb{N}$ is the horizon, $s_0 \in \mathcal{S}$ is the initial state, and $p \in \Delta^{\mathcal{S}}_{\mathcal{S} \times \mathcal{A} \times [\![H]\!]}$ is the transition model. An MDP\R $\mathcal{M}$ can be enriched with a reward $r : \mathcal{S} \times \mathcal{A} \times [\![H]\!] \to [0, 1]$, to obtain an MDP $\mathcal{M}_r := (\mathcal{S}, \mathcal{A}, H, s_0, p, r)$. We denote the set of state-action trajectories of length $h - 1$ as $\Omega_h := (\mathcal{S} \times \mathcal{A})^{h-1}$ for all $h \in [\![H + 1]\!]$, and define $\Omega := \bigcup_{h \in [\![H]\!]} \Omega_h$. For any trajectory $\omega = (s_1, a_1, \ldots, s_h, a_h)$ and reward $r$, we let $G(\omega; r) = \sum_{h' \in [\![h]\!]} r_{h'}(s_{h'}, a_{h'})$ denote the sum of rewards of $\omega$. A policy $\pi$ prescribes actions in states. We denote by $\Pi^{\mathrm{NM}} := \Delta^{\mathcal{A}}_{\Omega \times \mathcal{S}}$ the set of non-Markovian (i.e., history-dependent) policies,[1] and by $\Pi^{\mathrm{M}} := \Delta^{\mathcal{A}}_{[\![H]\!] \times \mathcal{S}}$ the set of Markovian policies. Note that $\Pi^{\mathrm{M}} \subseteq \Pi^{\mathrm{NM}}$. Playing a policy $\pi \in \Pi^{\mathrm{NM}}$ in an MDP\R $\mathcal{M}$ (or an MDP) induces a probability distribution over trajectories $\mathbb{P}^\pi \in \Delta^{\Omega_{H+1}}$. The occupancy measure $d^\pi$ of $\pi$ in $\mathcal{M}$ is the marginal of $\mathbb{P}^\pi$ over state-action pairs at a given stage: $d_h^\pi(s, a) := \mathbb{P}^\pi(s_h = s, a_h = a)$. Given a reward $r$, the *random* sum of rewards $\sum_{h=1}^H r_h(s_h, a_h)$ induced by the execution of $\pi$ is the return, and we denote its distribution, called the *return distribution* (Bellemare et al., 2023), as $\eta_r^\pi(g) := \mathbb{P}^\pi\left(\sum_{h=1}^H r_h(s_h, a_h) = g\right)$ for all $g \in [0, H]$. Lastly, we denote by $J_r^\pi$ the expectation of $\eta_r^\pi$.

---

[1] Neglecting past rewards in $\Pi^{\mathrm{NM}}$ is without loss of generality since we consider deterministic rewards.

**Imitation Learning (IL).** In IL, we are given a dataset $\mathcal{D}^E = \{\omega_i\}_{i \in [\![N]\!]}$ of $N$ trajectories $\omega_i \in \Omega_{H+1}$, collected by a (potentially non-Markovian) expert policy $\pi^E \in \Pi^{\mathrm{NM}}$, and the goal is to find a policy $\widehat{\pi}$ with expected return close to that of $\pi^E$ under the expert's reward $r^E$ (Abbeel & Ng, 2004):

$$\widehat{\pi} \in \arg\min_{\pi \in \Pi^{\mathrm{NM}}} \left| J_{r^E}^{\pi^E} - J_{r^E}^{\pi} \right|. \tag{1}$$

Since $r^E$ is usually unknown, the problem is reformulated in robust terms as finding a policy that performs comparably to the expert for any possible reward:

$$\widehat{\pi} \in \arg\min_{\pi \in \Pi^{\mathrm{NM}}} \max_{r:\mathcal{S} \times \mathcal{A} \times [\![H]\!] \to [0,1]} \left| J_r^{\pi^E} - J_r^{\pi} \right|. \tag{2}$$

Interestingly, Abbeel & Ng (2004); Ho & Ermon (2016) showed that Eq. (2) essentially reduces to finding a policy $\widehat{\pi}$ whose occupancy measure $d^{\widehat{\pi}}$ is close to the expert's $d^{\pi^E}$. Thus, Eq. (2) (and Eq. 1) can be addressed with Markovian policies $\Pi^{\mathrm{M}}$, which are known to be expressive enough for occupancy measure matching problems (e.g., see Laroche & Tachet Des Combes (2023)). Based on these insights, recent theoretical work (Rajaraman et al., 2020; Foster et al., 2024) demonstrated that IL can be solved *provably* efficiently.

**Risk-sensitive IL.** Optimizing Eqs. (1)–(2) guarantees imitation of the expert's *average performance*, i.e., its expected return $J_{r^E}^{\pi^E}$, but does not guarantee imitation of its *risk attitude*, encoded in the shape of its return distribution $\eta_{r^E}^{\pi^E}$. For this reason, Santara et al. (2018); Lacotte et al. (2019) proposed strengthening the standard IL formulation by also matching the CVaR at some level $\alpha \in (0,1)$ of $\eta_{r^E}^{\pi^E}$ in addition to $J_{r^E}^{\pi^E}$. Formally, in the unknown $r^E$ setting, they propose:[2]

$$\widehat{\pi} \in \arg\min_{\pi \in \Pi^{\mathrm{NM}}} \max_{r:\mathcal{S} \times \mathcal{A} \times [\![H]\!] \to [0,1]} \left( \left( J_r^{\pi^E} - J_r^{\pi} \right) + \rho\left( \mathrm{CVaR}_\alpha(\eta_r^{\pi^E}) - \mathrm{CVaR}_\alpha(\eta_r^{\pi}) \right) \right), \tag{3}$$

where $\rho(x) = x$ if $x \leqslant 0$ and $+\infty$ otherwise. This extension, however, makes the problem substantially harder than standard IL, since the optimal solution to Eq. (3) cannot, in general, be found among the Markovian policies $\Pi^{\mathrm{M}}$ (even if $r^E$ was known).[3] Nevertheless, Santara et al. (2018); Lacotte et al. (2019) ignored this aspect and proposed algorithms that output Markovian policies, which, however, may not be suited for general non-Markovian experts like humans (Mandlekar et al., 2022).

## 3 RETURN DISTRIBUTION MATCHING

Our goal is to train agents that match not only the expert's expected return but also its risk attitude. Standard IL is unsuitable since it ignores risk, while existing risk-sensitive IL methods only capture a limited aspect of the expert's return distribution, i.e., the CVaR at a fixed level. We therefore propose an alternative formulation, called *return distribution matching* (RDM), which requires matching the *entire* expert return distribution $\eta_{r^E}^{\pi^E}$ in Wasserstein distance:

$$\widehat{\pi} \in \arg\min_{\pi \in \Pi^{\mathrm{NM}}} \mathcal{W}\left( \eta_{r^E}^{\pi}, \eta_{r^E}^{\pi^E} \right). \tag{4}$$

This objective extends Eq. (1) and assumes knowledge of the expert reward $r^E$. Our focus will primarily be on this known-reward setting (see Section 4), both because it is of independent interest (similar to inverse constrained RL (Malik et al., 2021) and utility learning (Lazzati & Metelli, 2025)), and because it provides a foundation for the more challenging unknown-reward case, which we next formalize. When $r^E$ is unknown, following Eqs. (2)–(3), we propose a *robust* version of RDM:

$$\widehat{\pi} \in \arg\min_{\pi \in \Pi^{\mathrm{NM}}} \max_{r:\mathcal{S} \times \mathcal{A} \times [\![H]\!] \to [0,1]} \mathcal{W}\left( \eta_r^{\pi}, \eta_r^{\pi^E} \right), \tag{5}$$

which requires matching the expert's return distribution for *all* possible rewards (see Section 5). We now provide three key observations about RDM. First, matching return distributions in Wasserstein

---

[2]The formulation of Santara et al. (2018) is slightly different, as they require matching the expectation while optimizing the CVaR; however, the high-level idea and the issues with non-Markovian policies remain the same.

[3]Indeed, since the optimal policy to a CVaR optimization problem is, in general, non-Markovian (Bäuerle & Ott, 2011), the solution to Eq. (3) also belongs to $\Pi^{\mathrm{NM}}$ due to the hard constraint enforced by $\rho$.

distance is strictly more general than matching expected return or CVaR. Indeed, if $\mathcal{W}(\eta_{r^E}^{\widehat{\pi}}, \eta_{r^E}^{\pi^E}) \leqslant \epsilon$, then $J_{r^E}^{\pi^E} - J_{r^E}^{\widehat{\pi}} \leqslant \epsilon$ and $\left| \text{CVaR}_\alpha(\eta_{r^E}^{\pi^E}) - \text{CVaR}_\alpha(\eta_{r^E}^{\widehat{\pi}}) \right| \leqslant \epsilon/\alpha$ for *any* $\alpha \in (0,1)$ (see Appendix B.1). Second, Wasserstein distance is essential for favorable sample complexity. Stronger metrics, such as total variation, require exponentially many expert trajectories in some instances, even if the MDP $\mathcal{M}_{r^E}$ is fully known (see Appendix B.2 for the proof):

**Theorem 3.1.** *There exist an MDP $\mathcal{M}_{r^E}$ with $S, A, H \geqslant 2$ and an expert policy $\pi^E \in \Pi^{NM}$ such that, even with $N = (S-1)^{H-1} - 1$ trajectories, any algorithm $\mathfrak{A}$ satisfies*

$$\mathbb{E}_{\mathcal{D}^E \sim \mathbb{P}^{\pi^E}} TV\left( \eta_{r^E}^{\pi^E}, \eta_{r^E}^{\widehat{\pi}} \right) \geqslant \frac{1}{2e},$$

*where $\widehat{\pi}$ is the output of $\mathfrak{A}$ given in input $\mathcal{M}_{r^E}$ and $\mathcal{D}^E$.*

Finally, we remark that Markovian policies are not expressive enough for RDM, since they fail to capture the behavior of non-Markovian experts even for the simpler risk-sensitive IL problem in Eq. (3). Note that the gap can be significant even with very short horizons (proof in Appendix B.2):

**Proposition 3.2.** *There exist an MDP $\mathcal{M}_{r^E}$ with horizon $H = 3$ and an expert policy $\pi^E \in \Pi^{NM}$ such that* any *Markovian policy $\pi \in \Pi^M$ satisfies*

$$\mathcal{W}\left( \eta_{r^E}^{\pi^E}, \eta_{r^E}^{\pi} \right) \geqslant 0.5.$$

Therefore, new algorithms that output non-Markovian policies are needed to tackle RDM effectively.

## 4 KNOWN-REWARD SETTING

In this section, we assume the expert's reward $r^E$ is known and present our main contributions. In Section 4.1, we introduce an efficient and sufficiently expressive subset of non-Markovian policies for RDM. Building on this, in Sections 4.2 and 4.3, we develop two provably efficient algorithms, **RS-BC** and **RS-KT**, for the cases where the transition model is unknown and known, respectively.

### 4.1 AN EFFICIENT CLASS OF NON-MARKOVIAN POLICIES

Proposition 3.2 shows that Markovian policies $\Pi^M$ are not expressive enough for RDM. At the same time, optimizing Eq. (4) over the entire set of non-Markovian policies $\Pi^{NM}$ is intractable due to the curse of dimensionality. In this section, we introduce a subclass of non-Markovian policies $\Pi(r_\theta^E)$, lying between $\Pi^{NM}$ and $\Pi^M$, that allows us to address RDM accurately without sacrificing efficiency. The trade-off between accuracy and efficiency is controlled by a parameter $\theta > 0$. To this end, we first establish some notation. For any reward $r$, define $\Pi(r) \subseteq \Pi^{NM}$ as the set of policies whose choice of action depends only on the current state $s$, stage $h$, and the cumulative reward so far $G(\omega; r)$:

$$\Pi(r) := \left\{ \pi \in \Pi^{NM} \, \middle| \, \exists \phi \in \Delta_{[\![H]\!] \times \mathcal{S} \times \mathcal{G}_r}^{\mathcal{A}} : \pi(a|s, \omega) = \phi_h(a|s, G(\omega; r)) \; \forall s \in \mathcal{S}, a \in \mathcal{A}, h \in [\![H]\!], \omega \in \Omega_h \right\},$$

where $\mathcal{G}_r := \{ g \in [0, H-1] \, | \, \exists \omega \in \Omega : G(\omega; r) = g \}$ denotes the set of possible cumulative reward values attainable at any stage except the last.[4] Observe that each $\pi \in \Pi(r)$ can be interpreted as a Markovian policy in the MDP obtained by augmenting the state space $\mathcal{S}$ with the cumulative reward $\mathcal{G}_r$. Next, for any reward $r$ and expert policy $\pi^E \in \Pi^{NM}$, define $\pi_r \in \Pi(r)$ as the policy whose probability of taking an action $a$ in state $s$ with history $\omega \in \Omega_h$ coincides with the "average" probability with which $\pi^E$ selects $a$ in $s$ after accumulating $G(\omega; r)$ reward:

$$\pi_r(a|s, \omega) := \frac{\mathbb{P}^{\pi^E}(s_h = s, \ a_h = a, \ \sum_{h'=1}^{h-1} r_{h'}(s_{h'}, a_{h'}) = G(\omega; r))}{\mathbb{P}^{\pi^E}(s_h = s, \ \sum_{h'=1}^{h-1} r_{h'}(s_{h'}, a_{h'}) = G(\omega; r))}. \tag{6}$$

If the denominator is zero, we set $\pi_r(a|s, \omega) = 1/A$. With these two ingredients, $\Pi(r)$ and $\pi_r$, we can state the following important intermediate result (see Appendix C.1.1 for the proof):

**Lemma 4.1.** *Let $\mathcal{M}_{r^E}$ be any MDP and let $\pi^E \in \Pi^{NM}$ be any policy. Then, the policy $\pi_{r^E} \in \Pi(r^E)$ satisfies $\eta_{r^E}^{\pi_{r^E}}(g) = \eta_{r^E}^{\pi^E}(g)$ for all $g \in [0, H]$.*

---

[4]Note that $\mathcal{G}_r$ is always finite in tabular MDPs with deterministic rewards.

---

**Algorithm 1: RS-BC (Risk-Sensitive Behavior Cloning)**

---

**Input :** Dataset $\mathcal{D}^E = \{(s_1^i, a_1^i, \ldots, s_H^i, a_H^i)\}_{i \in \llbracket N \rrbracket}$, reward $r^E$, parameter $\theta$

`// Count the state-action-cumulative reward occurrences`

1 $M_h(s, g, a) \leftarrow \sum_{i \in \llbracket N \rrbracket} \mathbb{1}\{s_h^i = s, a_h^i = a, \sum_{h'=1}^{h-1} r_{\theta, h'}^E(s_{h'}^i, a_{h'}^i) = g\} \quad \forall h \in \llbracket H \rrbracket, s \in \mathcal{S}, g \in \mathcal{Y}_h^\theta, a \in \mathcal{A}$

`// Retrieve the policy`

2 $\hat{\pi}(a|s, \omega) \leftarrow \begin{cases} \frac{M_h(s, G(\omega; r_\theta^E), a)}{\sum_{a'} M_h(s, G(\omega; r_\theta^E), a')} & \text{if } \sum_{a'} M_h(s, G(\omega; r_\theta^E), a') > 0 \\ \frac{1}{A} & \text{otherwise} \end{cases} \quad \forall h \in \llbracket H \rrbracket, s \in \mathcal{S}, \omega \in \Omega_h, a \in \mathcal{A}$

3 **Return** $\hat{\pi}$

---

In words, Lemma 4.1 guarantees that $\Pi(r^E)$ always contains at least one policy with *exactly* the same return distribution as the expert, namely one that minimizes Eq. (4). Moreover, it provides an analytical expression for such a policy, $\pi_{r^E}$ (see Eq. 6). Unfortunately, $\Pi(r^E)$ is not always desirable. As discussed in Appendix C.1.2, there exist reward functions $r^E$ for which $\Pi(r^E) = \Pi^{\text{NM}}$, and, thus, $\pi_{r^E}$ may be an arbitrary non-Markovian policy, inefficient to store. Intuitively, this occurs when each trajectory yields a different return value, leading to the *exponential* dependence $|\mathcal{G}_{r^E}| = (SA)^{H-1}$. To overcome this limitation, inspired by prior work (Bastani et al., 2022; Lazzati & Metelli, 2025), we adopt a *discretization* approach. Given a parameter $\theta \in (0, 1]$, we define a $\theta$-covering of the interval $[0, h-1]$ as $\mathcal{Y}_h^\theta := \{0, \theta, 2\theta, \ldots, \lfloor h - 1/\theta \rfloor \theta\}$ for all $h \in \llbracket H+1 \rrbracket$, and set $\mathcal{Y}^\theta := \mathcal{Y}_{H+1}^\theta$. Then, for any reward $r$, we construct the discretized reward $r_\theta$ as (breaking ties arbitrarily):

$$r_{\theta,h}(s, a) := \underset{x \in \mathcal{Y}_2^\theta}{\arg\min} |x - r_h(s, a)|, \qquad \forall (s, a, h) \in \mathcal{S} \times \mathcal{A} \times \llbracket H \rrbracket. \tag{7}$$

Crucially, note that $\mathcal{G}_{r_\theta} \subseteq \mathcal{Y}^\theta$ for any reward $r$, since summing discretized rewards always yields discretized values. Because $\mathcal{Y}^\theta$ has "small" (polynomial) size, $|\mathcal{Y}^\theta| = \mathcal{O}(H/\theta)$, the policy set $\Pi(r_\theta^E)$ is also small, $|\Pi(r_\theta^E)| \ll |\Pi^{\text{NM}}|$, and every policy in $\Pi(r_\theta^E)$, including $\pi_{r_\theta^E}$, can be efficiently stored (with $\mathcal{O}(SAH|\mathcal{Y}^\theta|)$ memory). The following lemma shows that the approximation error introduced by using policies in $\Pi(r_\theta^E)$ instead of $\Pi(r^E)$ for RDM can be tightly controlled by $\theta$ (see Appendix C.1.1 for the proof):

**Lemma 4.2.** *Let $\theta \in (0, 1]$. Let $\mathcal{M}_{r^E}$ be any MDP and $\pi^E \in \Pi^{\text{NM}}$ any policy. Then, the policy $\pi_{r_\theta^E} \in \Pi(r_\theta^E)$ satisfies $\mathcal{W}\big(\eta_{r^E}^{\pi_{r_\theta^E}}, \eta_{r^E}^{\pi^E}\big) \leq H\theta$.*

In short, Lemma 4.2 shows that *efficient and accurate* solutions to the RDM problem can be sought within $\Pi(r_\theta^E)$. In particular, $\Pi(r_\theta^E)$ contains $\pi_{r_\theta^E}$, whose error can be reduced by decreasing $\theta$, at the cost of increased memory requirements for storing the policy, which scale as $\mathcal{O}(1/\theta)$. In the next two sections, we show how to build efficient RDM algorithms based on $\Pi(r_\theta^E)$ and $\pi_{r_\theta^E}$.

## 4.2 NO-INTERACTION SETTING

In this section, we present **RS-BC** (**R**isk-**S**ensitive **B**ehavior **C**loning, Algorithm 1),[5] a provably efficient algorithm for RDM in the no-interaction (offline) setting, where we neither know nor have access to the transition model of the environment $\mathcal{M}$, and are instead given only a dataset $\mathcal{D}^E$ of $N$ expert trajectories together with the expert's reward $r^E$. The idea of **RS-BC** is simple: directly use $\mathcal{D}^E$ to estimate policy $\pi_{r_\theta^E}$, whose return distribution is guaranteed by Lemma 4.2 to be close to that of the expert. **RS-BC** estimates $\pi_{r_\theta^E}(a|s, \omega)$ as the fraction of the times in $\mathcal{D}^E$ that the expert took action $a$ in state $s$ after collecting $G(\omega; r_\theta^E)$ discretized cumulative reward (see Line 2). This "empirical" estimator follows closely the definition of $\pi_{r_\theta^E}$ in Eq. (6), with probability terms replaced by counts $M$ (computed at Line 1). The next result shows that **RS-BC** is sample-efficient by providing a worst-case upper bound on its sample complexity (proof in Appendix C.2.1):

**Theorem 4.3.** *Let $\epsilon \in (0, H]$ and $\delta \in (0, 1)$. Let $\mathcal{M}_{r^E}$ be any MDP and let $\pi^E \in \Pi^{NM}$ be any policy. Then, choosing $\theta = \epsilon/(4H)$, with probability at least $1 - \delta$, the policy $\hat{\pi}$ output by Algorithm 1*

---

[5]The "behavior cloning" part in **RS-BC** comes from the intuition that **RS-BC** can be seen as performing BC after augmenting the state space with the (discretized) cumulative reward.

---

**Algorithm 2: RS-KT (Risk-Sensitive imitation with Known Transition)**

---

**Input :** Dataset $\mathcal{D}^E = \{(s_1^i, a_1^i, \ldots, s_H^i, a_H^i)\}_{i \in [\![N]\!]}$, reward $r^E$, parameter $\theta$, transition model $p$

    `// Estimate the return distribution of the expert` $\eta_{r^E}^{\pi^E}$

1   $\widehat{\eta}(g) \leftarrow \frac{1}{N} \sum_{i \in [\![N]\!]} \mathbb{1}\{\sum_{h=1}^{H} r_{\theta,h}^E(s_h^i, a_h^i) = g\} \quad \forall g \in \mathcal{Y}^\theta$

    `// Compute the policy in` $\Pi(r_\theta^E)$ `closest to` $\widehat{\eta}$ `via Eq.` (10)

2   $\widehat{\pi} \in \arg\min_{\pi \in \Pi(r_\theta^E)} \mathcal{W}(\eta_{r_\theta^E}^\pi, \widehat{\eta})$

3   **Return** $\widehat{\pi}$

---

*satisfies $\mathcal{W}(\eta_{r^E}^{\pi^E}, \eta_{r^E}^{\widehat{\pi}}) \leqslant \epsilon$, with a number of samples:*

$$N \leqslant \widetilde{\mathcal{O}}\left( \frac{SH^6 \ln \frac{1}{\delta}}{\epsilon^3} \left( A + \ln \frac{1}{\delta} \right) \right). \tag{8}$$

In words, Theorem 4.3 shows that **RS-BC** requires a polynomial (in the quantities of interest $S, A, H, 1/\epsilon, \ln(1/\delta)$) number of samples to output a good imitation policy for RDM with high probability.[6] Compared to the best existing upper bound for standard IL, $\widetilde{\mathcal{O}}(SAH^3/\epsilon^2 \ln(1/\delta))$ (Corollary 3.1 of Foster et al. (2024)), we observe a gap of $\mathcal{O}(H^3/\epsilon \ln(1/\delta))$. This is reasonable, as RDM appears more complex than occupancy measure matching (e.g., it requires non-Markovian policies). We conjecture that the $\epsilon$ gap is unimprovable, while the $\mathcal{O}(H^6)$ dependence may be large but is comparable to the $\mathcal{O}(H^5)$ rate in the related setting of IL from observation alone (Theorem 3.3 of Sun et al. (2019)). See Appendix C.2.2 for further discussion. Finally, **RS-BC** is also computationally efficient, since both Lines 1–2 require only $\mathcal{O}(SAH|\mathcal{Y}^\theta|)$ iterations and memory.

## 4.3 KNOWN-TRANSITION SETTING

In this section, we present **RS-KT** (Risk-Sensitive imitation with Known Transition, Algorithm 2), a provably efficient algorithm for RDM in the known-transition setting, where we have access to the transition model $p$ of the environment, in addition to the expert's dataset $\mathcal{D}^E$ and reward $r^E$. By leveraging knowledge of $p$, **RS-KT** achieves a drastic reduction in sample complexity compared to **RS-BC**. The idea behind **RS-KT** is straightforward. First, use the expert dataset $\mathcal{D}^E$ to compute an estimate $\widehat{\eta}$ of the expert's return distribution $\eta_{r^E}^{\pi^E}$ (Line 1). Then, exploit knowledge of $r^E$ and $p$ to identify the policy in $\Pi(r_\theta^E)$ whose return distribution is closest to $\widehat{\eta}$ (Line 2). We now show that **RS-KT** is sample efficient (see Appendix C.3.1 for proof):

**Theorem 4.4.** *Let $\epsilon \in (0, H]$ and $\delta \in (0, 1)$. Let $\mathcal{M}_{r^E}$ be any MDP and $\pi^E \in \Pi^{NM}$ any policy. Assume that the optimization problem in Line 2 is solved exactly. Then, choosing $\theta = \epsilon/(7H)$, with probability $1 - \delta$, the policy $\widehat{\pi}$ output by Algorithm 2 satisfies $\mathcal{W}(\eta_{r^E}^{\pi^E}, \eta_{r^E}^{\widehat{\pi}}) \leqslant \epsilon$, with:*

$$N \leqslant \mathcal{O}\left( \frac{H^2}{\epsilon^2} \ln \frac{1}{\delta} \right). \tag{9}$$

Interestingly, Theorem 4.4 shows that **RS-KT** has sample complexity *independent* of $S$ and $A$, making it substantially more sample efficient than any algorithm for standard IL in large MDPs, where $\Omega(S)$ samples are required even with knowledge of $p$ (Theorem 5.1 of Rajaraman et al. (2020)). We believe the $\mathcal{O}(1/\epsilon^2)$ rate is tight, as it matches the lower bound for estimating a distribution in Wasserstein distance (Theorem 3.1 of Bobkov & Ledoux (2019)). Compared to **RS-BC**, the reduction in sample complexity is drastic, $\mathcal{O}(SAH^4/\epsilon)$. Extending **RS-KT** to settings where $p$ is unknown but can be estimated from online interaction with the environment is an interesting direction for future work (see Xu et al. (2023) for the standard IL case). If Line 2 is solved with error $\epsilon_{\text{apx}}$, then Theorem 4.4 guarantees $\mathcal{W}(\eta_{r^E}^{\pi^E}, \eta_{r^E}^{\widehat{\pi}}) \leqslant \epsilon + \epsilon_{\text{apx}}$. In distributional RL terms (Bellemare et al., 2023), **RS-KT** adopts a fixed-size categorical representation at Line 1. Finally, we refer the reader to Appendices C.4 and C.5 for extensions of **RS-BC** and **RS-KT** to the cases where $r^E$ is unknown but either belongs to a given finite set or is linear in a given feature map, and to Appendix C.6 for greater generalization.

---

[6]$\widetilde{\mathcal{O}}$ notation omits logarithmic terms in $S, A, H, 1/\epsilon, \ln(1/\delta)$.

We now turn to the computational complexity of **RS-KT**. Crucially, the optimization problem in Line 2 can be formulated as a linear program (LP) with a polynomial number $\mathcal{O}\big(SAH|\mathcal{Y}^\theta|\big)$ of variables and constraints, and therefore can be solved efficiently. To see this, let $\overline{\mathcal{M}} := (\overline{\mathcal{S}}, \mathcal{A}, H, \overline{s}_0, \overline{p})$ be the MDP\R with augmented state space $\overline{\mathcal{S}} := \mathcal{S} \times \mathcal{Y}^\theta$, initial state $\overline{s}_0 = (s_0, 0) \in \overline{\mathcal{S}}$, and transition model $\overline{p}$ defined as $\overline{p}_h(s, g|s', g', a) := p_h(s'|s, a)\mathbb{1}\{g = g' + r^E_{\theta, h}(s', a')\}$, i.e., the probability of reaching $(s, g) \in \overline{\mathcal{S}}$ by playing $a$ in $(s', g')$ at stage $h$ coincides with $p_h(s'|s, a)$ if the reward received in $s', a'$ is $g - g'$, and 0 otherwise. Since $\Pi(r^E_\theta)$ coincides with the set of Markovian policies in $\overline{\mathcal{M}}$ (Bäuerle & Rieder, 2014; Lazzati & Metelli, 2025), we can rewrite Line 2 as a variant of the occupancy measure matching problem in $\overline{\mathcal{M}}$:

$$\min_{d \in \mathcal{K}, \eta \in \Delta^{\mathcal{Y}^\theta}} \mathcal{W}(\eta, \widehat{\eta}) \quad \text{s.t.} \quad \eta(g) = \sum_{(s,a) \in \mathcal{S} \times \mathcal{A}} d_H(s, g - r^E_{\theta, H}(s, a), a) \quad \forall g \in \mathcal{Y}^\theta, \tag{10}$$

where the constraint enforces that $\eta$ is the return distribution induced by the occupancy measure $d$, and $\mathcal{K}$ denotes the set of feasible occupancy measures in $\overline{\mathcal{M}}$ (Puterman, 1994):

$$\mathcal{K} := \Big\{ d \in \Delta^{\overline{\mathcal{S}} \times \mathcal{A}}_{[\![H]\!]} \Big| \sum_a d_1(\overline{s}_0, a) = 1 \wedge \forall \overline{s} \in \overline{\mathcal{S}}, h \geqslant 2 : \sum_a d_h(\overline{s}, a) = \sum_{\overline{s}', a'} d_{h-1}(\overline{s}', a')\overline{p}_{h-1}(\overline{s}|\overline{s}', a') \Big\}.$$

In words, Eq. (10) searches for an occupancy measure $d \in \mathcal{K}$ that induces the return distribution $\eta$ closest to $\widehat{\eta}$. From such a solution, a policy $\widehat{\pi} \in \Pi(r^E_\theta)$ with occupancy $d^{\widehat{\pi}} = d$ (and thus return distribution $\eta^{\widehat{\pi}}_{r^E_\theta} = \eta$) can be recovered via

$$\widehat{\pi}(a|s, \omega) = \frac{d_h(s, G(\omega; r^E_\theta), a)}{\sum_{a'} d_h(s, G(\omega; r^E_\theta), a')} \quad \forall h \in [\![H]\!], \ s \in \mathcal{S}, \ a \in \mathcal{A}, \ \omega \in \Omega_h,$$

when the denominator is nonzero, and $\widehat{\pi}(a|s, \omega) = 1/A$ otherwise (Syed et al., 2008). Observe that all the constraints in Eq. (10) are linear in $d$ and $\eta$, and that the objective $\mathcal{W}(\eta, \widehat{\eta})$ can also be written linearly (see Peyré & Cuturi (2019) and Appendix C.3.2).

## 5 STATISTICAL INSIGHTS ON THE UNKNOWN-REWARD SETTING

In this section, we assume that the expert's reward $r^E$ is unknown, and provide some *statistical* insights on RDM in the "robust" form of Eq. (5). Specifically, we show that, perhaps surprisingly, this complex problem requires only a polynomial number of expert demonstrations to be accurately solved when the transition model is known, even in the *worst case*. To establish this, we first prove that a polynomial number of expert demonstrations suffices to accurately estimate the expert's return distribution *under any reward* (proof in Appendix D):

**Theorem 5.1.** *Let $\epsilon \in (0, H]$ and $\delta \in (0, 1)$. Let $\mathcal{M}_{r^E}$ be any MDP and $\pi^E \in \Pi^{NM}$ any policy. Then, choosing $\theta = \epsilon/(2H)$, a number of samples*

$$N \leqslant \widetilde{\mathcal{O}}\bigg( \frac{SAH^3}{\epsilon^2} \ln \frac{1}{\delta} \bigg), \tag{11}$$

*suffices to guarantee that, with probability at least $1 - \delta$, for the estimator $\widehat{\eta}_r(g) := \frac{1}{N} \sum_{\omega \in \mathcal{D}^E} \mathbb{1}\{G(\omega; r_\theta) = g\} \ \forall g, r$, we have:*

$$\max_{r: \mathcal{S} \times \mathcal{A} \times [\![H]\!] \to [0,1]} \mathcal{W}\Big( \eta^{\pi^E}_r, \widehat{\eta}_r \Big) \leqslant \epsilon.$$

In brief, an expert dataset $\mathcal{D}^E$ of size in Eq. (11) suffices to accurately estimate the expert's return distribution $\eta^{\pi^E}_r$ under any reward $r$ via the estimator $\widehat{\eta}_r$. As a consequence, any policy $\widehat{\pi}$ that induces return distributions close to $\widehat{\eta}_r$ for all possible rewards $r$ accurately solves the robust RDM problem:

**Theorem 5.2.** *Under the conditions of Theorem 5.1, assume access to a computational oracle that takes as input the dataset $\mathcal{D}^E$ and the transition model $p$, and outputs a solution to*

$$\widehat{\pi} \in \arg\min_{\pi \in \Pi^{NM}} \max_{r: \mathcal{S} \times \mathcal{A} \times [\![H]\!] \to [0,1]} \mathcal{W}\Big( \eta^\pi_r, \widehat{\eta}_r \Big). \tag{12}$$

*Then, with probability at least $1 - \delta$, using the number of samples in Eq. (11), it holds that*

$$\max_{r: \mathcal{S} \times \mathcal{A} \times [\![H]\!] \to [0,1]} \mathcal{W}\Big( \eta^{\pi^E}_r, \eta^{\widehat{\pi}}_r \Big) \leqslant 2\epsilon.$$

|  | $N = 20$ | $N = 80$ | $N = 300$ | $N = 1000$ | $N = 10000$ |
|---|---|---|---|---|---|
| **RS-BC** | **0.081±0.039** | **0.038±0.016** | **0.022±0.013** | **0.012±0.005** | **0.005±0.002** |
| **RS-KT** | **0.095±0.036** | **0.049±0.017** | **0.03±0.013** | **0.019±0.007** | **0.011±0.006** |
| BC | **0.099±0.056** | 0.076±0.054 | 0.072±0.056 | 0.069±0.058 | 0.068±0.058 |
| MIMIC-MD | 0.127±0.062 | 0.086±0.055 | 0.074±0.056 | 0.07±0.057 | 0.068±0.058 |

|  | $N = 20$ | $N = 80$ | $N = 300$ | $N = 1000$ | $N = 10000$ |
|---|---|---|---|---|---|
| **RS-BC** | **0.087±0.04** | **0.051±0.022** | **0.035±0.015** | **0.027±0.016** | **0.022±0.016** |
| **RS-KT** | 0.144±0.053 | 0.119±0.039 | 0.109±0.04 | 0.108±0.038 | 0.106±0.039 |
| BC | 0.103±0.057 | 0.08±0.053 | 0.072±0.056 | 0.069±0.058 | 0.068±0.058 |
| MIMIC-MD | 0.132±0.065 | 0.09±0.055 | 0.076±0.055 | 0.071±0.057 | 0.068±0.058 |

|  | $N = 20$ | $N = 80$ | $N = 300$ | $N = 1000$ | $N = 10000$ |
|---|---|---|---|---|---|
| **RS-BC** | 0.102±0.031 | 0.052±0.015 | **0.026±0.008** | **0.015±0.005** | **0.004±0.001** |
| **RS-KT** | 0.118±0.036 | 0.059±0.017 | 0.031±0.009 | 0.021±0.007 | **0.01±0.004** |
| BC | **0.085±0.035** | **0.041±0.016** | **0.021±0.008** | **0.012±0.005** | **0.003±0.002** |
| MIMIC-MD | 0.132±0.052 | 0.06±0.022 | 0.03±0.01 | **0.016±0.006** | **0.005±0.002** |

|  | $N = 20$ | $N = 80$ | $N = 300$ | $N = 1000$ | $N = 10000$ |
|---|---|---|---|---|---|
| **RS-BC** | 0.169±0.079 | 0.168±0.079 | 0.165±0.081 | 0.165±0.081 | 0.166±0.081 |
| BC | 0.168±0.078 | 0.166±0.078 | 0.169±0.085 | 0.177±0.091 | 0.174±0.093 |
| $\widehat{\eta}$ | 0.169±0.049 | **0.08±0.018** | **0.043±0.01** | **0.024±0.006** | **0.008±0.002** |

Table 1: Results of the simulations described in Section 6. The best results in each column are highlighted in bold. (Top) Simulation with $S, A, H = (2, 2, 5)$ for Q1. (Upper middle) Simulation with $\theta = 0.5$ for Q2. (Lower middle) Simulation with a Markovian expert for Q3. (Bottom) Simulation with $S, A, H = (300, 5, 5)$ for Q4.

In words, Theorem 5.2 establishes that the robust RDM problem is sample efficient for any algorithm that accurately solves Eq. (12). Intuitively, such an algorithm can be viewed as an extension of **RS-KT** to the unknown-reward setting. Note that the minimization in Eq. (12) is over $\Pi^{\mathrm{NM}}$, since a satisfactory policy may no longer exist in the policy class $\Pi(r_\theta^E)$, which is also unknown due to the unknown reward. We leave to future work the interesting question of whether an (approximate) solution to Eq. (12) can be computed efficiently, and note that restricting the optimization to a subset $\Pi \subset \Pi^{\mathrm{NM}}$ can reduce computation time at the cost of some misspecification error.

## 6 NUMERICAL SIMULATIONS

In this section, we study **RS-BC** and **RS-KT** from a practical perspective by conducting simulations aimed at answering the following four questions:[7]

1. What is the performance improvement of **RS-BC** and **RS-KT** on the RDM problem compared to standard IL algorithms?
2. How are the results affected by the choice of $\theta$?
3. What happens if the expert's policy is Markovian?
4. Does **RS-KT** truly reduce sample complexity compared to **RS-BC**, as predicted by theory?

We select BC (Foster et al., 2024) and MIMIC-MD (Rajaraman et al., 2020; 2021) as baseline IL algorithms for, respectively, the no-interaction and known-transition settings, and address all questions by conducting various simulations following the next three-step process. $(i)$ First, we randomly generate 50 MDPs and expert policies with fixed size $S, A, H$ (details in Appendix E.1). $(ii)$ Second, for each number of expert trajectories $N \in \{20, 80, 300, 1000, 10000\}$, we collect three different expert datasets of $N$ trajectories for each MDP, provide them as input to the four algorithms **RS-BC**, **RS-KT**, BC, and MIMIC-MD, and record the average (over the three seeds) Wasserstein distance

---

[7]The code for running our simulations can be found at https://github.com/filippolazzati/risk-IL.

between the expert's return distribution and the return distribution induced by each algorithm's output policy. $(iii)$ Finally, we average these distances across all 50 MDPs, obtaining, for each algorithm and value of $N$, a number representing the expected error of that algorithm when given access to $N$ expert trajectories in the considered setting.

Below, we describe the specific simulations conducted and the results obtained for each question. The numerical results are reported in Table 1.

**Question 1 (Q1).** To address Q1, we conducted three simulations with different problem sizes $S, A, H \in \{(2, 2, 5), (50, 5, 5), (2, 2, 20)\}$, all with non-Markovian policies and $\theta = 0.05$. The results for $(2, 2, 5)$ are reported in Table 1 (top), while the other two are shown in Tables 3–4 in Appendix E.6. Crucially, Table 1 (top) reveals that BC and MIMIC-MD, by relying on *Markovian* policies, are biased and cannot match the expert's return distribution satisfactorily, even with a large dataset of $N = 10000$ trajectories. In contrast, by leveraging our efficient non-Markovian policy class, **RS-BC** and **RS-KT** continue to reduce the error as the number of trajectories $N$ increases, up to a limit determined by our choice of $\theta$. A similar pattern is observed for larger $S, A$ in Table 3, and especially for larger horizons $H$ in Table 4, where the limited expressivity of Markovian policies becomes even more pronounced. We also note that larger $S, A, H$ slows down **RS-KT** significantly due to solving an LP with poly$(S, A, H)$ variables and constraints, and increases the approximation error due to $\theta$, as discussed in Appendix E.6 and in Q2.

**Question 2 (Q2).** To address Q2, we first conducted a simulation with an increased $\theta = 0.5$, keeping $S, A, H = (2, 2, 5)$ and a non-Markovian expert. As shown in Table 1 (upper middle), a larger $\theta$ consistently increases the approximation error, causing **RS-BC** and **RS-KT** to perform worse than with $\theta = 0.05$ (see Table 1 (top)). Nevertheless, while **RS-BC** remains fairly robust and continues to outperform BC and MIMIC-MD —intuitively because it corresponds to BC with a more expressive policy class—**RS-KT** tends to reach the worst-case approximation error predicted by Lemma 4.2, $H\theta/2 \approx 0.2$, as also discussed in Appendix E.6. Additionally, we conducted three further simulations with values of $\theta$ small enough to eliminate approximation error, observing a consistent increase in performance, particularly for **RS-KT** in settings with larger $S, A, H$ (see Appendix E.7).

**Question 3 (Q3).** We ran a simulation with a *Markovian* expert and $S, A, H = (2, 2, 5)$, with results reported in Table 1 (lower middle). Interestingly, BC outperforms all other algorithms in terms of both sample and computational efficiency. Intuitively, this occurs because it operates on a much smaller hypothesis space (i.e., $\Pi^{\mathrm{M}}$) than **RS-BC** and **RS-KT**, and it is *unbiased*, since the expert is Markovian. The policy output by BC aims to match the expert's trajectory distribution (Foster et al., 2024), and thus also its return distribution for any reward, which is not the case for MIMIC-MD, explaining its comparatively worse performance.

**Question 4 (Q4).** According to our theoretical results (Theorems 4.3 and 4.4), **RS-KT** should outperform **RS-BC** in terms of sample complexity for large $S, A$, as its performance does not depend on them. To verify this, we ran a simulation with large $S, A, H = (300, 5, 5)$, a non-Markovian expert, and $\theta = 0.05$. To speed up computation (particularly the LP in **RS-KT**), we avoided running **RS-KT** directly and instead compared the expert's return distribution with the estimate $\widehat{\eta}$ computed at Line 1. By the triangle inequality, this guarantees that the error between the output of **RS-KT** and the expert is at most twice the error of $\widehat{\eta}$. Results are reported in Table 1 (bottom), showing a dramatic improvement in sample complexity. In particular, both **RS-BC** and BC struggle even with $N = 10000$, while **RS-KT** achieves significant performance with as few as $N = 300$ or 1000 trajectories.

In summary, the key takeaways of this section are:

- **RS-BC** and **RS-KT** generally outperform BC and MIMIC-MD due to the use of more expressive non-Markovian policies and reward information, particularly for large $H$.

- **RS-BC** is faster than **RS-KT** because it does not require solving a linear program and is more robust to large $\theta$, while **RS-KT** is much more sample efficient for large MDPs.

- BC performs well when the expert is Markovian, even without access to reward information.

## 7 CONCLUSION

In this paper, we introduced and analyzed RDM, a general formulation of (risk-sensitive) IL as the problem of matching the expert's return distribution. Remarkably, we showed that both the known- and unknown-reward settings are statistically tractable. For the known-reward case, we proposed two algorithms, **RS-BC** and **RS-KT**, which not only come with strong theoretical guarantees but also empirically outperform standard IL methods at accurately matching the expert's return distribution.

**Limitations and future directions.** This work has several limitations. We focus only on the tabular setting, and our theoretical analysis lacks lower bounds, so it remains unclear whether the upper bounds in Theorems 4.3, 4.4, and 5.2 are tight. Furthermore, the unknown-reward setting lacks a practical algorithm, and our empirical study does not include real-world data. Future work could extend our results to state-only feedback settings (Sun et al., 2019), develop practical and scalable versions of **RS-BC** and **RS-KT** for large or continuous environments, and design algorithms for the unknown-reward setting.

### ETHICS STATEMENT

This work aims to develop learning machines that replicate human behavior not only in task performance but also in risk preferences. While AI research has extensively explored improving performance, aligning machines with human risk attitudes remains underexplored and raises important ethical considerations. We envision a future where autonomous agents assist humans in everyday tasks, from driving to cooking. Because uncertainty is inherent to human life, agents must account for risk to behave safely and effectively. Embedding risk-sensitivity is therefore essential for supporting human well-being, but it must be approached with care to minimize potential harm. Being foundational research, this work does not pose direct societal risks. Nevertheless, we recognize that ethical considerations will be crucial as future applications of risk-sensitive agents move closer to real-world deployment.

### REPRODUCIBILITY STATEMENT

All theoretical results in this work are accompanied by formal proofs in the appendix. In the main text, immediately before or after each theorem, lemma, or proposition, we provide a reference to the specific appendix section containing the formal proof, as well as additional discussion on the implications of the result when present. For the numerical simulations, Appendix E contains all supplementary details not included in the main text, along with additional simulation results. We include with this submission a zip file containing all code required to replicate the experiments, as well as .npy files with the exact simulation results. Following publication, we will make a publicly accessible repository containing all code and data, and we will reference it in the main text.

### LLM USAGE

We used LLMs in paper writing only to fix potential grammar mistakes and improve clarity.

### ACKNOWLEDGMENTS

This publication was funded with the contribution of Ministero dell'Università e della ricerca pursuant to D.D. n. 7206 of 17 April 2025 - BANDO FIS 2. Project FIS-2023-02598 (Starting Grant), title: "Unified Learning from Diverse Human Feedback" (HUmLrn). CUP: D53C25000710001.

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

# A  ADDITIONAL RELATED WORK

**Standard IL.**   There are two main approaches to address the standard occupancy measure matching formulation of IL: Behavior Cloning (BC), that treats IL as a supervised learning problem, directly learning a mapping from states to actions (Pomerleau, 1988; Ross & Bagnell, 2010), and Inverse Reinforcement Learning (IRL) that infers a reward function that reflects the expert's preferences, and then derives a policy via planning (Ng & Russell, 2000; Abbeel & Ng, 2004; Ziebart, 2010; Finn et al., 2016; Fu et al., 2017). Other popular occupancy measure matching methods include Ho & Ermon (2016); Kostrikov et al. (2019); Reddy et al. (2020); Brantley et al. (2020); Garg et al. (2021); Dadashi et al. (2021). Recently, there have been various efforts into providing theoretical guarantees on BC (Rajaraman et al., 2020; Foster et al., 2024) and IRL (Metelli et al., 2021; 2023), and also for related IL algorithms (Sun et al., 2019; Viano et al., 2024). In this paper we provide a theoretical analysis analogous to that in Rajaraman et al. (2020); Foster et al. (2024), but for the novel return distribution matching setting.

**Risk-sensitive RL.**   The first paper to address risk sensitivity in Markov Decision Processes (MDPs) is Howard & Matheson (1972). Since then, various researchers have explored this problem (see the survey A. & Fu (2022) and the recent book Bellemare et al. (2023)). Notably, Mannor & Tsitsiklis (2011) emphasizes the importance of non-Markovian policies in mean-variance optimization. Additionally, Bäuerle & Ott (2011) and Bäuerle & Rieder (2014) show that optimal behavior in CVaR and expected utility planning problems can generally be achieved using non-Markovian policies that base actions on both the current state and the cumulative reward. This idea is exploited in Section 4 to construct our algorithms. Some risk-sensitive RL algorithms employing such non-Markovian policies include Haskell & Jain (2015); Chow et al. (2018). In addition, we mention Fei et al. (2020; 2021), which study the risk-sensitive RL problem from a theoretical perspective, by providing provably efficient risk-sensitive RL algorithms under the entropic risk measure. Bastani et al. (2022); Wang et al. (2023) provide analogous regret analysis for the CVaR objective. Lastly, recent Chen et al. (2024); Liang & Luo (2024) consider a theoretical distributional RL analysis for risk-sensitive RL.

**Risk-Sensitive IL.**   Some works extend occupancy measure matching with additional objectives to capture the expert's risk attitude. Santara et al. (2018) and Lacotte et al. (2019) are most similar to ours, as they aim to match both expected return and CVaR. However, as noted in the introduction, they restrict to Markovian policies, which limits their ability to fully capture the expert's risk preferences. Ratliff & Mazumdar (2020) propose a risk-sensitive IRL algorithm focused on learning risk parameters, also assuming a Markovian expert. Majumdar et al. (2017) study risk-sensitive IL with a form of non-Markovian policy, but in environments far simpler than tabular MDPs. Lazzati & Metelli (2025) consider non-Markovian experts and imitation, but primarily aim to recover the expert's utility, and their IL approach struggles when demonstrations come from a single environment. Nevertheless, our policy class in Section 4 draws inspiration on their discretization approach. Muni et al. (2025) also mentions the importance of non-Markovian policies for risk-sensitive IL, but do not present any algorithm.

**Non-Markovian IL.**   The importance of adopting non-Markovian policies for IL has already been recognized by some IL works, especially in the field of robotics. Mandlekar et al. (2020; 2022) identify one cause of non-Markovianity in human demonstrations in partial observability, and consider variants of BC with recurrent neural networks for imitation. Zhao et al. (2023) and subsequent literature (e.g., see Torne et al. (2025); Ren et al. (2025)), learn non-Markovian policies implicitly through action chunking, an open-loop control technique in which at each state our policy outputs a "chunk" (i.e., a sequence) of actions. We mention Qin et al. (2023), which learn non-Markovian policies as energy-based priors from state-only sequences, and Block et al. (2023), which study imitation in non-linear systems. Importantly, none of these works address non-Markovian policies arising from risk sensitivity, which is a novel aspect of ours. Moreover, note that fitting general non-Markovian policies may require an amount of data exponential in the horizon in the worst-case (see Appendix B.3).

## B ADDITIONAL RESULTS AND PROOFS FOR SECTION 3

In this appendix, we first show that matching the return distribution in Wasserstein distance is strictly more expressive than matching the expectation or the CVaR at a given level (Appendix B.1). Next, we provide the proof of other results in Section 3 (Appendix B.2). Finally, we prove that matching the trajectory distribution of arbitray non-Markovian policies is sample inefficient (Appendix B.3).

### B.1 ADDITIONAL INSIGHTS ON RDM

We show here that matching the expert's return distribution in Wasserstein distance implies closeness in terms of expected return, the variance of the return, and the CVaR at any level.

First, observe that, for any MDP $\mathcal{M}_{r^E}$ and policies $\pi^E, \hat{\pi}$:

$$J_{r^E}^{\pi^E} - J_{r^E}^{\hat{\pi}} \leqslant \mathcal{W}\big(\eta_{r^E}^{\pi^E}, \eta_{r^E}^{\hat{\pi}}\big).$$

This follows after having realized that the identity function (denoted below as $I(\cdot)$) is $1-$Lipschitz, and using the dual form of the Wasserstein distance (see Eq. 6.3 of Villani (2008)).

$$\begin{aligned}
J_{r^E}^{\pi^E} - J_{r^E}^{\hat{\pi}} &= \mathbb{E}_{X \sim \eta_{r^E}^{\pi^E}}[X] - \mathbb{E}_{Y \sim \eta_{r^E}^{\hat{\pi}}}[Y] \\
&= \mathbb{E}_{X \sim \eta_{r^E}^{\pi^E}}[I(X)] - \mathbb{E}_{Y \sim \eta_{r^E}^{\hat{\pi}}}[I(Y)] \\
&\leqslant \sup_{f: \|f\|_{\text{Lip}} \leqslant 1} \mathbb{E}_{X \sim \eta_{r^E}^{\pi^E}}[f(X)] - \mathbb{E}_{Y \sim \eta_{r^E}^{\hat{\pi}}}[f(Y)] \\
&= \mathcal{W}\big(\eta_{r^E}^{\pi^E}, \eta_{r^E}^{\hat{\pi}}\big).
\end{aligned}$$

Second, for any $\alpha \in (0, 1)$, we have:

$$|\text{CVaR}_\alpha(\eta_{r^E}^{\pi^E}) - \text{CVaR}_\alpha(\eta_{r^E}^{\hat{\pi}})| \leqslant \frac{1}{\alpha} \mathcal{W}\big(\eta_{r^E}^{\pi^E}, \eta_{r^E}^{\hat{\pi}}\big).$$

This follows using an alternative expression for the Wasserstein distance. Formally, for any pair of distributions $p, q$ with support on $[0, H]$:

$$\begin{aligned}
\big|\text{CVaR}_\alpha(p) - \text{CVaR}_\alpha(q)\big| &= \left|\frac{1}{\alpha} \int_0^\alpha \Big(F_p^{-1}(x) - F_q^{-1}(x)\Big) dx\right| \\
&\leqslant \frac{1}{\alpha} \int_0^\alpha \Big|F_p^{-1}(x) - F_q^{-1}(x)\Big| dx \\
&\leqslant \frac{1}{\alpha} \int_0^1 \Big|F_p^{-1}(x) - F_q^{-1}(x)\Big| dx \\
&\overset{(1)}{=} \frac{1}{\alpha} \mathcal{W}(p, q),
\end{aligned}$$

where at (1) we use that the 1-Wasserstein distance can be written this way (Panaretos & Zemel, 2019), where recall that $F_p^{-1}(x) := \inf_{z \in \mathbb{R}: F_p(z) \geqslant x} z$.

Lastly, we observe that closedness in Wasserstein distance also implies closeness between the variance of returns:

$$|\text{var}(\eta_{r^E}^{\pi^E}) - \text{var}(\eta_{r^E}^{\hat{\pi}})| \leqslant 4H \cdot \mathcal{W}\big(\eta_{r^E}^{\pi^E}, \eta_{r^E}^{\hat{\pi}}\big).$$

To see it, for any pair of distributions $p, q$ with support on $[0, H]$, we can write:

$$\begin{aligned}
&|\text{var}(p) - \text{var}(q)| \\
&= \Big|\mathbb{E}_{X \sim p}[X^2] - (\mathbb{E}_{X \sim p}[X])^2 - \mathbb{E}_{Y \sim q}[Y^2] + (\mathbb{E}_{Y \sim q}[Y])^2\Big| \\
&\leqslant \Big|\mathbb{E}_{X \sim p}[X^2] - \mathbb{E}_{Y \sim q}[Y^2]\Big| + \Big|(\mathbb{E}_{X \sim p}[X])^2 - (\mathbb{E}_{Y \sim q}[Y])^2\Big| \\
&\overset{(1)}{\leqslant} 2H \sup_{f: \|f\|_{\text{Lip}} \leqslant 1} \Big|\mathbb{E}_{X \sim p}[f(X)] - \mathbb{E}_{Y \sim q}[f(Y)]\Big| \\
&\quad + \Big|(\mathbb{E}_{X \sim p}[X] - \mathbb{E}_{Y \sim q}[Y]) \cdot (\mathbb{E}_{X \sim p}[X] + \mathbb{E}_{Y \sim q}[Y])\Big|
\end{aligned}$$

$$\overset{(2)}{\leqslant} 2H \cdot \mathcal{W}(p,q) + 2H\big|\mathbb{E}_{X \sim p}[X] - \mathbb{E}_{Y \sim q}[Y]\big|$$

$$\overset{(3)}{\leqslant} 4H \cdot \mathcal{W}(p,q),$$

where at (1) we use that the function $f : x \to x^2$ is $2H-$Lipschitz on $[0, H]$ (since $\sup_{x \neq y} \frac{|f(x)-f(y)|}{|x-y|} = \sup_{x \neq y} \frac{|x^2-y^2|}{|x-y|} = \sup_{x \neq y}(x+y) = 2H$ for $x, y \in [0, H]$), and we rescaled to obtain functions that are $1-$Lipschitz, at (2) we use the dual form of the Wasserstein distance (see Eq. 6.3 of Villani (2008)) after having noticed that the absolute value can be removed as set $\{f : \|f\|_{\text{Lip}} \leqslant 1\}$ is symmetric, and we upper bounded $\mathbb{E}_{X \sim p}[X] + \mathbb{E}_{Y \sim q}[Y] \leqslant 2H$. Finally, at (3) we use the result proved earlier.

## B.2 PROOFS

**Theorem 3.1.** *There exist an MDP $\mathcal{M}_{r^E}$ with $S, A, H \geqslant 2$ and an expert policy $\pi^E \in \Pi^{NM}$ such that, even with $N = (S-1)^{H-1} - 1$ trajectories, any algorithm $\mathfrak{A}$ satisfies*

$$\mathbb{E}_{\mathcal{D}^E \sim \mathbb{P}^{\pi^E}} TV\Big(\eta_{r^E}^{\pi^E}, \eta_{r^E}^{\widehat{\pi}}\Big) \geqslant \frac{1}{2e},$$

*where $\widehat{\pi}$ is the output of $\mathfrak{A}$ given in input $\mathcal{M}_{r^E}$ and $\mathcal{D}^E$.*

*Proof.* To prove this result, we simply provide a reward function $r^E : \mathcal{S} \times \mathcal{A} \times [\![H]\!] \to [0,1]$ for which, in the "hard" MDP used in the proof of Theorem B.1, we have:

$$\text{TV}\Big(\eta_{r^E}^{\pi^E}, \eta_{r^E}^{\widehat{\pi}}\Big) = \frac{1}{2}\Big\|\mathbb{P}^{\pi^E} - \mathbb{P}^{\widehat{\pi}}\Big\|_1,$$

for any pair of policies. Then, the claim in the theorem follows directly from the result in Theorem B.1.

First, let us associate a different integer $0, 1, \ldots, S-1$ to each state $\{s_1, \ldots, s_S\}$, and a different integer $0, 1, \ldots, A-1$ to each action $\{a_1, \ldots, a_A\}$, and denote $x_s : \mathcal{S} \to \mathbb{N}$ and $x_a : \mathcal{A} \to \mathbb{N}$ these mappings. Then, the reward that we provide is the following:

$$r_h^E(s,a) := 10^{-hSA - x_s(s)A - x_a(a)}.$$

Simply, observe that every triple $s, a, h$ is associated a reward value belonging to a different power of 10, thus when we sum them to compute the return of trajectory we obtain a different return value for every possible trajectory:

$$\begin{aligned}
\text{TV}\Big(\eta_{r^E}^{\pi^E}, \eta_{r^E}^{\widehat{\pi}}\Big) &:= \frac{1}{2}\sum_g \Big|\eta_{r^E}^{\pi^E}(g) - \eta_{r^E}^{\widehat{\pi}}(g)\Big| \\
&= \frac{1}{2}\sum_g \Big|\sum_{\omega : G(\omega; r^E)=g} \mathbb{P}^{\pi^E}(\omega) - \mathbb{P}^{\widehat{\pi}}(\omega)\Big| \\
&= \frac{1}{2}\sum_\omega \Big|\mathbb{P}^{\pi^E}(\omega) - \mathbb{P}^{\widehat{\pi}}(\omega)\Big| \\
&= \frac{1}{2}\Big\|\mathbb{P}^{\pi^E} - \mathbb{P}^{\widehat{\pi}}\Big\|_1.
\end{aligned}$$

This concludes the proof. $\square$

**Proposition 3.2.** *There exist an MDP $\mathcal{M}_{r^E}$ with horizon $H = 3$ and an expert policy $\pi^E \in \Pi^{NM}$ such that* any *Markovian policy $\pi \in \Pi^M$ satisfies*

$$\mathcal{W}\Big(\eta_{r^E}^{\pi^E}, \eta_{r^E}^{\pi}\Big) \geqslant 0.5.$$

*Proof.* Consider the MDP $\mathcal{M}_{r^E} = (\mathcal{S}, \mathcal{A}, H, s_0, p, r^E)$ in Fig. 1, where $\mathcal{S} = \{s_{\text{init}}, s_1, s_2, s_3\}$, $\mathcal{A} = \{a_1, a_2\}$, $H = 3$, $s_0 = s_{\text{init}}$, the transition model $p$ is such that:

$$p_1(s_1|s_{\text{init}}, a) = p_1(s_2|s_{\text{init}}, a) = 1/2 \quad \forall a \in \mathcal{A},$$

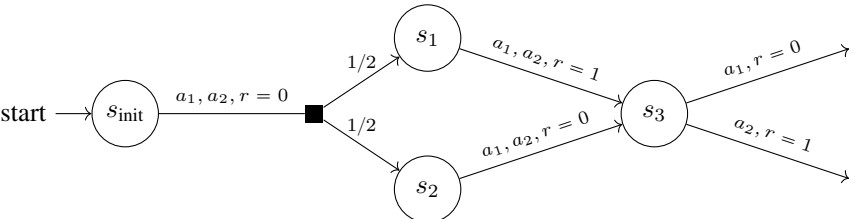

Figure 1: MDP for the proof of Proposition 3.2.

$$p_2(s_3|s_1, a) = p_2(s_3|s_2, a) = 1 \quad \forall a \in \mathcal{A},$$
$$p_3(s_3|s_3, a) = 1,$$

and the reward function $r^E$ is defined as:

$$r_1^E(s_{\text{init}}, a) = 0 \quad \forall a \in \mathcal{A},$$
$$r_2^E(s_1, a) = 1 \quad \forall a \in \mathcal{A},$$
$$r_2^E(s_2, a) = 0 \quad \forall a \in \mathcal{A},$$
$$r_3^E(s_3, a_1) = 0,$$
$$r_3^E(s_3, a_2) = 1.$$

Let $\pi^E$ be the deterministic non-Markovian policy that plays always action $a_1$ every time it is not in $s_3$, and then in $s_3$ plays $a_1$ if previously we passed through $s_1$, otherwise play $a_2$. Formally, for any $\omega \in \Omega$ and $s \in \mathcal{S}$:

$$\pi^E(a_1|s, \omega) = \begin{cases} 1 & \text{if } s \neq s_3 \vee (s = s_3 \wedge \omega = (s_{\text{init}}, a_1, s_1, a_1)) \\ 0 & \text{otherwise} \end{cases}.$$

It is easy to note that the return distribution of $\pi^E$ in $\mathcal{M}_{r^E}$ is:

$$\eta_{r^E}^{\pi^E} = \delta_1,$$

where notation $\delta_x$ denotes the Dirac delta on $x$. With $\delta_1$, we always get return 1.

Now, let $\pi^\alpha$ be any Markovian policy parameterized by $\alpha \in [0, 1]$ as (we do not specify values in other states and stages, because they are not relevant for the return distribution):

$$\pi_3^\alpha(a_1|s_3) = \alpha.$$

The return distribution of $\pi^\alpha$ in $\mathcal{M}_{r^E}$ is:

$$\eta_{r^E}^{\pi^\alpha} = \frac{\alpha}{2}\delta_0 + \frac{1}{2}\delta_1 + \frac{1 - \alpha}{2}\delta_2,$$

namely, irrespective of the policy $\pi^\alpha$, the return distribution is 1 w.p. $1/2$.

Let us compute the Wasserstein distance between these return distributions:

$$\mathcal{W}\left(\eta_{r^E}^{\pi^E}, \eta_{r^E}^{\pi^\alpha}\right) = \int_{-\infty}^{+\infty} \left| F_{\eta_{r^E}^{\pi^E}}(g) - F_{\eta_{r^E}^{\pi^\alpha}}(g) \right| dg$$
$$= 1 \cdot \left| 0 - \frac{\alpha}{2} \right| + 1 \cdot \left| 1 - \frac{1 + \alpha}{2} \right|$$
$$= \frac{1}{2},$$

irrespective of $\alpha$. This concludes the proof.

$\square$

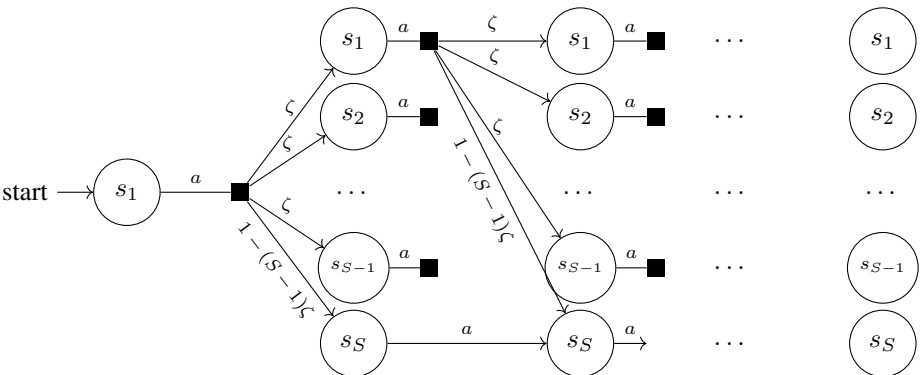

Figure 2: The MDP\R $\mathcal{M}$ used in the proof of Theorem B.1.

## B.3 STATISTICAL INEFFICIENCY OF GENERAL NON-MARKOVIAN POLICIES

In this appendix, we provide an explicit proof that the problem of matching the *distribution over trajectories* of an arbitrary non-Markovian expert's policies may require an exponential (in the horizon) amount of data. Then, the proof of Theorem 3.1 can be simply obtained through a reduction to this problem. The lower bound is provided in the next theorem, which makes use of the family of hard instances in Fig. 2.

**Theorem B.1.** *There exist an MDP\R $\mathcal{M} = (\mathcal{S}, \mathcal{A}, H, s_0, p)$ with $S \geqslant 2$, $A \geqslant 2$, $H \geqslant 2$ and a non-Markovian expert's policy $\pi^E \in \Pi^{NM}$, such that any learning algorithm $\mathfrak{A}$ taking in input a dataset of $N = (S-1)^{H-1} - 1$ expert's trajectories $\mathcal{D}^E = \{\omega_i\}_{i \in [\![N]\!]}$ satisfies:*

$$\mathbb{E}_{\mathcal{D}^E \sim \mathbb{P}^{\pi^E}} \left\| \mathbb{P}^{\pi^E} - \mathbb{P}^{\widehat{\pi}} \right\|_1 \geqslant \frac{1}{e},$$

*where $\widehat{\pi}$ is the policy outputted by $\mathfrak{A}$ when taking in input both $\mathcal{D}^E$ and $\mathcal{M}$.*

*Proof.* The proof draws inspiration from that of Theorem 5.1 in Rajaraman et al. (2020) for the known-transition setting.

We begin by describing the class of hard instances that will be considered for proving this lower bound. Note that we are considering here the "IL" problem of computing a policy with trajectory distribution close to that of the expert. Thus, a problem instance is a pair MDP\R -expert's policy. Let $\mathfrak{P} = \{(\mathcal{M}, \pi_i^E)\}_i$ be the family of problem instances where the MDP\R $\mathcal{M} = (\mathcal{S}, \mathcal{A}, H, s_0, p)$ is represented in Fig. 2, and the expert's policy $\pi_i^E$ is any deterministic policy that for $h < H$ always plays $a_1$, while at the last stage can play an arbitrary action. Formally, $\mathcal{S} \supseteq \{s_1, s_S\}$ (i.e., it has $S \geqslant 2$ at least $s_1, s_S$, and potentially other states $s_2, \dots$), $\mathcal{A} = \{a_1, a_2\}$, $H \geqslant 2$, $s_0 = s_1$, and the transition model $p$ is defined as a function of a scalar $\zeta \in [0, \frac{1}{S-1}]$ that we will choose later:

$$p_h(s'|s, a) = \begin{cases} \zeta & \text{if } s \neq s_S \wedge s' \neq s_S \\ 1 - (S-1)\zeta & \text{if } s \neq s_S \wedge s' = s_S \ , \\ 1 & \text{otherwise} \end{cases}$$

for any action $a \in \mathcal{A}$. Then, the expert's policy $\pi_i^E$ is defined as any policy in $\Pi^{\text{hard}}$, defined as:

$$\Pi^{\text{hard}} := \left\{ \pi \in \Pi^{\text{NM}} \,\middle|\, \forall h \in [\![H-1]\!], \forall s \in \mathcal{S}, \forall \omega \in \Omega_h : \pi(a_1|s, \omega) = 1 \right\}.$$

Then, $\mathfrak{P}$ is formally defined as:

$$\mathfrak{P} := \left\{ (\mathcal{M}', \pi') \,\middle|\, \mathcal{M}' = \mathcal{M} \wedge \pi' \in \Pi^{\text{hard}} \right\}.$$

We set $\zeta := \frac{1}{(N+1)^{1/(H-1)}}$, and observe that, to guarantee that $\mathcal{M}$ exists, we need to enforce that $(S-1)\zeta$, i.e., the total probability assigned to reaching a state $\neq s_S$, is smaller than 1. So:

$$(S-1)\zeta \leqslant 1 \iff \frac{S-1}{(N+1)^{\frac{1}{H-1}}} \leqslant 1 \iff N \geqslant (S-1)^{H-1} - 1.$$

To proceed, we choose a distribution $\mathcal{P}$ over problem instances in $\mathfrak{P}$. We select $\mathcal{P} \in \Delta^{\mathfrak{P}}$ as the uniform probability distribution over the family of problems $\mathfrak{P}$, i.e.:

$$\forall (\mathcal{M}', \pi') \in \mathfrak{P}: \ \mathcal{P}(\mathcal{M}', \pi') = \frac{1}{|\mathfrak{P}|}.$$

Then, if we can show that:

$$\mathbb{E}_{(\mathcal{M}', \pi') \sim \mathcal{P}} \mathbb{E}_{\mathcal{D}^E \sim \mathbb{P}^{\pi'}} \left\| \mathbb{P}^{\pi'} - \mathbb{P}^{\widehat{\pi}} \right\|_1 \geq \frac{(S-1)^{H-1}}{e(N+1)},$$

then we can conclude that there is at least a problem instance $(\overline{\mathcal{M}}, \overline{\pi}) \in \mathfrak{P}$ such that:

$$\mathbb{E}_{\mathcal{D}^E \sim \mathbb{P}^{\overline{\pi}}} \left\| \mathbb{P}^{\overline{\pi}} - \mathbb{P}^{\widehat{\pi}} \right\|_1 \geq \frac{(S-1)^{H-1}}{e(N+1)},$$

from which if we insert the choice $N = (S-1)^{H-1} - 1$, we obtain the claim of the theorem.

To this aim, we can write:

$$\mathbb{E}_{(\mathcal{M}', \pi') \sim \mathcal{P}} \mathbb{E}_{\mathcal{D}^E \sim \mathbb{P}^{\pi'}} \left\| \mathbb{P}^{\pi'} - \mathbb{P}^{\widehat{\pi}} \right\|_1$$

$$:= \mathbb{E}_{(\mathcal{M}', \pi') \sim \mathcal{P}} \mathbb{E}_{\mathcal{D}^E \sim \mathbb{P}^{\pi'}} \sum_{\omega \in \Omega_{H+1}} \left| \mathbb{P}^{\pi'}(\omega) - \mathbb{P}^{\widehat{\pi}}(\omega) \right|$$

$$\overset{(1)}{=} \mathbb{E}_{\mathcal{D}^E \sim q} \mathbb{E}_{(\mathcal{M}', \pi') \sim \mathcal{P}'(\mathcal{D}^E)} \sum_{\omega \in \Omega_{H+1}} \left| \mathbb{P}^{\pi'}(\omega) - \mathbb{P}^{\widehat{\pi}}(\omega) \right|$$

$$= \mathbb{E}_{\mathcal{D}^E \sim q} \mathbb{E}_{(\mathcal{M}', \pi') \sim \mathcal{P}'(\mathcal{D}^E)} \sum_{s^1, a^1, \ldots, s^H, a^H} \left| \mathbb{P}^{\pi'}(s^1, a^1, \ldots, s^H, a^H) \right.$$

$$\left. - \mathbb{P}^{\widehat{\pi}}(s^1, a^1, \ldots, s^H, a^H) \right|$$

$$\overset{(2)}{\geq} \mathbb{E}_{\mathcal{D}^E \sim q} \mathbb{E}_{(\mathcal{M}', \pi') \sim \mathcal{P}'(\mathcal{D}^E)} \sum_{s^2, s^3, \ldots, s^H} \sum_{a \in \mathcal{A}} \left| \mathbb{P}^{\pi'}(s_1, a_1, s^2, a_1, \ldots, s^H, a) \right.$$

$$\left. - \mathbb{P}^{\widehat{\pi}}(s_1, a_1, s^2, a_1, \ldots, s^H, a) \right|$$

$$\overset{(3)}{=} \mathbb{E}_{\mathcal{D}^E \sim q} \mathbb{E}_{(\mathcal{M}', \pi') \sim \mathcal{P}'(\mathcal{D}^E)} \sum_{s^2, s^3, \ldots, s^H} \sum_{a \in \mathcal{A}} \left| \rho(s^2) \cdot \rho(s^3) \cdot \ldots \cdot \rho(s^H) \right.$$

$$\left. \cdot \pi'(a | s_1, a_1, s^2, a_1, \ldots, s^H) - \mathbb{P}^{\widehat{\pi}}(s_1, a_1, s^2, a_1, \ldots, s^H, a) \right|$$

$$\overset{(4)}{=} \mathbb{E}_{\mathcal{D}^E \sim q} \mathbb{E}_{(\mathcal{M}', \pi') \sim \mathcal{P}'(\mathcal{D}^E)} \sum_{s^2, s^3, \ldots, s^H} \sum_{a \in \mathcal{A}} \left| \rho(s^2) \cdot \rho(s^3) \cdot \ldots \cdot \rho(s^H) \right.$$

$$\cdot \pi'(a | s_1, a_1, s^2, a_1, \ldots, s^H) - \rho(s^2) \cdot \rho(s^3) \cdot \ldots \cdot \rho(s^H)$$

$$\left. \cdot \widehat{\pi}(a | s_1, a_1, s^2, a_1, \ldots, s^H) \cdot \prod_{h \in [\![H-1]\!]} \widehat{\pi}(a_1 | s_1, a_1, s^2, a_1, \ldots, s^h) \right|$$

$$\overset{(5)}{=} \mathbb{E}_{\mathcal{D}^E \sim q} \mathbb{E}_{(\mathcal{M}', \pi') \sim \mathcal{P}'(\mathcal{D}^E)} \sum_{s^2, s^3, \ldots, s^H} \rho(s^2) \cdot \rho(s^3) \cdot \ldots \cdot \rho(s^H) \sum_{a \in \mathcal{A}} \Big|$$

$$\pi'(a | s_1, a_1, s^2, a_1, \ldots, s^H) - \widehat{\pi}(a | s_1, a_1, s^2, a_1, \ldots, s^H) \cdot K \Big|$$

$$= \mathbb{E}_{\mathcal{D}^E \sim q} \sum_{s^2, s^3, \ldots, s^H} \rho(s^2) \cdot \rho(s^3) \cdot \ldots \cdot \rho(s^H) \mathbb{E}_{(\mathcal{M}', \pi') \sim \mathcal{P}'(\mathcal{D}^E)} \sum_{a \in \mathcal{A}} \Big|$$

$$\pi'(a | s_1, a_1, s^2, a_1, \ldots, s^H) - \widehat{\pi}(a | s_1, a_1, s^2, a_1, \ldots, s^H) \cdot K \Big|$$

$$\geq \mathbb{E}_{\mathcal{D}^E \sim q} \sum_{s^2, s^3, \ldots, s^H} \rho(s^2) \cdot \rho(s^3) \cdot \ldots \cdot \rho(s^H) \mathbb{1}\{(s_1, a_1, s^2, a_1, \ldots, s^H) \notin \mathcal{D}^E\}$$

$$\mathbb{E}_{(\mathcal{M}', \pi') \sim \mathcal{P}'(\mathcal{D}^E)} \sum_{a \in \mathcal{A}} \left| \pi'(a | s_1, a_1, s^2, a_1, \ldots, s^H) \right.$$

$$- \widehat{\pi}(a|s_1, a_1, s^2, a_1, \ldots, s^H) \cdot K \Big|$$

$$\overset{(6)}{=} \mathbb{E}_{\mathcal{D}^E \sim q} \sum_{s^2, s^3, \ldots, s^H} \rho(s^2) \cdot \rho(s^3) \cdot \ldots \cdot \rho(s^H) \mathbb{1}\{(s_1, a_1, s^2, a_1, \ldots, s^H) \notin \mathcal{D}^E\}$$

$$\sum_{b \in \mathcal{A}} \frac{1}{A} \sum_{a \in \mathcal{A}} \Big| \mathbb{1}\{a = b\} - \widehat{\pi}(a|s_1, a_1, s^2, a_1, \ldots, s^H) \cdot K \Big|$$

$$= \mathbb{E}_{\mathcal{D}^E \sim q} \sum_{s^2, s^3, \ldots, s^H} \rho(s^2) \cdot \rho(s^3) \cdot \ldots \cdot \rho(s^H) \mathbb{1}\{(s_1, a_1, s^2, a_1, \ldots, s^H) \notin \mathcal{D}^E\}$$

$$\cdot \sum_{b \in \mathcal{A}} \frac{1}{A} \Bigg( \Big( 1 - \widehat{\pi}(b|s_1, a_1, s^2, a_1, \ldots, s^H) \cdot K \Big)$$

$$+ \sum_{a \in \mathcal{A} \backslash \{b\}} \widehat{\pi}(a|s_1, a_1, s^2, a_1, \ldots, s^H) \cdot K \Bigg)$$

$$= \mathbb{E}_{\mathcal{D}^E \sim q} \sum_{s^2, s^3, \ldots, s^H} \rho(s^2) \cdot \rho(s^3) \cdot \ldots \cdot \rho(s^H) \mathbb{1}\{(s_1, a_1, s^2, a_1, \ldots, s^H) \notin \mathcal{D}^E\}$$

$$\cdot \Bigg( 1 + \frac{K}{A} \sum_{b \in \mathcal{A}} \Big( \sum_{a \in \mathcal{A} \backslash \{b\}} \widehat{\pi}(a|s_1, a_1, s^2, a_1, \ldots, s^H) $$

$$- \widehat{\pi}(b|s_1, a_1, s^2, a_1, \ldots, s^H) \Big) \Bigg)$$

$$= \mathbb{E}_{\mathcal{D}^E \sim q} \sum_{s^2, s^3, \ldots, s^H} \rho(s^2) \cdot \rho(s^3) \cdot \ldots \cdot \rho(s^H) \mathbb{1}\{(s_1, a_1, s^2, a_1, \ldots, s^H) \notin \mathcal{D}^E\}$$

$$\cdot \Bigg( 1 + \frac{K}{A} \sum_{b \in \mathcal{A}} \Big( 1 - 2\widehat{\pi}(b|s_1, a_1, s^2, a_1, \ldots, s^H) \Big) \Bigg)$$

$$= \mathbb{E}_{\mathcal{D}^E \sim q} \sum_{s^2, s^3, \ldots, s^H} \rho(s^2) \cdot \rho(s^3) \cdot \ldots \cdot \rho(s^H) \mathbb{1}\{(s_1, a_1, s^2, a_1, \ldots, s^H) \notin \mathcal{D}^E\}$$

$$\cdot \big( 1 + K - 2K/A \big)$$

$$\overset{(7)}{\geqslant} \mathbb{E}_{\mathcal{D}^E \sim q} \sum_{s^2, s^3, \ldots, s^H} \rho(s^2) \cdot \rho(s^3) \cdot \ldots \cdot \rho(s^H) \mathbb{1}\{(s_1, a_1, s^2, a_1, \ldots, s^H) \notin \mathcal{D}^E\}$$

$$\overset{(8)}{=} \sum_{s^2, s^3, \ldots, s^H} \rho(s^2) \cdot \rho(s^3) \cdot \ldots \cdot \rho(s^H) q\big((s_1, a_1, s^2, a_1, \ldots, s^H) \notin \mathcal{D}^E\big)$$

$$\overset{(9)}{=} \sum_{s^2, s^3, \ldots, s^H} \rho(s^2) \cdot \rho(s^3) \cdot \ldots \cdot \rho(s^H) \Big( 1 - \rho(s^2) \cdot \rho(s^3) \cdot \ldots \cdot \rho(s^H) \Big)^N$$

$$\geqslant \sum_{\substack{s^2, s^3, \ldots, s^H : \\ s^h \neq s_S \forall h \in \{2, \ldots, H\}}} \rho(s^2) \cdot \rho(s^3) \cdot \ldots \cdot \rho(s^H) \Big( 1 - \rho(s^2) \cdot \rho(s^3) \cdot \ldots \cdot \rho(s^H) \Big)^N$$

$$= \sum_{\substack{s^2, s^3, \ldots, s^H \\ s^h \neq s_S \forall h \in \{2 \ldots, H\}}} \frac{1}{N+1} \Big( 1 - \frac{1}{N+1} \Big)^N$$

$$\overset{(10)}{\geqslant} \frac{1}{e} \sum_{\substack{s^2, s^3, \ldots, s^H \\ s^h \neq s_S \forall h \in \{2 \ldots, H\}}} \frac{1}{N+1}$$

$$= \frac{1}{e} \frac{(S-1)^{H-1}}{N+1},$$

where at (1) we define $q \in \Delta^{\Omega_{H+1}}$ as a distribution that generates datasets $\mathcal{D}^E$ as if we sampled first $(\mathcal{M}'', \pi'') \sim \mathcal{P}$, and then we collected $N$ trajectories from $\pi''$ in $\mathcal{M}''$, and we define $\mathcal{P}'(\mathcal{D}^E)$ as the

uniform distribution over the set of deterministic policies:

$$\Pi^{\text{mimic}}(\mathcal{D}^E) := \left\{ \pi \in \Pi^{\text{hard}} \, \middle| \, \forall h \in [\![H-1]\!], \forall \omega \in \Omega_h^s(\mathcal{D}^E) : \pi(\omega) = \pi^{\mathcal{D}^E}(\omega) \right\},$$

where $\Omega_h^s(\mathcal{D}^E)$ denotes the set of trajectories $(s_1, a_1, \ldots, s_{h-1}, a_{h-1}, s_h)$ visited in some trajectory in $\mathcal{D}^E$, and $\pi^{\mathcal{D}^E}(\omega)$ denotes the corresponding action present in $\mathcal{D}^E$, and $\pi(\omega)$ denotes the deterministic action taken by $\pi$. At (2) we lower bound by considering the error only for the trajectories in which at all $h < H$ the action played is $a_1$, and we note that in $\mathcal{M}' = \mathcal{M}$, the initial state is always $s_1$. At (3) we define distribution $\rho \in \Delta^{\mathcal{S}}$ over states as $\rho(s) = \zeta$ if $s \neq s_S$, and $\rho(s_S) = 1 - (S-1)\zeta$. Note that since any policy $\pi'' \in \Pi^{\text{hard}}$ plays action $a_1$ for $h < H$, then we set this probability to 1. At (4) we rewrite also $\mathbb{P}^{\hat{\pi}}$ using $\rho$ and chain rule of conditional probabilities. At (5) we bring the $\rho$ terms outside and define $K := \prod_{h \in [\![H-1]\!]} \hat{\pi}(a_1|s_1, a_1, s^2, a_1, \ldots, s^h)$ for brevity. At (6) we use that $\mathcal{P}'(\mathcal{D}^E)$ gives always $\mathcal{M}' = \mathcal{M}$, and that it gives, when the trajectory is not observed (i.e., it is not in $\mathcal{D}^E$), with equal weight, policies that play any action $b \in \mathcal{A}$ given trajectory $s_1, a_1, s^2, a_1, \ldots, s^H$. Thus, since there is no dependence on the actions assigned by such policies given different trajectories, then the expectation simplifies to just $\sum_{b \in \mathcal{A}} \frac{1}{A}(\ldots)$. Then, note that, using this notation, $\pi'(a|s_1, a_1, s^2, a_1, \ldots, s^H) = \mathbb{1}\{a = b\}$. At (7) we use that $K$ is a product of probabilities, thus it lies in $[0, 1]$, and so, since for $A \geq 2$ we have $1 - 2/A = (A-2)/A \geq 0$, then the quantity is lower bounded by using $K = 0$. At (8) we bring the expectation inside and use the fact that the expectation of the indicator is the probability. At (9) we realize that the probability of a trajectory does not depend on the specific policy (inside $\Pi^{\text{hard}}$) that generated it. Thus, we can replace $q$ using $\rho$, and use the complementary of an event of a Binomial distribution. At (10) we proceed as in Lemma A.21 of Rajaraman et al. (2020) by lower bounding the exponential term by $1/e$. $\qquad\square$

## C ADDITIONAL RESULTS AND PROOFS FOR SECTION 4

In Appendix C.1, we collect additional results and proofs for Section 4.1, while in Appendices C.2-C.3, we collect those for, respectively, Sections 4.2-4.3. Lastly, in Appendices C.4-C.5, we extend our results on **RS-BC** and **RS-KT** result to the settings in which, respectively, $r^E$ is known to belong to a given finite set $\mathcal{R}$ and it is known to be linear in a known feature map $\phi$. Finally, in Appendix C.6, we sketch how to extend **RS-BC** to arbitrary IL problems.

### C.1 ON THE CLASS OF POLICIES IN SECTION 4.1

In Appendix C.1.1, we report the proof of Lemmas 4.1 and 4.2, while in Appendix C.1.2 we discuss the size of the policy class $\Pi(r^E)$.

#### C.1.1 PROOFS

**Lemma 4.1.** *Let $\mathcal{M}_{r^E}$ be any MDP and let $\pi^E \in \Pi^{NM}$ be any policy. Then, the policy $\pi_{r^E} \in \Pi(r^E)$ satisfies $\eta_{r^E}^{\pi_{r^E}}(g) = \eta_{r^E}^{\pi^E}(g)$ for all $g \in [0, H]$.*

*Proof.* Thanks to Lemma C.2, we know that:

$$\mathbb{P}^{\pi_{r^E}}\left(G_H = g \wedge s_H = s\right) = \mathbb{P}^{\pi^E}\left(G_H = g \wedge s_H = s\right) \qquad \forall g \in [0, H-1], \forall s \in \mathcal{S},$$

where $G_H := \sum_{h'=1}^{H-1} r_{h'}^E(s_{h'}, a_{h'})$ denotes the random return at stage $H$. Then, for any $g' \in [0, H]$, we can write:

$$\eta_{r^E}^{\pi_{r^E}}(g') = \mathbb{P}^{\pi_{r^E}}(G_H + r_H^E(s_H, a_H) = g')$$

$$\overset{(1)}{=} \sum_{g \in \mathcal{G}_{r^E,H}} \sum_{s \in \mathcal{S}} \mathbb{P}^{\pi_{r^E}}(G_H = g, s_H = s, r_H^E(s, a_H) = g' - g)$$

$$\overset{(2)}{=} \sum_{g \in \mathcal{G}_{r^E,H}} \sum_{s \in \mathcal{S}} \sum_{a \in \mathcal{A}: r_H^E(s,a) = g'-g} \mathbb{P}^{\pi_{r^E}}(G_H = g, s_H = s)\pi_{r^E}(a|s, g)$$

$$\overset{(3)}{=} \sum_{g \in \mathcal{G}_{r^E,H}} \sum_{s \in \mathcal{S}} \sum_{a \in \mathcal{A}: r_H^E(s,a) = g'-g} \mathbb{P}^{\pi^E}(G_H = g, s_H = s)\pi_{r^E}(a|s, g)$$

$$
\begin{aligned}
&= \sum_{g \in \mathcal{G}_{r^E,H}} \sum_{s \in \mathcal{S}} \mathbb{P}^{\pi^E}(G_H = g, s_H = s) \sum_{a \in \mathcal{A}: r_H^E(s,a) = g'-g} \pi_{r^E}(a|s,g) \\
&\overset{(4)}{=} \sum_{g \in \mathcal{G}_{r^E,H}} \sum_{s \in \mathcal{S}} \mathbb{P}^{\pi^E}(G_H = g, s_H = s) \sum_{a \in \mathcal{A}: r_H^E(s,a) = g'-g} \frac{\mathbb{P}^{\pi^E}(G_H = g, s_H = s, a_H = a)}{\mathbb{P}^{\pi^E}(G_H = g, s_H = s)} \\
&= \sum_{g \in \mathcal{G}_{r^E,H}} \sum_{s \in \mathcal{S}} \sum_{a \in \mathcal{A}} \mathbb{P}^{\pi^E}(G_H = g, s_H = s, a_H = a) \mathbb{1}\{r_H^E(s,a) = g'-g\} \\
&= \mathbb{P}^{\pi^E}\Big( \sum_{h'=1}^{H} r_{h'}^E(s_{h'}, a_{h'}) = g' \Big) \\
&= \eta_{r^E}^{\pi^E}(g'),
\end{aligned}
$$

where at (1) we define symbol $\mathcal{G}_{r,H} := \{g \in [0, H-1] \,|\, \exists \omega \in \Omega_H : G(\omega; r^E) = g\}$, at (2) we recognize that, by definition, $\pi_{r^E}$ takes actions only depending on the current state, stage and past rewards, and we denote with brevity this fact with $\pi_{r^E}(a|s,g)$, at (3) we use the aforementioned result from Lemma C.2, at (4) we use the definition of $\pi_{r^E}(a|s,g)$ where the denominator is not 0, noting that, in that case, the entire expression evaluates to 0, □

**Lemma 4.2.** *Let $\theta \in (0, 1]$. Let $\mathcal{M}_{r^E}$ be any MDP and $\pi^E \in \Pi^{NM}$ any policy. Then, the policy $\pi_{r_\theta^E} \in \Pi(r_\theta^E)$ satisfies $\mathcal{W}\big( \eta_{r^E}^{\pi_{r_\theta^E}}, \eta_{r^E}^{\pi^E} \big) \leqslant H\theta$.*

*Proof.* We can write:

$$
\begin{aligned}
\mathcal{W}\Big( \eta_{r^E}^{\pi_{r_\theta^E}}, \eta_{r^E}^{\pi^E} \Big) &\overset{(1)}{\leqslant} \mathcal{W}\Big( \eta_{r^E}^{\pi_{r_\theta^E}}, \eta_{r_\theta^E}^{\pi_{r_\theta^E}} \Big) + \mathcal{W}\Big( \eta_{r_\theta^E}^{\pi_{r_\theta^E}}, \eta_{r_\theta^E}^{\pi^E} \Big) + \mathcal{W}\Big( \eta_{r_\theta^E}^{\pi^E}, \eta_{r^E}^{\pi^E} \Big) \\
&\overset{(2)}{\leqslant} 2H\|r^E - r_\theta^E\|_\infty + \mathcal{W}\Big( \eta_{r_\theta^E}^{\pi_{r_\theta^E}}, \eta_{r_\theta^E}^{\pi^E} \Big) \\
&\overset{(3)}{\leqslant} H\theta + \mathcal{W}\Big( \eta_{r_\theta^E}^{\pi_{r_\theta^E}}, \eta_{r_\theta^E}^{\pi^E} \Big) \\
&\overset{(4)}{\leqslant} H\theta,
\end{aligned}
$$

where at (1) we apply twice the triangle's inequality, at (2) we apply twice Lemma C.1, at (3) we realize that, by definition of $r_\theta^E$, it holds that $\|r^E - r_\theta^E\|_\infty \leqslant \theta/2$, and finally, at (4), we apply Lemma 4.1 with reward $r_\theta^E$ and expert's policy $\pi^E$. □

**Lemma C.1.** *Let $\mathcal{M}$ be any MDP\R and let $\pi \in \Pi^{NM}$ be any policy. Let $r^1, r^2 : \mathcal{S} \times \mathcal{A} \times [\![H]\!] \to [0,1]$ be any pair of reward functions. Then, it holds that:*

$$
\mathcal{W}(\eta_{r^1}^\pi, \eta_{r^2}^\pi) \leqslant H\|r^1 - r^2\|_\infty.
$$

*Proof.* We can write:

$$
\begin{aligned}
\mathcal{W}(\eta_{r^1}^\pi, \eta_{r^2}^\pi) &\overset{(1)}{=} \sup_{f:\|f\|_{\mathrm{Lip}} \leqslant 1} \Big( \int_{[0,H]} f \, d\eta_{r^1}^\pi - \int_{[0,H]} f \, d\eta_{r^2}^\pi \Big) \\
&\overset{(2)}{=} \sup_{f:\|f\|_{\mathrm{Lip}} \leqslant 1} \Big( \sum_{g \in \mathrm{supp}(\eta_{r^1}^\pi)} f(g)\eta_{r^1}^\pi(g) - \sum_{g \in \mathrm{supp}(\eta_{r^2}^\pi)} f(g)\eta_{r^2}^\pi(g) \Big) \\
&\overset{(3)}{=} \sup_{f:\|f\|_{\mathrm{Lip}} \leqslant 1} \Big( \sum_{g \in \mathrm{supp}(\eta_{r^1}^\pi)} f(g) \sum_{\substack{\omega \in \Omega_{H+1}: \\ G(\omega;r^1)=g}} \mathbb{P}^\pi(\omega) \\
&\qquad\qquad\qquad - \sum_{g \in \mathrm{supp}(\eta_{r^2}^\pi)} f(g) \sum_{\substack{\omega \in \Omega_{H+1}: \\ G(\omega;r^2)=g}} \mathbb{P}^\pi(\omega) \Big) \\
&= \sup_{f:\|f\|_{\mathrm{Lip}} \leqslant 1} \Big( \sum_{\omega \in \Omega_{H+1}} f(G(\omega;r^1))\mathbb{P}^\pi(\omega)
\end{aligned}
$$

$$- \sum_{\omega \in \Omega_{H+1}} f(G(\omega; r^2)) \mathbb{P}^\pi(\omega) \Big)$$

$$= \sup_{f: \|f\|_{\text{Lip}} \leqslant 1} \sum_{\omega \in \Omega_{H+1}} \mathbb{P}^\pi(\omega) \Big( f(G(\omega; r^1)) - f(G(\omega; r^2)) \Big)$$

$$\leqslant \sup_{f: \|f\|_{\text{Lip}} \leqslant 1} \sum_{\omega \in \Omega_{H+1}} \mathbb{P}^\pi(\omega) \Big| f(G(\omega; r^1)) - f(G(\omega; r^2)) \Big|$$

$$\overset{(4)}{\leqslant} \sum_{\omega \in \Omega_{H+1}} \mathbb{P}^\pi(\omega) \Big| G(\omega; r^1) - G(\omega; r^2) \Big|$$

$$\overset{(5)}{=} \sum_{\omega \in \Omega_{H+1}} \mathbb{P}^\pi(\omega) \Big| \sum_{h \in [\![H]\!]} \Big( r_h^1(s_h, a_h) - r_h^2(s_h, a_h) \Big) \Big|$$

$$\leqslant \sum_{\omega \in \Omega_{H+1}} \mathbb{P}^\pi(\omega) \sum_{h \in [\![H]\!]} \Big| r_h^1(s_h, a_h) - r_h^2(s_h, a_h) \Big|$$

$$\leqslant \sum_{\omega \in \Omega_{H+1}} \mathbb{P}^\pi(\omega) \sum_{h \in [\![H]\!]} \max_{(s,a) \in \mathcal{S} \times \mathcal{A}} \Big| r_h^1(s, a) - r_h^2(s, a) \Big|$$

$$= \sum_{h \in [\![H]\!]} \max_{(s,a) \in \mathcal{S} \times \mathcal{A}} \Big| r_h^1(s, a) - r_h^2(s, a) \Big| \cdot \sum_{\omega \in \Omega_{H+1}} \mathbb{P}^\pi(\omega)$$

$$= \sum_{h \in [\![H]\!]} \max_{(s,a) \in \mathcal{S} \times \mathcal{A}} \Big| r_h^1(s, a) - r_h^2(s, a) \Big|$$

$$\leqslant H \max_{(s,a,h) \in \mathcal{S} \times \mathcal{A} \times [\![H]\!]} \Big| r_h^1(s, a) - r_h^2(s, a) \Big|,$$

where at (1) we apply the duality formula for the Wasserstein distance (see Eq. 6.3 in Section 6 of Villani (2008)). Note that we interpret the return distributions $\eta_{r^1}^\pi$ and $\eta_{r^2}^\pi$ as probability measures, and that a function $f : \mathbb{R} \to \mathbb{R}$ is L-Lipschitz (i.e., $\|f\|_{\text{Lip}} = L$) if, for all $x, y \in \mathbb{R}$, we have $|f(x) - f(y)| \leqslant L|x - y|$. At (2) we realize that $\eta_{r^1}^\pi$ and $\eta_{r^2}^\pi$ have finite supports, since in a tabular MDP with deterministic reward the number of trajectories is finite, and, thus, also the number of corresponding returns must be finite. We denote by $\text{supp}(\cdot)$ the support of a distribution. At (3) we use the definition of return distributions, at (4) we use the definition of Lipschitz functions, at (5) we denote by $(s_1, a_1, \ldots, s_H, a_H)$ the state-action trajectory $\omega \in \Omega_{H+1}$, and we use the definition of operators $G(\cdot; r^1)$ and $G(\cdot; r^2)$. $\qquad \square$

**Lemma C.2.** *Let $\mathcal{M}_r$ be any MDP and let $\pi \in \Pi^{NM}$ be any expert's policy. Let $\pi_r \in \Pi(r)$ be the policy defined as in Eq. (6) for expert's policy $\pi$. Then, for all $h \in [\![H]\!]$, $s \in \mathcal{S}$ and $g \in [0, h-1]$, it holds that:*

$$\mathbb{P}^{\pi_r} \Big( \sum_{h'=1}^{h-1} r_{h'}(s_{h'}, a_{h'}) = g \wedge s_h = s \Big) = \mathbb{P}^\pi \Big( \sum_{h'=1}^{h-1} r_{h'}(s_{h'}, a_{h'}) = g \wedge s_h = s \Big).$$

*Proof.* For simplicity, let us denote by $G_h$ the *random* return at time step $h$ under reward $r$. Formally:

$$G_h := \sum_{h'=1}^{h-1} r_{h'}(s_{h'}, a_{h'}) \quad \forall h \in [\![H]\!].$$

In this way, the claim of the theorem can be easily rewritten as:

$$\mathbb{P}^{\pi_r} \Big( G_h = g \wedge s_h = s \Big) = \mathbb{P}^\pi \Big( G_h = g \wedge s_h = s \Big).$$

We prove the result by induction. Let us begin with the base case: $h = 1$. For all $s \in \mathcal{S}$ and $g \in \{0\}$, we have:

$$\mathbb{P}^{\pi_r} \Big( G_1 = g \wedge s_1 = s \Big) = \mathbb{1}\{g = 0\} \mathbb{1}\{s = s_0\} = \mathbb{P}^\pi \Big( G_1 = g \wedge s_1 = s \Big),$$

where we noticed that, for $h = 1$, no action is taken yet. Now, let us consider any stage $h \in \{2, 3, \ldots, H\}$, and let us make the induction hypothesis that, for all $h' \in [\![h-1]\!]$, for all $s \in \mathcal{S}$ and $g \in [0, h'-1]$, it holds that:

$$\mathbb{P}^{\pi_r}\Big(G_{h'} = g \wedge s_{h'} = s\Big) = \mathbb{P}^{\pi}\Big(G_{h'} = g \wedge s_{h'} = s\Big).$$

Then, for any $s' \in \mathcal{S}$ and $g' \in [0, h-1]$, we can write:

$\mathbb{P}^{\pi_r}(G_h = g' \wedge s_h = s')$

$$\overset{(1)}{=} \sum_{\substack{\omega \in \Omega_{h-1}, (s,a) \in \mathcal{S} \times \mathcal{A}: \\ G(\omega;r) + r_{h-1}(s,a) = g'}} \mathbb{P}^{\pi_r}\big(\omega_{h-1} = \omega \wedge s_{h-1} = s \wedge a_{h-1} = a \wedge s_h = s'\big)$$

$$\overset{(2)}{=} \sum_{g \in \mathcal{G}_{r,h-1}} \sum_{\substack{\omega \in \Omega_{h-1}: \\ G(\omega;r) = g}} \sum_{\substack{(s,a) \in \mathcal{S} \times \mathcal{A}: \\ r_{h-1}(s,a) = g'-g}} \mathbb{P}^{\pi_r}\big(\omega_{h-1} = \omega \wedge s_{h-1} = s \wedge a_{h-1} = a \wedge s_h = s'\big)$$

$$\overset{(3)}{=} \sum_{g \in \mathcal{G}_{r,h-1}} \sum_{\substack{\omega \in \Omega_{h-1}: \\ G(\omega;r) = g}} \sum_{\substack{(s,a) \in \mathcal{S} \times \mathcal{A}: \\ r_{h-1}(s,a) = g'-g}} \mathbb{P}^{\pi_r}\big(\omega_{h-1} = \omega \wedge s_{h-1} = s\big)$$
$$\cdot \mathbb{P}^{\pi_r}\big(a_{h-1} = a | \omega, s\big) \mathbb{P}^{\pi_r}\big(s_h = s' | \omega, s, a\big)$$

$$\overset{(4)}{=} \sum_{g \in \mathcal{G}_{r,h-1}} \sum_{\substack{\omega \in \Omega_{h-1}: \\ G(\omega;r) = g}} \sum_{\substack{(s,a) \in \mathcal{S} \times \mathcal{A}: \\ r_{h-1}(s,a) = g'-g}} \mathbb{P}^{\pi_r}\big(\omega_{h-1} = \omega \wedge s_{h-1} = s\big)$$
$$\cdot \mathbb{P}^{\pi_r}\big(a_{h-1} = a | \omega, s\big) p_{h-1}(s' | s, a)$$

$$\overset{(5)}{=} \sum_{g \in \mathcal{G}_{r,h-1}} \sum_{\substack{\omega \in \Omega_{h-1}: \\ G(\omega;r) = g}} \sum_{\substack{(s,a) \in \mathcal{S} \times \mathcal{A}: \\ r_{h-1}(s,a) = g'-g}} \mathbb{P}^{\pi_r}\big(\omega_{h-1} = \omega \wedge s_{h-1} = s\big)$$
$$\cdot \pi_r(a | \omega, s) p_{h-1}(s' | s, a)$$

$$\overset{(6)}{=} \sum_{g \in \mathcal{G}_{r,h-1}} \sum_{\substack{(s,a) \in \mathcal{S} \times \mathcal{A}: \\ r_{h-1}(s,a) = g'-g}} \sum_{\substack{\omega \in \Omega_{h-1}: \\ G(\omega;r) = g}} \mathbb{P}^{\pi_r}\big(\omega_{h-1} = \omega \wedge s_{h-1} = s\big)$$
$$\cdot \pi_r(a | \omega, s) p_{h-1}(s' | s, a)$$

$$\overset{(7)}{=} \sum_{g \in \mathcal{G}_{r,h-1}} \sum_{\substack{(s,a) \in \mathcal{S} \times \mathcal{A}: \\ r_{h-1}(s,a) = g'-g}} \pi_r(a | g, s) p_{h-1}(s' | s, a) \sum_{\substack{\omega \in \Omega_{h-1}: \\ G(\omega;r) = g}} \mathbb{P}^{\pi_r}\big(\omega_{h-1} = \omega \wedge s_{h-1} = s\big)$$

$$= \sum_{g \in \mathcal{G}_{r,h-1}} \sum_{\substack{(s,a) \in \mathcal{S} \times \mathcal{A}: \\ r_{h-1}(s,a) = g'-g}} \pi_r(a | g, s) p_{h-1}(s' | s, a) \mathbb{P}^{\pi_r}\big(G_{h-1} = g \wedge s_{h-1} = s\big)$$

$$\overset{(8)}{=} \sum_{g \in \mathcal{G}_{r,h-1}} \sum_{\substack{(s,a) \in \mathcal{S} \times \mathcal{A}: \\ r_{h-1}(s,a) = g'-g}} \pi_r(a | g, s) p_{h-1}(s' | s, a) \mathbb{P}^{\pi}\big(G_{h-1} = g \wedge s_{h-1} = s\big)$$

$$\overset{(9)}{=} \sum_{g \in \mathcal{G}_{r,h-1}} \sum_{\substack{(s,a) \in \mathcal{S} \times \mathcal{A}: \\ r_{h-1}(s,a) = g'-g}} \frac{\mathbb{P}^{\pi}\big(G_{h-1} = g \wedge s_{h-1} = s \wedge a_{h-1} = a\big)}{\mathbb{P}^{\pi}\big(G_{h-1} = g \wedge s_{h-1} = s\big)}$$
$$\cdot p_{h-1}(s' | s, a) \mathbb{P}^{\pi}\big(G_{h-1} = g \wedge s_{h-1} = s\big)$$

$$= \sum_{g \in \mathcal{G}_{r,h-1}} \sum_{\substack{(s,a) \in \mathcal{S} \times \mathcal{A}: \\ r_{h-1}(s,a) = g'-g}} \mathbb{P}^{\pi}\big(G_{h-1} = g \wedge s_{h-1} = s \wedge a_{h-1} = a\big) p_{h-1}(s' | s, a)$$

$$= \sum_{g \in \mathcal{G}_{r,h-1}} \sum_{\substack{(s,a) \in \mathcal{S} \times \mathcal{A}: \\ r_{h-1}(s,a) = g'-g}} \mathbb{P}^{\pi}\big(G_{h-1} = g \wedge s_{h-1} = s \wedge a_{h-1} = a \wedge s_h = s'\big)$$

$$= \sum_{g \in \mathcal{G}_{r,h-1}} \mathbb{P}^{\pi}\big(G_{h-1} = g \wedge r_{h-1}(s_{h-1}, a_{h-1}) = g' - g \wedge s_h = s'\big)$$

$$= \mathbb{P}^\pi(G_h = g' \wedge s_h = s'),$$

where at (1) we use symbol $\omega_{h''}$ to denote the random trajectory long $h''$ stages, i.e., whose realizations belong to $\Omega_{h''}$, for any $h'' \in [\![H]\!]$. At (2) we define symbols $\mathcal{G}_{r,h} := \{g \in [0, h-1] \,|\, \exists \omega \in \Omega_h : G(\omega; r) = g\}$ for any $h \in [\![H]\!]$, denoting the set of possible values of cumulative reward obtainable at stage $h$, and we sum over all such values (note that they are finite in tabular MDPs with deterministic rewards), at (3) we use the chain rule of conditional probabilities, at (4) we use the Markovianity of the environment, at (5) we note that $\mathbb{P}^{\pi_r}(a_{h-1} = a|\omega, s)$ actually is $\pi_r(a|\omega, s)$, at (6) we exchange the two summations, at (7) we recognize that, by definition, $\pi_r(a|\omega, s)$ takes on the same value for all the trajectories $\omega$ with the same value of return, and thus we can bring this quantity outside the summation over the $\omega$. We use symbol $\pi_r(a|g, s)$ to denote this fact for brevity. We do the same also for $p_{h-1}(s'|s, a)$. At (8) we use the induction hypothesis, at (9) we replace $\pi_r(a|g, s)$ with its definition when $\mathbb{P}^\pi(G_{h-1} = g \wedge s_{h-1} = s) > 0$ as in the opposite case the entire formula takes on value zero. $\qquad\square$

### C.1.2 $\Pi(r^E)$ IS TOO LARGE FOR SOME $r^E$

Consider any reward $r^E : \mathcal{S} \times \mathcal{A} \times [\![H]\!] \to [0, 1]$ that assigns, to every possible trajectory $\omega \in \Omega$, a different return value $G(\omega; r^E) \in [0, H-1]$. Observe that rewards $r^E$ of this kind exist as the number of possible trajectories $|\Omega| = \mathcal{O}((SA)^{H-1})$ is finite, while the set of possible return values that we can assign to each $s, a, h$ is infinite (continuous) $[0, 1]$. An example of a reward of this kind was provided in the proof of Theorem 3.1:

$$r_h^E(s, a) = 10^{-hSA - x_s(s)A - x_a(a)} \qquad \forall(s, a, h) \in \mathcal{S} \times \mathcal{A} \times [\![H]\!],$$

where $x_s : \mathcal{S} \to \mathbb{N}, x_a : \mathcal{A} \to \mathbb{N}$ are arbitrary injective functions mapping the set of states and actions to the set of natural numbers. Intuitively, given the return $G(\omega; r^E)$ of any trajectory:

$$\omega = (s_1, a_1, s_2, a_2, \ldots, s_{h-1}, a_{h-1}),$$

we can easily reconstruct which $s, a, h$ actually belong to $\omega$ by checking which decimal numbers in $G(\omega; r^E)$ are "flagged".

Given a reward $r^E$ of this kind, we have that $|\mathcal{G}_{r^E}|$ is in the same order as $|\Omega| = \mathcal{O}((SA)^{H-1})$, and so the set of policies $\Pi(r^E)$ defined in Section 4.1 reduces to the whole set of non-Markovian policies $\Pi^{NM}$, with all its disadvantages.

### C.2 NO INTERACTION SETTING

In Appendix C.2.1, we prove Theorem 4.3, while in Appendix C.2.2 we provide additional discussion on Theorem 4.3.

### C.2.1 PROOF OF THEOREM 4.3

**Theorem 4.3.** *Let $\epsilon \in (0, H]$ and $\delta \in (0, 1)$. Let $\mathcal{M}_{r^E}$ be any MDP and let $\pi^E \in \Pi^{NM}$ be any policy. Then, choosing $\theta = \epsilon/(4H)$, with probability at least $1 - \delta$, the policy $\widehat{\pi}$ output by Algorithm 1 satisfies $\mathcal{W}(\eta_{r^E}^{\pi^E}, \eta_{r^E}^{\widehat{\pi}}) \leqslant \epsilon$, with a number of samples:*

$$N \leqslant \widetilde{\mathcal{O}}\left( \frac{SH^6 \ln\frac{1}{\delta}}{\epsilon^3} \left( A + \ln\frac{1}{\delta} \right) \right). \tag{8}$$

*Proof.* We can write:

$$\mathcal{W}\left(\eta_{r^E}^{\pi^E}, \eta_{r^E}^{\widehat{\pi}}\right) \overset{(1)}{\leqslant} \mathcal{W}\left(\eta_{r^E}^{\pi^E}, \eta_{r_\theta^E}^{\pi_{r^E}}\right) + \mathcal{W}\left(\eta_{r_\theta^E}^{\pi_{r^E}}, \eta_{r_\theta^E}^{\pi_{r^E}}\right) + \mathcal{W}\left(\eta_{r_\theta^E}^{\pi_{r^E}}, \eta_{r_\theta^E}^{\widehat{\pi}}\right) + \mathcal{W}\left(\eta_{r_\theta^E}^{\widehat{\pi}}, \eta_{r^E}^{\widehat{\pi}}\right)$$

$$\overset{(2)}{\leqslant} 2H\theta + \mathcal{W}\left(\eta_{r_\theta^E}^{\pi_{r^E}}, \eta_{r_\theta^E}^{\widehat{\pi}}\right)$$

$$\overset{(3)}{\leqslant} 2H\theta + H\left\| \eta_{r_\theta^E}^{\pi_{r^E}} - \eta_{r_\theta^E}^{\widehat{\pi}} \right\|_1$$

$$\overset{(4)}{\leqslant} 2H\theta + H \sum_{h\in[\![H]\!]} \sum_{g\in\mathcal{G}_{r_\theta^E, h}} \sum_{s\in\mathcal{S}} \mathbb{P}^{\pi_{r_\theta^E}}(G_h = g \wedge s_h = s)\left\| \pi_{r_\theta^E, h}(\cdot|s, g) - \widehat{\pi}_h(\cdot|s, g) \right\|_1$$

$$\overset{(5)}{=} 2H\theta + H \sum_{h\in[\![H]\!]} \sum_{g\in\mathcal{G}_{r_\theta^E,h}} \sum_{s\in\mathcal{S}} \mathbb{P}^{\pi^E}(G_h = g \wedge s_h = s)\left\|\pi_{r_\theta^E,h}(\cdot|s,g) - \widehat{\pi}_h(\cdot|s,g)\right\|_1$$

$$\overset{(6)}{\leqslant} 2H\theta + H\epsilon',$$

where at (1) we apply triangle's inequality, at (2) we apply Lemma 4.2 and Lemma C.1 twice, at (3) we use Particular Case 6.13 of Villani (2008), which tells us that we can upper bound the Wasserstein distance between two distributions supported on set $\mathcal{X}$ by the diameter of $\mathcal{X}$ ($\max_{x,x'\in\mathcal{X}} |x - x'|$) times the one norm between the two distributions. Since $\mathcal{X} = [0, H]$ in our case, we get the expression written above. At (4) we apply Lemma C.3 with the notation defined in that lemma, observing that policies $\pi_{r_\theta^E}$ and $\widehat{\pi}$ satisfy the hypothesis for reward $r_\theta^E$. At (5) we use Lemma C.2 observing that the random return $G_h$ is in terms of reward $r_\theta^E$, and recalling the definition of $\pi_{r_\theta^E}$. Lastly, at (6) we apply Lemma C.4 with accuracy $\epsilon'$.

The result follows by imposing that $\epsilon' \leqslant \frac{\epsilon}{2H}$ and $2H\theta \leqslant \frac{\epsilon}{2}$, which can be achieved by taking $\epsilon' = \frac{\epsilon}{2H}$ and $\theta = \frac{\epsilon}{4H}$, and by observing that:

$$\overline{\mathcal{G}} := \sum_{h\in[\![H]\!]} |\mathcal{G}_{r_\theta^E,h}|$$

$$\leqslant \sum_{h\in[\![H]\!]} |\mathcal{Y}_h^\theta|$$

$$= \sum_{h\in[\![H]\!]} |\{0, \theta, 2\theta, \ldots, \lfloor (h-1)/\theta \rfloor\theta\}|$$

$$\leqslant \sum_{h\in[\![H]\!]} (1 + (h-1)/\theta)$$

$$= H(1 - 1/\theta) + \frac{1}{\theta} \sum_{h\in[\![H]\!]} h$$

$$\overset{(10)}{=} H(1 - 1/\theta) + \frac{H(H+1)}{2\theta}$$

$$\leqslant \frac{H^2}{\theta}$$

$$\overset{(11)}{\leqslant} \frac{4H^3}{\epsilon},$$

where at (10) we used the formula for arithmetic sums, and at (11) we used the previous choice $\theta = \frac{\epsilon}{4H}$.

Replacing into the number of samples in Lemma C.4 (and also $\epsilon' = \frac{\epsilon}{2H}$) we get the result:

$$N \geqslant \frac{3072 S H^6 \ln\frac{8SH^3}{\delta\epsilon}}{\epsilon^3}\left(\ln\frac{8SH^3}{\delta\epsilon} + (A-1)\ln\left(\frac{2048eSH^6 \ln\frac{8SH^3}{\delta\epsilon}}{\epsilon^3}\right)\right).$$

By using $\widetilde{\mathcal{O}}$ notation to hide logarithmic terms in $S, A, H, \frac{1}{\epsilon}, \ln\frac{1}{\delta}$, we get the result. $\qquad\square$

**Lemma C.3** (Error Propagation). *Let $\mathcal{M}_r$ be any MDP. For any pair of policies $\pi, \pi' \in \Pi^{NM}$ such that, for all $h \in [\![H]\!]$, $a \in \mathcal{A}$, $s \in \mathcal{S}$ and $\omega, \omega' \in \Omega_h$ with $G(\omega; r) = G(\omega'; r)$:*

$$\pi(a|\omega, s) = \pi(a|\omega', s) \qquad \wedge \qquad \pi'(a|\omega, s) = \pi'(a|\omega', s),$$

*it holds that:*

$$\left\|\eta_r^\pi - \eta_r^{\pi'}\right\|_1 \leqslant \sum_{h\in[\![H]\!]} \sum_{g\in\mathcal{G}_{r,h}} \sum_{s\in\mathcal{S}} \mathbb{P}^\pi(G_h = g \wedge s_h = s)\left\|\pi_h(\cdot|s,g) - \pi'_h(\cdot|s,g)\right\|_1,$$

*where $\mathcal{G}_{r,h} := \{g \in [0, h-1] \,|\, \exists\omega \in \Omega_h : G(\omega; r) = g\}$ for all $h \in [\![H+1]\!]$, $G_h := \sum_{h'=1}^{h-1} r_{h'}(s_{h'}, a_{h'})$ denotes the* random *return at stage $h$, and $\pi_h(\cdot|s,g)$ and $\pi'_h(\cdot|s,g)$ denote the unique probability with which the policies $\pi$ and $\pi'$ prescribe actions in $s$ at $h$ under any trajectory $\omega \in \Omega_h$ with $G(\omega; r) = g$.*

*Proof.* To prove the result, we first demonstrate by induction that, for all $h \in [\![2, H]\!]$, it holds that:

$$\sum_{g \in \mathcal{G}_{r,h}} \sum_{s \in \mathcal{S}} \left| \mathbb{P}^\pi(G_h = g \wedge s_h = s) - \mathbb{P}^{\pi'}(G_h = g \wedge s_h = s) \right|$$

$$\leqslant \sum_{h' \in [\![h-1]\!]} \sum_{g \in \mathcal{G}_{r,h'}} \sum_{s \in \mathcal{S}} \mathbb{P}^\pi(G_{h'} = g \wedge s_{h'} = s) \left\| \pi_{h'}(\cdot|s, g) - \pi'_{h'}(\cdot|s, g) \right\|_1.$$

We remark that, by hypothesis, both $\pi$ and $\pi'$ prescribe the same actions when faced with trajectories $\omega, \omega'$ with the same return $G(\omega; r) = G(\omega'; r)$ that are followed by the same state $s$.

We begin with the base case $h = 2$. We can write:

$$\sum_{g \in \mathcal{G}_{r,2}} \sum_{s' \in \mathcal{S}} \left| \mathbb{P}^\pi(G_2 = g \wedge s_2 = s') - \mathbb{P}^{\pi'}(G_2 = g \wedge s_2 = s') \right|$$

$$= \sum_{g \in \mathcal{G}_{r,2}} \sum_{s' \in \mathcal{S}} \left| \mathbb{P}^\pi(r_1(s_1, a_1) = g \wedge s_2 = s') - \mathbb{P}^{\pi'}(r_1(s_1, a_1) = g \wedge s_2 = s') \right|$$

$$= \sum_{g \in \mathcal{G}_{r,2}} \sum_{s' \in \mathcal{S}} \left| \sum_{\substack{(s,a) \in \mathcal{S} \times \mathcal{A}: \\ r_1(s,a)=g}} \left( \mathbb{P}^\pi(s_2 = s'|s, a) \mathbb{P}^\pi(a_1 = a|s) \mathbb{P}^\pi(s_1 = s) \right. \right.$$

$$\left. \left. - \mathbb{P}^{\pi'}(s_2 = s'|s, a) \mathbb{P}^{\pi'}(a_1 = a|s) \mathbb{P}^{\pi'}(s_1 = s) \right) \right|$$

$$\stackrel{(1)}{=} \sum_{g \in \mathcal{G}_{r,2}} \sum_{s' \in \mathcal{S}} \left| \sum_{\substack{(s,a) \in \mathcal{S} \times \mathcal{A}: \\ r_1(s,a)=g}} \left( p(s'|s, a) \mathbb{1}\{s = s_0\} \pi(a|s) \right. \right.$$

$$\left. \left. - p(s'|s, a) \mathbb{1}\{s = s_0\} \pi'(a|s) \right) \right|$$

$$\stackrel{(2)}{\leqslant} \sum_{g \in \mathcal{G}_{r,2}} \sum_{\substack{a \in \mathcal{A}: \\ r_1(s_0,a)=g}} \sum_{s' \in \mathcal{S}} p(s'|s_0, a) \left| \pi(a|s_0) - \pi'(a|s_0) \right|$$

$$= \sum_{g \in \mathcal{G}_{r,2}} \sum_{\substack{a \in \mathcal{A}: \\ r_1(s_0,a)=g}} \left| \pi(a|s_0) - \pi'(a|s_0) \right|$$

$$= \sum_{a \in \mathcal{A}} \left| \pi(a|s_0) - \pi'(a|s_0) \right|$$

$$= \left\| \pi(\cdot|s_0) - \pi'(\cdot|s_0) \right\|_1$$

$$\stackrel{(3)}{=} \sum_{h' \in [\![1]\!]} \sum_{g \in \mathcal{G}_{r,h'}} \sum_{s \in \mathcal{S}} \mathbb{P}^\pi(G_{h'} = g \wedge s_{h'} = s) \left\| \pi_{h'}(\cdot|s, g) - \pi'_{h'}(\cdot|s, g) \right\|_1,$$

where at (1) we realize that in $\mathcal{M}_r$ the initial state is always $s_0$, and that the transition model is Markovian and independent of the policy, at (2) we apply triangle's inequality and keep only $s_0$ because of the indicator, and at (3) we have simply rewritten the expression in a more convenient way for proving the result (note that $[\![1]\!] = \{1\}$ and $\mathcal{G}_{r',1} = \{0\}$ and $\mathbb{P}^\pi(G_{h'} = g \wedge s_1 = s) = \mathbb{1}\{g = 0 \wedge s = s_0\}$).

Now, let us consider any stage $h \in [\![3, H]\!]$. Let us make the inductive hypothesis that:

$$\sum_{g \in \mathcal{G}_{r,h-1}} \sum_{s \in \mathcal{S}} \left| \mathbb{P}^\pi(G_{h-1} = g \wedge s_{h-1} = s) - \mathbb{P}^{\pi'}(G_{h-1} = g \wedge s_{h-1} = s) \right|$$

$$\leqslant \sum_{h' \in [\![h-2]\!]} \sum_{g \in \mathcal{G}_{r,h'}} \sum_{s \in \mathcal{S}} \mathbb{P}^\pi(G_{h'} = g \wedge s_{h'} = s) \cdot \left\| \pi_{h'}(\cdot|s, g) - \pi'_{h'}(\cdot|s, g) \right\|_1.$$

Then, we can write (we use symbol $\omega_h$ to denote the random trajectory $(s_1, a_1, \ldots, s_h, a_h)$ up to stage $h$):

$$\sum_{g' \in \mathcal{G}_{r,h}} \sum_{s' \in \mathcal{S}} \left| \mathbb{P}^\pi(G_h = g' \wedge s_h = s') - \mathbb{P}^{\pi'}(G_h = g' \wedge s_h = s') \right|$$

$$
\begin{aligned}
&= \sum_{g' \in \mathcal{G}_{r,h}} \sum_{s' \in \mathcal{S}} \Big| \sum_{g \in \mathcal{G}_{r,h-1}} \Big( \\
&\qquad \mathbb{P}^{\pi}\big(G_{h-1} = g \wedge r_{h-1}(s_{h-1}, a_{h-1}) = g' - g \wedge s_h = s'\big) \\
&\qquad - \mathbb{P}^{\pi'}\big(G_{h-1} = g \wedge r_{h-1}(s_{h-1}, a_{h-1}) = g' - g \wedge s_h = s'\big)\Big)\Big| \\[4pt]
&= \sum_{g' \in \mathcal{G}_{r,h}} \sum_{s' \in \mathcal{S}} \Big| \sum_{g \in \mathcal{G}_{r,h-1}} \sum_{\substack{\omega \in \Omega_{h-1}: \\ G(\omega;r)=g}} \sum_{\substack{(s,a) \in \mathcal{S} \times \mathcal{A}: \\ r_{h-1}(s,a)=g'-g}} \Big( \\
&\qquad \mathbb{P}^{\pi}\big(\omega_{h-2} = \omega \wedge s_{h-1} = s \wedge a_{h-1} = a \wedge s_h = s'\big) \\
&\qquad - \mathbb{P}^{\pi'}\big(\omega_{h-2} = \omega \wedge s_{h-1} = s \wedge a_{h-1} = a \wedge s_h = s'\big)\Big)\Big| \\[4pt]
&\overset{(4)}{=} \sum_{g' \in \mathcal{G}_{r,h}} \sum_{s' \in \mathcal{S}} \Big| \sum_{g \in \mathcal{G}_{r,h-1}} \sum_{\substack{\omega \in \Omega_{h-1}: \\ G(\omega;r)=g}} \sum_{\substack{(s,a) \in \mathcal{S} \times \mathcal{A}: \\ r_{h-1}(s,a)=g'-g}} \Big( \\
&\qquad \mathbb{P}^{\pi}\big(\omega_{h-2} = \omega \wedge s_{h-1} = s\big) \mathbb{P}^{\pi}\big(a_{h-1} = a \wedge s_h = s' | \omega, s\big) \\
&\qquad - \mathbb{P}^{\pi'}\big(\omega_{h-2} = \omega \wedge s_{h-1} = s\big) \mathbb{P}^{\pi'}\big(a_{h-1} = a \wedge s_h = s' | \omega, s\big)\Big)\Big| \\[4pt]
&\overset{(5)}{=} \sum_{g' \in \mathcal{G}_{r,h}} \sum_{s' \in \mathcal{S}} \Big| \sum_{g \in \mathcal{G}_{r,h-1}} \sum_{\substack{\omega \in \Omega_{h-1}: \\ G(\omega;r)=g}} \sum_{\substack{(s,a) \in \mathcal{S} \times \mathcal{A}: \\ r_{h-1}(s,a)=g'-g}} \Big( \\
&\qquad \mathbb{P}^{\pi}\big(\omega_{h-2} = \omega \wedge s_{h-1} = s\big) \pi(a|\omega, s) p_{h-1}(s'|s,a) \\
&\qquad - \mathbb{P}^{\pi'}\big(\omega_{h-2} = \omega \wedge s_{h-1} = s\big) \pi'(a|\omega, s) p_{h-1}(s'|s,a)\Big)\Big| \\[4pt]
&= \sum_{g' \in \mathcal{G}_{r,h}} \sum_{s' \in \mathcal{S}} \Big| \sum_{g \in \mathcal{G}_{r,h-1}} \sum_{\substack{\omega \in \Omega_{h-1}: \\ G(\omega;r)=g}} \sum_{\substack{(s,a) \in \mathcal{S} \times \mathcal{A}: \\ r_{h-1}(s,a)=g'-g}} p_{h-1}(s'|s,a) \Big( \\
&\qquad \mathbb{P}^{\pi}\big(\omega_{h-2} = \omega \wedge s_{h-1} = s\big) \pi(a|\omega, s) \\
&\qquad - \mathbb{P}^{\pi'}\big(\omega_{h-2} = \omega \wedge s_{h-1} = s\big) \pi'(a|\omega, s) \\
&\qquad \pm \mathbb{P}^{\pi}\big(\omega_{h-2} = \omega \wedge s_{h-1} = s\big) \pi'(a|\omega, s)\Big)\Big| \\[4pt]
&\overset{(6)}{\leqslant} \sum_{g' \in \mathcal{G}_{r,h}} \sum_{s' \in \mathcal{S}} \sum_{g \in \mathcal{G}_{r,h-1}} \sum_{\substack{\omega \in \Omega_{h-1}: \\ G(\omega;r)=g}} \sum_{\substack{(s,a) \in \mathcal{S} \times \mathcal{A}: \\ r_{h-1}(s,a)=g'-g}} p_{h-1}(s'|s,a) \\
&\qquad \cdot \mathbb{P}^{\pi}\big(\omega_{h-2} = \omega \wedge s_{h-1} = s\big) \Big| \pi(a|\omega, s) - \pi'(a|\omega, s)\Big| \\
&\qquad + \sum_{g' \in \mathcal{G}_{r,h}} \sum_{s' \in \mathcal{S}} \sum_{g \in \mathcal{G}_{r,h-1}} \Big| \sum_{\substack{\omega \in \Omega_{h-1}: \\ G(\omega;r)=g}} \sum_{\substack{(s,a) \in \mathcal{S} \times \mathcal{A}: \\ r_{h-1}(s,a)=g'-g}} p_{h-1}(s'|s,a) \pi'(a|\omega, s) \\
&\qquad \cdot \Big( \mathbb{P}^{\pi}\big(\omega_{h-2} = \omega \wedge s_{h-1} = s\big) - \mathbb{P}^{\pi'}\big(\omega_{h-2} = \omega \wedge s_{h-1} = s\big)\Big)\Big| \\[4pt]
&\overset{(7)}{\leqslant} \sum_{g' \in \mathcal{G}_{r,h}} \sum_{g \in \mathcal{G}_{r,h-1}} \sum_{\substack{\omega \in \Omega_{h-1}: \\ G(\omega;r)=g}} \sum_{\substack{(s,a) \in \mathcal{S} \times \mathcal{A}: \\ r_{h-1}(s,a)=g'-g}} \\
&\qquad \mathbb{P}^{\pi}\big(\omega_{h-2} = \omega \wedge s_{h-1} = s\big) \Big| \pi(a|\omega, s) - \pi'(a|\omega, s)\Big| \\
&\qquad + \sum_{g' \in \mathcal{G}_{r,h}} \sum_{s' \in \mathcal{S}} \sum_{g \in \mathcal{G}_{r,h-1}} \sum_{\substack{(s,a) \in \mathcal{S} \times \mathcal{A}: \\ r_{h-1}(s,a)=g'-g}} p_{h-1}(s'|s,a) \Big| \sum_{\substack{\omega \in \Omega_{h-1}: \\ G(\omega;r)=g}} \pi'(a|\omega, s) \\
&\qquad \cdot \Big( \mathbb{P}^{\pi}\big(\omega_{h-2} = \omega \wedge s_{h-1} = s\big) - \mathbb{P}^{\pi'}\big(\omega_{h-2} = \omega \wedge s_{h-1} = s\big)\Big)\Big|
\end{aligned}
$$

$$\stackrel{(8)}{=} \sum_{g\in\mathcal{G}_{r,h-1}} \sum_{(s,a)\in\mathcal{S}\times\mathcal{A}} \left|\pi_{h-1}(a|s,g) - \pi'_{h-1}(a|s,g)\right|$$
$$\cdot \sum_{\substack{\omega\in\Omega_{h-1}:\\ G(\omega;r)=g}} \mathbb{P}^\pi(\omega_{h-2} = \omega \wedge s_{h-1} = s)$$
$$+ \sum_{g'\in\mathcal{G}_{r,h}} \sum_{g\in\mathcal{G}_{r,h-1}} \sum_{\substack{(s,a)\in\mathcal{S}\times\mathcal{A}:\\ r_{h-1}(s,a)=g'-g}} \pi'_{h-1}(a|s,g) \left| \sum_{\substack{\omega\in\Omega_{h-1}:\\ G(\omega;r)=g}} \right.$$
$$\left. \cdot \left(\mathbb{P}^\pi(\omega_{h-2} = \omega \wedge s_{h-1} = s) - \mathbb{P}^{\pi'}(\omega_{h-2} = \omega \wedge s_{h-1} = s)\right)\right|$$
$$= \sum_{g\in\mathcal{G}_{r,h-1}} \sum_{(s,a)\in\mathcal{S}\times\mathcal{A}} \left|\pi_{h-1}(a|s,g) - \pi'_{h-1}(a|s,g)\right| \mathbb{P}^\pi(G_{h-1} = g \wedge s_{h-1} = s)$$
$$+ \sum_{g'\in\mathcal{G}_{r,h}} \sum_{g\in\mathcal{G}_{r,h-1}} \sum_{\substack{(s,a)\in\mathcal{S}\times\mathcal{A}:\\ r_{h-1}(s,a)=g'-g}} \pi'_{h-1}(a|s,g)$$
$$\left|\mathbb{P}^\pi(G_{h-1} = g \wedge s_{h-1} = s) - \mathbb{P}^{\pi'}(G_{h-1} = g \wedge s_{h-1} = s)\right|$$
$$\stackrel{(9)}{=} \sum_{g\in\mathcal{G}_{r,h-1}} \sum_{s\in\mathcal{S}} \mathbb{P}^\pi(G_{h-1} = g \wedge s_{h-1} = s)\left\|\pi_{h-1}(\cdot|s,g) - \pi'_{h-1}(\cdot|s,g)\right\|_1$$
$$+ \sum_{g\in\mathcal{G}_{r,h-1}} \sum_{s\in\mathcal{S}} \left|\mathbb{P}^\pi(G_{h-1} = g \wedge s_{h-1} = s) - \mathbb{P}^{\pi'}(G_{h-1} = g \wedge s_{h-1} = s)\right|$$
$$\stackrel{(10)}{\leq} \sum_{g\in\mathcal{G}_{r,h-1}} \sum_{s\in\mathcal{S}} \mathbb{P}^\pi(G_{h-1} = g \wedge s_{h-1} = s)\left\|\pi_{h-1}(\cdot|s,g) - \pi'_{h-1}(\cdot|s,g)\right\|_1$$
$$+ \sum_{h'\in[\![h-2]\!]} \sum_{g\in\mathcal{G}_{r,h'}} \sum_{s\in\mathcal{S}} \mathbb{P}^\pi(G_{h'} = g \wedge s_{h'} = s)$$
$$\cdot\left\|\pi_{h'}(\cdot|s,g) - \pi'_{h'}(\cdot|s,g)\right\|_1$$
$$= \sum_{h'\in[\![h-1]\!]} \sum_{g\in\mathcal{G}_{r,h'}} \sum_{s\in\mathcal{S}} \mathbb{P}^\pi(G_{h'} = g \wedge s_{h'} = s) \cdot \left\|\pi_{h'}(\cdot|s,g) - \pi'_{h'}(\cdot|s,g)\right\|_1,$$

where at (4) we use the chain rule of conditional probabilities, at (5) we do it again, and we recognize the policies $\pi$ and $\pi'$, and also that the transition model is Markovian, at (6) we use triangle's inequality to split the summations and bring the absolute value inside, at (7), in the first term, we note that $p_{h-1}(s'|s,a)$ is the only term that depends on $s'$ and that it sums to 1, while in the second term we exchange the order of two summations and apply triangle's inequality to bring one inside, at (8), in the first term, we first remove the summation on $g'$ along with the indicator function that forces us to consider a subset of state-action pairs, and then we exchange two other summations and note that the policies do not depend by hypothesis on the entire past trajectory, but just on the return so far. Instead, in the second term, we use that $\sum_{s'\in\mathcal{S}} p_{h-1}(s'|s,a) = 1$, and also that, by hypothesis, $\pi'$ does not depend on the entire past trajectory, but just on $g$. At (9), i.a., we use that $\sum_{g'\in\mathcal{G}_{r,h}} \mathbb{1}\{r_{h-1}(s,a) = g' - g\} = 1$ and that $\sum_{a\in\mathcal{A}} \pi'_{h-1}(a|s,g) = 1$. Finally, at (10), we apply the inductive hypothesis.

Thanks to this result, we can finally prove the claim in the lemma, using passages analogous to those above, with the difference that we do not have the summation over the states at the current stage (i.e., $H + 1$):

$$\left\|\eta_r^\pi - \eta_r^{\pi'}\right\|_1 = \sum_{g\in\mathcal{G}_{r,H+1}} \left|\eta_r^\pi(g) - \eta_r^{\pi'}(g)\right|$$
$$= \sum_{g\in\mathcal{G}_{r,H+1}} \left|\mathbb{P}^\pi(G_{H+1} = g) - \mathbb{P}^{\pi'}(G_{H+1} = g)\right|$$
$$= \sum_{g\in\mathcal{G}_{r,H+1}} \left|\sum_{s'\in\mathcal{S}} \sum_{a'\in\mathcal{A}} \sum_{g'\in\mathcal{G}_{r,H}} \sum_{\omega\in\Omega_H} \mathbb{1}\{r_H(s',a') = g - g', G(\omega;r) = g'\}\left(\right.\right.$$

$$\cdot \mathbb{P}^\pi(\omega_{H-1} = \omega \wedge s_H = s' \wedge a_H = a')$$
$$- \mathbb{P}^{\pi'}(\omega_{H-1} = \omega \wedge s_H = s' \wedge a_H = a')\Big|$$

$$= \sum_{g \in \mathcal{G}_{r,H+1}} \Big| \sum_{s' \in \mathcal{S}} \sum_{a' \in \mathcal{A}} \sum_{g' \in \mathcal{G}_{r,H}} \sum_{\omega \in \Omega_H} \mathbb{1}\{r_H(s',a') = g - g', G(\omega;r) = g'\}\Big($$
$$\cdot \mathbb{P}^\pi(\omega_{H-1} = \omega \wedge s_H = s')\pi(a'|\omega,s')$$
$$- \mathbb{P}^{\pi'}(\omega_{H-1} = \omega \wedge s_H = s')\pi'(a'|\omega,s')$$
$$\pm \mathbb{P}^\pi(\omega_{H-1} = \omega \wedge s_H = s')\pi'(a'|\omega,s')\Big)\Big|$$

$$\overset{(11)}{\leqslant} \sum_{g \in \mathcal{G}_{r,H}} \sum_{s \in \mathcal{S}} \mathbb{P}^\pi(G_H = g \wedge s_H = s)\Big\|\pi_H(\cdot|s,g) - \pi'_H(\cdot|s,g)\Big\|_1$$
$$+ \sum_{g \in \mathcal{G}_{r,H}} \sum_{s \in \mathcal{S}} \Big|\mathbb{P}^\pi(G_H = g \wedge s_H = s) - \mathbb{P}^{\pi'}(G_H = g \wedge s_H = s)\Big|$$

$$\overset{(12)}{\leqslant} \sum_{h' \in \llbracket H \rrbracket} \sum_{g \in \mathcal{G}_{r,h'}} \sum_{s \in \mathcal{S}} \mathbb{P}^\pi(G_{h'} = g \wedge s_{h'} = s)\Big\|\pi_{h'}(\cdot|s,g) - \pi'_{h'}(\cdot|s,g)\Big\|_1,$$

where at (11) we made the same passages as above, all in one, and at (12) we applied the result proved earlier by induction.

This concludes the proof. Note that, from a high-level perspective, our proof approach resembles the "reduction to supervised learning" approach made in Rajaraman et al. (2020) for stochastic policies, with the difference that we work in an augmented state-space MDP. $\qquad\square$

**Lemma C.4** (Concentration). *Let $\epsilon \in (0, H]$ and $\delta \in (0,1)$. Let $\mathcal{M}_{r^E}$ be any MDP, $\pi^E \in \Pi^{NM}$ be any expert's policy, and $\widehat{\pi}$ be the output of Algorithm 1. Then, with probability $1 - \delta$, we have that (we use the notation in Lemma C.3):*

$$\sum_{h \in \llbracket H \rrbracket} \sum_{g \in \mathcal{G}_{r^E_\theta,h}} \sum_{s \in \mathcal{S}} \mathbb{P}^{\pi^E}(G_h = g \wedge s_h = s)\Big\|\pi_{r^E_\theta,h}(\cdot|s,g) - \widehat{\pi}_h(\cdot|s,g)\Big\|_1 \leqslant \epsilon,$$

*with a number of samples:*

$$N \leqslant \frac{193 S H \overline{\mathcal{G}} \ln \frac{2S\overline{\mathcal{G}}}{\delta}}{\epsilon^2}\left(\ln \frac{2S\overline{\mathcal{G}}}{\delta} + (A-1)\ln\Big(\frac{128 e S H \overline{\mathcal{G}} \ln \frac{2S\overline{\mathcal{G}}}{\delta}}{\epsilon^2}\Big)\right),$$

*where $\overline{\mathcal{G}} := \sum_{h \in \llbracket H \rrbracket} |\mathcal{G}_{r^E_\theta,h}|$.*

*Proof.* We can write:

$$\sum_{h \in \llbracket H \rrbracket} \sum_{g \in \mathcal{G}_{r^E_\theta,h}} \sum_{s \in \mathcal{S}} \mathbb{P}^{\pi^E}(G_h = g \wedge s_h = s)\Big\|\pi_{r^E_\theta,h}(\cdot|s,g) - \widehat{\pi}_h(\cdot|s,g)\Big\|_1$$

$$= \sum_{h \in \llbracket H \rrbracket} \sum_{g \in \mathcal{G}_{r^E_\theta,h}} \sum_{s \in \mathcal{S}} \mathbb{P}^{\pi^E}(G_h = g \wedge s_h = s) \sum_{a \in \mathcal{A}} \Big|\pi_{r^E_\theta,h}(a|g,s)$$
$$- \Big(\frac{M_h(s,g,a)}{\sum_{a'} M_h(s,g,a')}\mathbb{1}\Big\{\sum_{a'} M_h(s,g,a') > 0\Big\} + \frac{1}{A}\mathbb{1}\Big\{\sum_{a'} M_h(s,g,a') = 0\Big\}\Big)\Big|$$

$$\overset{(1)}{\leqslant} \sum_{h \in \llbracket H \rrbracket} \sum_{g \in \mathcal{G}_{r^E_\theta,h}} \sum_{s \in \mathcal{S}} \mathbb{P}^{\pi^E}(G_h = g \wedge s_h = s)$$
$$\cdot 2\sqrt{2}\sqrt{\frac{\ln \frac{2S\overline{\mathcal{G}}}{\delta} + (A-1)\ln\big(e\big(1 + \frac{\sum_{a'} M_h(s,g,a')}{A-1}\big)\big)}{\sum_{a'} M_h(s,g,a')}}$$

$$\overset{(2)}{\leqslant} 2\sqrt{2}\sqrt{\ln \frac{2S\overline{\mathcal{G}}}{\delta} + (A-1)\ln\Big(e\Big(1 + \frac{N}{A-1}\Big)\Big)}$$

$$\cdot \sum_{h\in[\![H]\!]} \sum_{g\in\mathcal{G}_{r_\theta^E,h}} \sum_{s\in\mathcal{S}} \mathbb{P}^{\pi^E}(G_h = g \wedge s_h = s)\sqrt{\frac{1}{\sum_{a'} M_h(s,g,a')}}$$

$$\overset{(3)}{\leqslant} 2\sqrt{2}\sqrt{\ln\frac{2S\overline{\mathcal{G}}}{\delta} + (A-1)\ln\Big(e\Big(1+\frac{N}{A-1}\Big)\Big)}$$

$$\cdot \sum_{h\in[\![H]\!]} \sum_{g\in\mathcal{G}_{r_\theta^E,h}} \sum_{s\in\mathcal{S}} \sqrt{\frac{8\ln\frac{2S\overline{\mathcal{G}}}{\delta}\mathbb{P}^{\pi^E}(G_h = g \wedge s_h = s)}{N}}$$

$$= 8\sqrt{\frac{\ln\frac{2S\overline{\mathcal{G}}}{\delta}}{N}}\sqrt{\ln\frac{2S\overline{\mathcal{G}}}{\delta} + (A-1)\ln\Big(e\Big(1+\frac{N}{A-1}\Big)\Big)}$$

$$\cdot \sum_{h\in[\![H]\!]} \sum_{g\in\mathcal{G}_{r_\theta^E,h}} \sum_{s\in\mathcal{S}} \sqrt{\mathbb{P}^{\pi^E}(G_h = g \wedge s_h = s)}$$

$$\overset{(4)}{\leqslant} 8\sqrt{\frac{\ln\frac{2S\overline{\mathcal{G}}}{\delta}}{N}}\sqrt{\ln\frac{2S\overline{\mathcal{G}}}{\delta} + (A-1)\ln\Big(e\Big(1+\frac{N}{A-1}\Big)\Big)}$$

$$\cdot \sum_{h\in[\![H]\!]} \sqrt{S|\mathcal{G}_{r_\theta^E,h}|}\sqrt{\sum_{g\in\mathcal{G}_{r_\theta^E,h}} \sum_{s\in\mathcal{S}} \mathbb{P}^{\pi^E}(G_h = g \wedge s_h = s)}$$

$$= 8\sqrt{\frac{S\ln\frac{2S\overline{\mathcal{G}}}{\delta}}{N}}\sqrt{\ln\frac{2S\overline{\mathcal{G}}}{\delta} + (A-1)\ln\Big(e\Big(1+\frac{N}{A-1}\Big)\Big)} \sum_{h\in[\![H]\!]} \sqrt{|\mathcal{G}_{r_\theta^E,h}|}$$

$$\overset{(5)}{\leqslant} 8\sqrt{\frac{SH\overline{\mathcal{G}}\ln\frac{2S\overline{\mathcal{G}}}{\delta}}{N}}\sqrt{\ln\frac{2S\overline{\mathcal{G}}}{\delta} + (A-1)\ln\Big(e\Big(1+\frac{N}{A-1}\Big)\Big)},$$

where at (1) we use that, if $\sum_{a'} M_h(s,g,a') = 0$, then:

$$\sum_{a\in\mathcal{A}} \Big|\pi_{r_\theta^E,h}(a|g,s) - \Big(\frac{M_h(s,g,a)}{\sum_{a'} M_h(s,g,a')}\mathbb{1}\{\sum_{a'} M_h(s,g,a') > 0\} + \frac{1}{A}\mathbb{1}\{\sum_{a'} M_h(s,g,a') = 0\}\Big)\Big|$$

$$= \sum_{a\in\mathcal{A}} \Big|\pi_{r_\theta^E,h}(a|g,s) - \frac{1}{A}\Big|$$

$$\leqslant 2,$$

as we the total variation distance between two probability distributions cannot exceed 1. Instead, if $\sum_{a'} M_h(s,g,a') > 0$, *conditioning* on $\sum_{a'} M_h(s,g,a')$, at all $s,g$ where $\mathbb{P}^{\pi^E}(G_h = g \wedge s_h = s) > 0$, we note that $M_h(s,g,a)/\sum_{a'} M_h(s,g,a')$ is the empirical vector of probabilities of $\pi_{r_\theta^E,h}(a|g,s)$ (recall its definition from Eq. 6), thus we can apply Lemma 8 of Kaufmann et al. (2021) to get that, for any $\delta \in (0,1)$:

$$\mathbb{P}^{\pi^E}\Big(KL\Big(\frac{M_h(s,g,\cdot)}{\sum_{a'} M_h(s,g,a')}\Big\|\pi_{r_\theta^E,h}(\cdot|g,s)\Big)$$

$$\leqslant \frac{\ln\frac{1}{\delta} + (A-1)\ln\big(e\big(1+\frac{\sum_{a'} M_h(s,g,a')}{A-1}\big)\big)}{\sum_{a'} M_h(s,g,a')}\Big) \geqslant 1-\delta.$$

Combining this result with the Pinsker's inequality, that tells us that $\|x - y\|_1 \leqslant \sqrt{2KL(x\|y)}$, and with a union bound over all $h \in [\![H]\!]$, $s \in \mathcal{S}$, $g \in \mathcal{G}_{r_\theta^E,h}$, we get the passage in (1) w.p. $1 - \delta/2$. Note that we add an additional 2 for the case $\sum_{a'} M_h(s,g,a') = 0$, and we define $\overline{\mathcal{G}} := \sum_{h\in[\![H]\!]} |\mathcal{G}_{r_\theta^E,h}|$. At (2) we bound $\sum_{a'} M_h(s,g,a') \leqslant N$, and bring that quantity outside, at (3) we apply Lemma A.1 of Xie et al. (2021), after having noticed that $\sum_{a'} M_h(s,g,a') \sim \text{Bin}\Big(N, \mathbb{P}^{\pi^E}(G_h = g \wedge s_h = s)\Big)$, and make it hold for all $s,g,h$ w.p. $1 - \delta/2$. At (4) and (5) we apply the Cauchy-Schwarz's inequality.

Now, we impose that this quantity is smaller than $\epsilon$:

$$8\sqrt{\frac{SH\overline{\mathcal{G}}\ln\frac{2S\overline{\mathcal{G}}}{\delta}}{N}}\sqrt{\ln\frac{2S\overline{\mathcal{G}}}{\delta}+(A-1)\ln\Big(e\Big(1+\frac{N}{A-1}\Big)\Big)}\leqslant\epsilon$$

$$\iff N\geqslant\frac{64SH\overline{\mathcal{G}}\ln^2\frac{2S\overline{\mathcal{G}}}{\delta}}{\epsilon^2}+\frac{64SH\overline{\mathcal{G}}(A-1)\ln\frac{2S\overline{\mathcal{G}}}{\delta}}{\epsilon^2}\ln\Big(\frac{eN}{A-1}+e\Big).$$

Thanks to Lemma J.3 of Lazzati et al. (2024), we know that this inequality is satisfied with:

$$N\leqslant\frac{128SH\overline{\mathcal{G}}\ln^2\frac{2S\overline{\mathcal{G}}}{\delta}}{\epsilon^2}+\frac{192SH\overline{\mathcal{G}}(A-1)\ln\frac{2S\overline{\mathcal{G}}}{\delta}}{\epsilon^2}\ln\Big(\frac{128eSH\overline{\mathcal{G}}\ln\frac{2S\overline{\mathcal{G}}}{\delta}}{\epsilon^2}\Big)+A-1.$$

Rearranging and applying a final union bound concludes the proof. $\qquad\square$

### C.2.2 ADDITIONAL DISCUSSION ON RS-BC

First, we observe that the sample complexity bound in Theorem 4.3 cannot be improved if we extend our analysis with that of Foster et al. (2024). Indeed, Foster et al. (2024) also provides an $1/\epsilon^2$ dependence as our proof, that combined with the additional $1/\epsilon$ due to discretization, would give the same $1/\epsilon^3$ rate. Moreover, Foster et al. (2024) would not allow to improve even the $H^6$ dependence in the horizon in our proof, as their Corollary 3.1 combined with a simple variation of our proof that considers an MDP with an augmented state space, would still provide an $H^4$ dependence that should be combined with the $H^2$ arising from bounding the Wasserstein with $H$ times the total variation, and taking the square. Note that these considerations on Foster et al. (2024) implicitly assumed that the proof of Foster et al. (2024) can be extended to non-Markovian expert's policies with the same rate, which has to be demonstrated as well.

Second, we mention that Rajaraman et al. (2020) provides for IL a $1/\epsilon$ dependence instead of $1/\epsilon^2$. However, note we remark that the result is in *expectation*, and not with *high probability*, as remarked also by Foster et al. (2024) in their footnote 21. Indeed, this explains why the $1/\epsilon$ rate of Rajaraman et al. (2020) seems to overcome the $1/\epsilon^2$ in the lower bound of Theorem G.1 of Foster et al. (2024).

Lastly, we mention that **RS-BC** (and also the theoretical guarantees in Theorem 4.3) can be easily extended to the setting in which $r^E$ is unknown but *observed*, namely, in which expert's trajectories are state-action-reward trajectories $(s_1,a_1,r_1,\dots)$. Indeed, looking at Algorithm 1, note that the computation of the returns $G(\omega;r^E)$ does not require knowledge of $r^E$ in state-action pairs never observed. This is different from **RS-KT**, in which we require knowledge of $r^E$ everywhere.

### C.3 KNOWN-TRANSITION SETTING

In Appendix C.3.1, we prove Theorem 4.4, while in Appendix C.3.2, we write down explicitly the LP in Eq. (10).

### C.3.1 PROOF OF THEOREM 4.4

**Theorem 4.4.** *Let $\epsilon\in(0,H]$ and $\delta\in(0,1)$. Let $\mathcal{M}_{r^E}$ be any MDP and $\pi^E\in\Pi^{NM}$ any policy. Assume that the optimization problem in Line 2 is solved exactly. Then, choosing $\theta=\epsilon/(7H)$, with probability $1-\delta$, the policy $\widehat{\pi}$ output by Algorithm 2 satisfies $\mathcal{W}(\eta_{r^E}^{\pi^E},\eta_{r^E}^{\widehat{\pi}})\leqslant\epsilon$, with:*

$$N\leqslant\mathcal{O}\Big(\frac{H^2}{\epsilon^2}\ln\frac{1}{\delta}\Big). \tag{9}$$

*Proof.* We begin by showing that the estimate of return distribution $\widehat{\eta}$ computed by **RS-KT** at Line 1 is close to the expert's return distribution with high probability. To this aim, we write:

$$\mathcal{W}\Big(\eta_{r^E}^{\pi^E},\widehat{\eta}\Big)\overset{(1)}{\leqslant}\mathcal{W}\Big(\eta_{r^E}^{\pi^E},\eta_{r^E_\theta}^{\pi^E}\Big)+\mathcal{W}\Big(\eta_{r^E_\theta}^{\pi^E},\widehat{\eta}\Big)$$

$$\overset{(2)}{\leqslant}H\theta/2+\mathcal{W}\Big(\eta_{r^E_\theta}^{\pi^E},\widehat{\eta}\Big)$$

$$\stackrel{(3)}{=} H\theta/2 + \int_0^H \left| F_{\eta_{r_\theta^E}^{\pi^E}}(x) - F_{\widehat{\eta}}(x) \right| dx$$

$$\leqslant H\theta/2 + \int_0^H \sup_{x' \in [0,H]} \left| F_{\eta_{r_\theta^E}^{\pi^E}}(x') - F_{\widehat{\eta}}(x') \right| dx$$

$$= H\theta/2 + \sup_{x' \in [0,H]} \left| F_{\eta_{r_\theta^E}^{\pi^E}}(x') - F_{\widehat{\eta}}(x') \right| \int_0^H dx$$

$$= H\theta/2 + H \sup_{x' \in [0,H]} \left| F_{\eta_{r_\theta^E}^{\pi^E}}(x') - F_{\widehat{\eta}}(x') \right|$$

$$\stackrel{(4)}{\leqslant} H\theta/2 + H\epsilon',$$

where at (1) we use triangle's inequality, at (2) we apply Lemma C.1, at (3) we use symbol $F_q$ for the distribution function of any probability measure $q$, at (4) we apply the DKW inequality (Kiefer & Wolfowitz, 1959; Massart, 1990), as $F_{\widehat{\eta}}(x) = \sum_{x' \leqslant x} \widehat{\eta}(x') = \sum_{x' \leqslant x} \frac{1}{N} \sum_{i \in [\![N]\!]} \mathbb{1}\{\sum_{h=1}^H r_{\theta,h}^E(s_h^i, a_h^i) = x'\} = \frac{1}{N} \sum_{i \in [\![N]\!]} \mathbb{1}\{\sum_{h=1}^H r_{\theta,h}^E(s_h^i, a_h^i) \leqslant x'\}$ corresponds to the empirical distribution function of $\eta_{r_\theta^E}^{\pi^E}$. Specifically, in our setting, the DKW inequality tells us that, for any $\epsilon' > 0$, it holds that:

$$\mathbb{P}^{\pi^E}\left( \sup_{x' \in [0,H]} \left| F_{\eta_{r_\theta^E}^{\pi^E}}(x') - F_{\widehat{\eta}}(x') \right| \leqslant \epsilon' \right) \geqslant 1 - 2e^{-2N(\epsilon')^2}. \tag{13}$$

By imposing the term on the right hand side to be $1 - \delta$, and solving w.r.t. $N$, we get that:

$$N \leqslant \frac{1}{2(\epsilon')^2} \ln \frac{2}{\delta}.$$

Now, building on this result, we can write:

$$\mathcal{W}\left( \eta_{r^E}^{\pi^E}, \eta_{r^E}^{\widehat{\pi}} \right) \stackrel{(5)}{\leqslant} \mathcal{W}\left( \eta_{r^E}^{\pi^E}, \widehat{\eta} \right) + \mathcal{W}\left( \widehat{\eta}, \eta_{r_\theta^E}^{\widehat{\pi}} \right) + \mathcal{W}\left( \eta_{r_\theta^E}^{\widehat{\pi}}, \eta_{r^E}^{\widehat{\pi}} \right)$$

$$\stackrel{(6)}{\leqslant} H\theta/2 + H\epsilon' + \mathcal{W}\left( \widehat{\eta}, \eta_{r_\theta^E}^{\widehat{\pi}} \right) + \mathcal{W}\left( \eta_{r_\theta^E}^{\widehat{\pi}}, \eta_{r^E}^{\widehat{\pi}} \right)$$

$$\stackrel{(7)}{\leqslant} H\theta + H\epsilon' + \mathcal{W}\left( \widehat{\eta}, \eta_{r_\theta^E}^{\widehat{\pi}} \right)$$

$$\stackrel{(8)}{=} H\theta + H\epsilon' + \min_{\pi \in \Pi(r_\theta^E)} \mathcal{W}\left( \widehat{\eta}, \eta_{r_\theta^E}^{\pi} \right)$$

$$\stackrel{(9)}{\leqslant} H\theta + H\epsilon' + \mathcal{W}\left( \widehat{\eta}, \eta_{r_\theta^E}^{\pi_{r_\theta^E}} \right)$$

$$\stackrel{(10)}{\leqslant} H\theta + H\epsilon' + \mathcal{W}\left( \widehat{\eta}, \eta_{r^E}^{\pi^E} \right) + \mathcal{W}\left( \eta_{r^E}^{\pi^E}, \eta_{r^E}^{\pi_{r_\theta^E}} \right) + \mathcal{W}\left( \eta_{r^E}^{\pi_{r_\theta^E}}, \eta_{r_\theta^E}^{\pi_{r_\theta^E}} \right)$$

$$\stackrel{(11)}{\leqslant} 2H\theta + 2H\epsilon' + \mathcal{W}\left( \eta_{r^E}^{\pi^E}, \eta_{r^E}^{\pi_{r_\theta^E}} \right) + \mathcal{W}\left( \eta_{r^E}^{\pi_{r_\theta^E}}, \eta_{r_\theta^E}^{\pi_{r_\theta^E}} \right)$$

$$\stackrel{(12)}{\leqslant} \frac{5}{2}H\theta + 2H\epsilon' + \mathcal{W}\left( \eta_{r^E}^{\pi^E}, \eta_{r^E}^{\pi_{r_\theta^E}} \right)$$

$$\stackrel{(12)}{\leqslant} \frac{7}{2}H\theta + 2H\epsilon',$$

where at (5) we apply triangle's inequality, at (6) we use the result above, at (7) we apply Lemma C.1, at (8) we use the definition of $\widehat{\pi}$ and the hypothesis of solving the minimization problem exactly, at (9) we upper bound with a specific choice of policy, i.e., $\pi_{r_\theta^E}$ (recall Eq. 6), at (10) we apply triangle's inequality again, at (11) we apply the result above again, at (12) we use again Lemma C.1, and finally, at (13), we apply Lemma 4.2.

If we now choose $\theta = \epsilon/(7H)$, and $\epsilon' = \epsilon/(4H)$, we get that, with probability $1 - \delta$, it holds that:

$$\mathcal{W}\left( \eta_{r^E}^{\pi^E}, \eta_{r^E}^{\widehat{\pi}} \right) \leqslant \epsilon,$$

with:

$$N \leqslant \frac{2H^2}{\epsilon^2} \ln \frac{2}{\delta}.$$

$\square$

### C.3.2 Explicit Formulation of the LP

The optimization problem in Eq. (10) can be written more explicitly as follows:

$$\min_{d \in \mathbb{R}_{\geqslant 0}^{SAH|\mathcal{Y}^\theta|}, \eta \in \mathbb{R}_{\geqslant 0}^{|\mathcal{Y}^\theta|}, t \in \mathbb{R}_{\geqslant 0}^{|\mathcal{Y}^\theta|}, x \in \mathbb{R}_{\geqslant 0}^{|\mathcal{Y}^\theta|}} \sum_{g \in \mathcal{Y}^\theta} t(g)$$

$$\text{s.t.} \quad \sum_{(s,a,g) \in \mathcal{S} \times \mathcal{A} \times \mathcal{Y}^\theta} d_1(s, g, a) = 1 \tag{14}$$

$$\sum_{a \in \mathcal{A}} d_1(s_0, 0, a) = 1 \tag{15}$$

$$\sum_{a \in \mathcal{A}} d_h(s, g, a) = \sum_{s', g', a'} d_{h-1}(s', g', a') p_{h-1}(s|s', a') \mathbb{1}\{r_{\theta,h}^E(s', a') = g - g'\}$$

$$\forall (s, g, h) \in \mathcal{S} \times \mathcal{Y}^\theta \times \{2, \ldots, H\} \tag{16}$$

$$\eta(g) = \sum_{(s,a) \in \mathcal{S} \times \mathcal{A}} d_H(s, g - r_{\theta,H}^E(s, a), a) \qquad \forall g \in \mathcal{Y}^\theta \tag{17}$$

$$x(g) = \sum_{g' \in \mathcal{Y}^\theta: g' \leqslant g} \left( \eta(g') - \widehat{\eta}(g') \right) \qquad \forall g \in \mathcal{Y}^\theta \tag{18}$$

$$-t(g) \leqslant x(g) \leqslant t(g) \qquad \forall g \in \mathcal{Y}^\theta \tag{19}$$

where Eqs. (14)-(15)-(16) denote the flow constraints, i.e., define the set of feasible occupancy measures $\mathcal{K}$, Eq. (17) enforces that $\eta$ is the return distribution in $\overline{\mathcal{M}}$ corresponding to occupancy measure $d$, while Eqs. (18)-(19) permit to rewrite the Wasserstein distance in a linear manner.

Observe that the number of optimization variables is $SAH|\mathcal{Y}^\theta| + |\mathcal{Y}^\theta| = \mathcal{O}(SAH|\mathcal{Y}^\theta|)$, while the number of constraints is $2 + (H-1)S|\mathcal{Y}^\theta| + 3|\mathcal{Y}^\theta| = \mathcal{O}(SAH|\mathcal{Y}^\theta|)$.

### C.4 When $r^E$ belongs to a finite set of $d$ Rewards

In this appendix, we consider a variant of the known-reward setting, in which $r^E$ is unknown, but we have knowledge of a set $\mathcal{R} = \{r^1, \ldots, r^d\}$ containing $d \geqslant 1$ reward functions, and we also know that $r^E \in \mathcal{R}$. We consider the following robust variant of RDM for this setting:

$$\widehat{\pi} \in \arg\min_{\pi \in \Pi^{\text{NM}}} \max_{r \in \mathcal{R}} \mathcal{W}\left( \eta_r^\pi, \eta_r^{\pi^E} \right). \tag{20}$$

To tackle this problem, we will proceed to Section 4, by considering the *cartesian product* of all the rewards in set $\mathcal{R}$. Specifically, we first present a class of non-Markovian policies sufficiently expressive for addressing this task, and then we present two variants of **RS-BC** and **RS-KT**. Crucially, we will have exponential dependencies in the number of rewards $d$ for both the computational and sample complexities.

Let us begin by presenting $\Pi(\mathcal{R})$, a generalization of $\Pi(r)$ to multiple rewards:

$$\Pi(\mathcal{R}) := \left\{ \pi \in \Pi^{\text{NM}} \,\middle|\, \exists \phi \in \Delta_{[\![H]\!] \times \mathcal{S} \times \mathcal{G}_{r1} \times \ldots \times \mathcal{G}_{rd}}^{\mathcal{A}} : \right.$$

$$\left. \pi(a|s, \omega) = \phi_h(a|s, G(\omega; r^1), \ldots, G(\omega; r^d)) \, \forall s \in \mathcal{S}, a \in \mathcal{A}, h \in [\![H]\!], \omega \in \Omega_h \right\}.$$

Intuitively, $\Pi(\mathcal{R})$ contains policies that depend on the amount of rewards collected so far for every possible reward $r^i$ in $\mathcal{R}$.

Now, let us define $\pi_{\mathcal{R}} \in \Pi(\mathcal{R})$, an analogous of policy $\pi_r$ (Eq. 6). For any set of rewards $\mathcal{R}$ and expert policy $\pi^E \in \Pi^{\text{NM}}$, define $\pi_{\mathcal{R}} \in \Pi(\mathcal{R})$ as the policy whose probability of taking an action $a$ in

---

**Algorithm 3:** Variant of **RS-BC** for a set of rewards

---

**Input :** Dataset $\mathcal{D}^E = \{(s_1^i, a_1^i, \ldots, s_H^i, a_H^i)\}_{i \in [\![N]\!]}$, set of rewards $\mathcal{R} = \{r^1, \ldots, r^d\}$, parameter $\theta$

```
// Count the state-action-cumulative reward occurrences
```

1 $M_h(s, g^1, \ldots, g^d, a) \leftarrow \sum_{i \in [\![N]\!]} \mathbb{1}\{s_h^i = s, a_h^i = a, \sum_{h'=1}^{h-1} r_{\theta,h'}^1(s_{h'}^i, a_{h'}^i) =$

$g^1, \ldots, \sum_{h'=1}^{h-1} r_{\theta,h'}^d(s_{h'}^i, a_{h'}^i) = g^d\}$    $\forall h \in [\![H]\!], s \in \mathcal{S}, g^1 \in \mathcal{Y}_h^\theta, \ldots, g^d \in \mathcal{Y}_h^\theta, a \in \mathcal{A}$

```
// Retrieve the policy
```

2 $\widehat{\pi}(a|s, \omega) \leftarrow$

$\begin{cases} \frac{M_h(s, G(\omega; r_\theta^1), \ldots, G(\omega; r_\theta^d), a)}{\sum_{a'} M_h(s, G(\omega; r_\theta^1), \ldots, G(\omega; r_\theta^d), a')} & \text{if denominator} > 0 \\ \frac{1}{A} & \text{otherwise} \end{cases}$    $\forall h \in [\![H]\!], s \in \mathcal{S}, \omega \in \Omega_h, a \in \mathcal{A}$

3 **Return** $\widehat{\pi}$

---

state $s$ with history $\omega \in \Omega_h$ coincides with the "average" probability with which $\pi^E$ selects $a$ in $s$ after accumulating $G(\omega; r^i)$ reward for each reward $r^i \in \mathcal{R}$:

$$\pi_{\mathcal{R}}(a|s, \omega) := \frac{\mathbb{P}^{\pi^E}(s_h = s, \ a_h = a, \ G_h^1 = G(\omega; r^1), \ \ldots, \ G_h^d = G(\omega; r^d))}{\mathbb{P}^{\pi^E}(s_h = s, \ G_h^1 = G(\omega; r^1), \ \ldots, \ G_h^d = G(\omega; r^d))}, \tag{21}$$

where we defined the random return at stage $h$ under reward $r^i \in \mathcal{R}$ as $G_h^i := \sum_{h'=1}^{h-1} r_{h'}^i(s_{h'}, a_{h'})$. If the denominator is zero, we set $\pi_r(a|s, \omega) = 1/A$.

We have the following result replicating Lemma 4.1:

**Lemma C.5.** *Let $\mathcal{M}$ be any MDP\R, $\mathcal{R} = \{r^1, \ldots, r^d\}$ any set of $d \geqslant 1$ rewards containing the unknown $r^E$, and $\pi^E \in \Pi^{NM}$ be any policy. Then, the policy $\pi_{\mathcal{R}} \in \Pi(\mathcal{R})$ is a minimizer of Eq. (20), and satisfies $\eta_{r^i}^{\pi_{\mathcal{R}}}(g) = \eta_{r^i}^{\pi^E}(g)$ for all g.*

We prove it in Appendix C.4.1. However, $\Pi(\mathcal{R}) = \Pi^{NM}$ of course for some rewards. Thus, we can discretize, by defining: $\mathcal{R}^\theta := \{r_\theta^1, \ldots, r_\theta^d\}$, i.e., by discretizing each reward inside $\mathcal{R}$. Then, it should be clear that the memory required for storing a policy in $\Pi(\mathcal{R}^\theta)$ scales as $\mathcal{O}(SAH|\mathcal{Y}^\theta|^d)$ (because we do the cartesian product). Analogously to Lemma 4.2, we can bound the approximation error (proof in Appendix C.4.1):

**Lemma C.6.** *Let $\theta \in (0, 1]$. Let $\mathcal{M}$ be any MDP\R, $\mathcal{R} = \{r^1, \ldots, r^d\}$ any set of $d \geqslant 1$ rewards containing the unknown $r^E$, and $\pi^E \in \Pi^{NM}$ be any policy. Then, the policy $\pi_{\mathcal{R}^\theta} \in \Pi(\mathcal{R}^\theta)$ satisfies $\mathcal{W}(\eta_{r^E}^{\pi_{\mathcal{R}^\theta}}, \eta_{r^E}^{\pi^E}) \leqslant H\theta$.*

Now, we address this problem with a variant of **RS-BC** for the no-interaction setting, and a variant of **RS-KT** for the known-transition setting.

We begin with a variant of **RS-BC**, reported in Algorithm 3. Simply, we count the (discretized) occurrences for every reward in $\mathcal{R}$. We have the following result (proof in Appendix C.4.2):

**Theorem C.7.** *Let $\epsilon \in (0, H]$ and $\delta \in (0, 1)$. Let $\mathcal{M}$ be any MDP\R, $\mathcal{R} = \{r^1, \ldots, r^d\}$ any set of $d \geqslant 1$ rewards containing the unknown $r^E$, and $\pi^E \in \Pi^{NM}$ be any policy. Then, choosing $\theta = \epsilon/(4H)$, with probability at least $1 - \delta$, the policy $\widehat{\pi}$ output by Algorithm 3 satisfies $\mathcal{W}(\eta_{r^E}^{\pi^E}, \eta_{r^E}^{\widehat{\pi}}) \leqslant \epsilon$, with a number of samples:*

$$N \leqslant \widetilde{\mathcal{O}}\left(\frac{SH^{4+2d}d^2 \ln \frac{1}{\delta}}{\epsilon^{2+d}}\left(A + \ln \frac{1}{\delta}\right)\right). \tag{22}$$

Observe that, for $d = 1$, we retrieve the known-reward setting, and the number of samples in Eq. (22) matches that of **RS-BC** (Eq. 8).

Now, we do the same for **RS-KT**. See Algorithm 4 for a variant of the algorithm. We have (proof in Appendix C.4.3):

**Theorem C.8.** *Let $\epsilon \in (0, H]$ and $\delta \in (0, 1)$. Let $\mathcal{M}$ be any MDP\R, $\mathcal{R} = \{r^1, \ldots, r^d\}$ any set of $d \geqslant 1$ rewards containing the unknown $r^E$, and $\pi^E \in \Pi^{NM}$ be any policy. Assume that the optimization problem in Line 2 is solved exactly. Then, choosing $\theta = \epsilon/(4H)$, with probability $1 - \delta$,*

---

**Algorithm 4:** Variant of `RS-KT` for a set of rewards

---

**Input :** Dataset $\mathcal{D}^E = \{(s_1^i, a_1^i, \ldots, s_H^i, a_H^i)\}_{i \in [\![N]\!]}$, set of rewards $\mathcal{R} = \{r^1, \ldots, r^d\}$, parameter $\theta$, transition model $p$

```
// Estimate the return distribution of the expert η_r^(π^E) for any r ∈ R
```
1 $\widehat{\eta}_r(g) \leftarrow \frac{1}{N} \sum_{i \in [\![N]\!]} \mathbb{1}\{\sum_{h=1}^H r_{\theta,h}(s_h^i, a_h^i) = g\} \quad \forall g \in \mathcal{Y}^\theta, \forall r \in \mathcal{R}$
```
// Compute the policy in Π(R^θ) closest to η̂_r ∀r ∈ R via Eq. (24)
```
2 $\widehat{\pi} \in \arg\min_{\pi \in \Pi(\mathcal{R}^\theta)} \max_{r \in \mathcal{R}} \mathcal{W}(\eta_{r_\theta}^\pi, \widehat{\eta})$
3 **Return** $\widehat{\pi}$

---

*the policy $\widehat{\pi}$ output by Algorithm 4 satisfies $\mathcal{W}(\eta_{r^E}^{\pi^E}, \eta_{r^E}^{\widehat{\pi}}) \leqslant \epsilon$, with:*

$$N \leqslant \mathcal{O}\left(\frac{H^2}{\epsilon^2} \ln \frac{d}{\delta}\right). \tag{23}$$

Interestingly, the bound here is still polynomial. Now, we show an extension of the LP formulation in Eq. (10) for addressing Line 2 of Algorithm 4. Specifically, we just construct an augmented MDP that keeps track of the past rewards for every possible reward in $\mathcal{R}$, and note that $\Pi(\mathcal{R}^\theta)$ describes the set of Markovian policies in this MDP. So, we want to match a sort of augmented return distribution for this problem:

$$\min_{d \in \mathcal{K}, \eta^1 \in \Delta(\mathcal{Y}^\theta), \ldots, \eta^d \in \Delta(\mathcal{Y}^\theta)} \max_{r^i \in \mathcal{R}} \mathcal{W}(\eta^i, \widehat{\eta}_{r^i}) \tag{24}$$

$$\text{s.t.} \quad \eta^i(g) = \sum_{s,a,g^1,\ldots,g^{i-1},g^{i+1},\ldots,g^d} d_H(s, g^1, \ldots, g^{i-1}, g - r_{\theta,H}^i(s,a), g^{i+1}, \ldots, g^d, a) \tag{25}$$

$$\forall g \in \mathcal{Y}^\theta \forall i \in [\![d]\!]. \tag{26}$$

Intuitively, the constraints above enforce that $\eta^i$ is the return distribution induced by the occupancy measure $d$ w.r.t. the reward $r_\theta^i$, for all the rewards $r^i \in \mathcal{R}$, and $\mathcal{K}$ denotes the set of feasible occupancy measures in this augmented MDP (Puterman, 1994):

$$\mathcal{K} := \left\{ d \in \Delta_{[\![H]\!]}^{\overline{\mathcal{S}} \times \mathcal{A}} \,\Big|\, \sum_a d_1(\overline{s}_0, a) = 1 \wedge \forall \overline{s} \in \overline{\mathcal{S}}, h \geqslant 2 : \sum_a d_h(\overline{s}, a) = \sum_{\overline{s}', a'} d_{h-1}(\overline{s}', a') \overline{p}_{h-1}(\overline{s}|\overline{s}', a') \right\},$$

where the state space is $\overline{\mathcal{S}} := \mathcal{S} \times \mathcal{Y}^\theta \times \ldots \times \mathcal{Y}^\theta$ $d$ times, $\overline{s}_0 := (s_0, 0, \ldots, 0)$ and the transition model is:

$$p_h(s', g^1, \ldots, g^d | s, \overline{g}^1, \ldots, \overline{g}^d, a) := p_h(s'|s,a) \mathbb{1}\{r_h^1(s,a) + \overline{g}^1 = g^1\} \ldots \mathbb{1}\{r_h^d(s,a) + \overline{g}^d = g^d\}.$$

In words, Eq. (24) searches for an occupancy measure $d \in \mathcal{K}$ that induces the return distribution $\eta^i$ closest to $\widehat{\eta}_{r^i}$. From such a solution, a policy $\widehat{\pi} \in \Pi(\mathcal{R}^\theta)$ with occupancy $d^{\widehat{\pi}} = d$ (and thus return distribution $\eta_{r_\theta^i}^{\widehat{\pi}} = \eta^i$ for all $i$) can be recovered via:

$$\widehat{\pi}(a|s, \omega) = \frac{d_h(s, G(\omega; r_\theta^1), \ldots, G(\omega; r_\theta^d), a)}{\sum_{a'} d_h(s, G(\omega; r_\theta^1), \ldots, G(\omega; r_\theta^d), a')} \quad \forall h \in [\![H]\!], \ s \in \mathcal{S}, \ a \in \mathcal{A}, \ \omega \in \Omega_h,$$

when the denominator is nonzero, and $\widehat{\pi}(a|s, \omega) = 1/A$ otherwise (Syed et al., 2008). We remark that, being the set $\mathcal{R}$ finite, then the minmax above can be formulated as an LP minimization problem.

### C.4.1 TECHNICAL RESULTS AND PROOFS FOR THE POLICY CLASS

**Lemma C.5.** *Let $\mathcal{M}$ be any MDP\R, $\mathcal{R} = \{r^1, \ldots, r^d\}$ any set of $d \geqslant 1$ rewards containing the unknown $r^E$, and $\pi^E \in \Pi^{NM}$ be any policy. Then, the policy $\pi_\mathcal{R} \in \Pi(\mathcal{R})$ is a minimizer of Eq. (20), and satisfies $\eta_{r^i}^{\pi_\mathcal{R}}(g) = \eta_{r^i}^{\pi^E}(g)$ for all $g$.*

*Proof.* To prove this result, we show that, for any $r^i \in \mathcal{R}$, the return distribution $\eta_{r^i}^{\pi_\mathcal{R}}$ coincides with the expert's return distribution $\eta_{r^i}^{\pi^E}$.

For any $r^i \in \mathcal{R}$ and $g' \in [0, H]$, we can write (we use $G_h^j := \sum_{h'=1}^{h-1} r_{h'}^j(s_{h'}, a_{h'})$ for all $j \in [\![d]\!]$):

$$\eta_{r^i}^{\pi_{\mathcal{R}}}(g') = \mathbb{P}^{\pi_{\mathcal{R}}}(G_H^i + r_H^i(s_H, a_H) = g')$$

$$\stackrel{(1)}{=} \sum_{g^i \in \mathcal{G}_{r^i, H}} \sum_{s \in \mathcal{S}} \mathbb{P}^{\pi_{\mathcal{R}}}(G_H^i = g^i, s_H = s, r_H^i(s, a_H) = g' - g^i)$$

$$= \sum_{g^1 \in \mathcal{G}_{r^1, H}} \cdots \sum_{g^d \in \mathcal{G}_{r^d, H}} \sum_{s \in \mathcal{S}}$$
$$\mathbb{P}^{\pi_{\mathcal{R}}}(G_H^1 = g^1, \ldots, G_H^d = g^d, s_H = s, r_H^i(s, a_H) = g' - g^i)$$

$$= \sum_{g^1 \in \mathcal{G}_{r^1, H}} \cdots \sum_{g^d \in \mathcal{G}_{r^d, H}} \sum_{s \in \mathcal{S}} \sum_{a \in \mathcal{A}: r_H^i(s, a) = g' - g^i}$$
$$\mathbb{P}^{\pi_{\mathcal{R}}}(G_H^1 = g^1, \ldots, G_H^d = g^d, s_H = s, a_H = a)$$

$$\stackrel{(2)}{=} \sum_{g^1 \in \mathcal{G}_{r^1, H}} \cdots \sum_{g^d \in \mathcal{G}_{r^d, H}} \sum_{s \in \mathcal{S}} \sum_{a \in \mathcal{A}: r_H^i(s, a) = g' - g^i}$$
$$\mathbb{P}^{\pi_{\mathcal{R}}}(G_H^1 = g^1, \ldots, G_H^d = g^d, s_H = s)\pi_{\mathcal{R}}(a|s, g^1, \ldots, g^d)$$

$$\stackrel{(3)}{=} \sum_{g^1 \in \mathcal{G}_{r^1, H}} \cdots \sum_{g^d \in \mathcal{G}_{r^d, H}} \sum_{s \in \mathcal{S}} \sum_{a \in \mathcal{A}: r_H^i(s, a) = g' - g^i}$$
$$\mathbb{P}^{\pi^E}(G_H^1 = g^1, \ldots, G_H^d = g^d, s_H = s)\pi_{\mathcal{R}}(a|s, g^1, \ldots, g^d)$$

$$= \sum_{g^1 \in \mathcal{G}_{r^1, H}} \cdots \sum_{g^d \in \mathcal{G}_{r^d, H}} \sum_{s \in \mathcal{S}} \mathbb{P}^{\pi^E}(G_H^1 = g^1, \ldots, G_H^d = g^d, s_H = s)$$
$$\sum_{a \in \mathcal{A}: r_H^i(s, a) = g' - g^i} \pi_{\mathcal{R}}(a|s, g^1, \ldots, g^d)$$

$$\stackrel{(4)}{=} \sum_{g^1 \in \mathcal{G}_{r^1, H}} \cdots \sum_{g^d \in \mathcal{G}_{r^d, H}} \sum_{s \in \mathcal{S}} \mathbb{P}^{\pi^E}(G_H^1 = g^1, \ldots, G_H^d = g^d, s_H = s)$$
$$\sum_{a \in \mathcal{A}: r_H^i(s, a) = g' - g^i} \frac{\mathbb{P}^{\pi^E}(s_H = s, \ a_H = a, \ G_H^1 = g^1, \ \ldots, \ G_H^d = g^d)}{\mathbb{P}^{\pi^E}(s_H = s, \ G_H^1 = g^1, \ \ldots, \ G_H^d = g^d)}$$

$$= \sum_{g^1 \in \mathcal{G}_{r^1, H}} \cdots \sum_{g^d \in \mathcal{G}_{r^d, H}} \sum_{s \in \mathcal{S}} \sum_{a \in \mathcal{A}} \mathbb{1}\{r_H^i(s, a) = g' - g^i\}$$
$$\mathbb{P}^{\pi^E}(s_H = s, \ a_H = a, \ G_H^1 = g^1, \ \ldots, \ G_H^d = g^d)$$

$$= \sum_{g^i \in \mathcal{G}_{r^i, H}} \sum_{s \in \mathcal{S}} \sum_{a \in \mathcal{A}} \mathbb{1}\{r_H^i(s, a) = g' - g^i\}\mathbb{P}^{\pi^E}(s_H = s, \ a_H = a, \ G_H^i = g^i)$$

$$= \eta_{r^i}^{\pi^E}(g'),$$

where at (1) we define symbol $\mathcal{G}_{r, H} := \{g \in [0, H-1] \mid \exists \omega \in \Omega_H : G(\omega; r) = g\}$ for any $r$, at (2) we recognize that, by definition, $\pi_{\mathcal{R}}$ takes actions only depending on the current state, stage and past rewards for any $r^i$, and we denote with brevity this fact with $\pi_{\mathcal{R}}(a|s, g^1, \ldots, g^d)$, at (3) we use Lemma C.9, at (4) we use the definition of $\pi_{\mathcal{R}}(a|s, g)$ (Eq. 21) where the denominator is not 0, noting that, in that case, the entire expression evaluates to 0. $\qquad\square$

**Lemma C.6.** *Let $\theta \in (0, 1]$. Let $\mathcal{M}$ be any MDP\R, $\mathcal{R} = \{r^1, \ldots, r^d\}$ any set of $d \geqslant 1$ rewards containing the unknown $r^E$, and $\pi^E \in \Pi^{NM}$ be any policy. Then, the policy $\pi_{\mathcal{R}^\theta} \in \Pi(\mathcal{R}^\theta)$ satisfies $\mathcal{W}\big(\eta_{r^E}^{\pi_{\mathcal{R}^\theta}}, \eta_{r^E}^{\pi^E}\big) \leqslant H\theta$.*

*Proof.* For any reward $r^i \in \mathcal{R}$, we can write:

$$\mathcal{W}\left(\eta_{r^i}^{\pi_{\mathcal{R}^\theta}}, \eta_{r^i}^{\pi^E}\right) \stackrel{(1)}{\leqslant} \mathcal{W}\left(\eta_{r^i}^{\pi_{\mathcal{R}^\theta}}, \eta_{r_\theta^i}^{\pi_{\mathcal{R}^\theta}}\right) + \mathcal{W}\left(\eta_{r_\theta^i}^{\pi_{\mathcal{R}^\theta}}, \eta_{r_\theta^i}^{\pi^E}\right) + \mathcal{W}\left(\eta_{r_\theta^i}^{\pi^E}, \eta_{r^i}^{\pi^E}\right)$$

$$\overset{(2)}{\leqslant} 2H\|r^i - r^i_\theta\|_\infty + \mathcal{W}\left(\eta^{\pi_{\mathcal{R}^\theta}}_{r^i_\theta}, \eta^{\pi^E}_{r^i_\theta}\right)$$

$$\overset{(3)}{\leqslant} H\theta + \mathcal{W}\left(\eta^{\pi_{\mathcal{R}^\theta}}_{r^i_\theta}, \eta^{\pi^E}_{r^i_\theta}\right)$$

$$\overset{(4)}{\leqslant} H\theta,$$

where at (1) we apply twice the triangle's inequality, at (2) we apply twice Lemma C.1, at (3) we realize that, by definition of $r^i_\theta$, it holds that $\|r^i - r^i_\theta\|_\infty \leqslant \theta/2$, and finally, at (4), we apply Lemma C.5 with set $\mathcal{R}^\theta$ and expert's policy $\pi^E$.

The proof is concluded after having observed that $r^E \in \mathcal{R}$ by hypothesis, and so these passages hold also for $r^E$. $\qquad\square$

**Lemma C.9.** *Let $\mathcal{M}$ be any MDP\R, $\mathcal{R} = \{r^1, \ldots, r^d\}$ any set of $d \geqslant 1$ rewards containing the unknown $r^E$, and $\pi \in \Pi^{NM}$ be any policy. Let $\pi_\mathcal{R} \in \Pi(\mathcal{R})$ be the policy defined as in Eq. (21) for expert's policy $\pi$. Then, for all $h \in [\![H]\!]$, $s \in \mathcal{S}$ and $g^1, \ldots, g^d \in [0, h-1]$, it holds that:*

$$\mathbb{P}^{\pi_\mathcal{R}}\left(G^1_h = g^1 \wedge \ldots \wedge G^d_h = g^d \wedge s_h = s\right) = \mathbb{P}^\pi\left(G^1_h = g^1 \wedge \ldots \wedge G^d_h = g^d \wedge s_h = s\right),$$

*where we used $G^i_h := \sum_{h'=1}^{h-1} r^i_{h'}(s_{h'}, a_{h'})$.*

*Proof.* We prove the result by induction. Let us begin with the base case: $h = 1$. For all $s \in \mathcal{S}$ and $g^1 \ldots, g^d \in \{0\}^d$, we have:

$$\mathbb{P}^{\pi_\mathcal{R}}\left(G^1_1 = g^1 \wedge \ldots \wedge G^d_1 = g^d \wedge s_1 = s\right) = \mathbb{1}\{g^1 = 0 \wedge \ldots \wedge g^d = 0\}\mathbb{1}\{s = s_0\}$$

$$= \mathbb{P}^\pi\left(G^1_1 = g^1 \wedge \ldots \wedge G^d_1 = g^d \wedge s_1 = s\right),$$

where we noticed that, for $h = 1$, no action is taken yet. Now, let us consider any stage $h \in \{2, 3, \ldots, H\}$, and let us make the induction hypothesis that, for all $h' \in [\![h-1]\!]$, for all $s \in \mathcal{S}$ and $g^1, \ldots, g^d \in [0, h'-1]^d$, it holds that:

$$\mathbb{P}^{\pi_\mathcal{R}}\left(G^1_{h'} = g^1 \wedge \ldots \wedge G^d_{h'} = g^d \wedge s_{h'} = s\right) = \mathbb{P}^\pi\left(G^1_{h'} = g^1 \wedge \ldots \wedge G^d_{h'} = g^d \wedge s_{h'} = s\right).$$

Then, for any $s' \in \mathcal{S}$ and $\overline{g}^1, \ldots, \overline{g}^d \in [0, h-1]^d$, we can write:

$$\mathbb{P}^{\pi_\mathcal{R}}(G^1_h = \overline{g}^1 \wedge \ldots \wedge G^d_h = \overline{g}^d \wedge s_h = s')$$

$$\overset{(1)}{=} \sum_{\substack{\omega \in \Omega_{h-1}, (s,a) \in \mathcal{S} \times \mathcal{A}: \\ G(\omega; r^i) + r^i_{h-1}(s,a) = \overline{g}^i \ \forall i}} \mathbb{P}^{\pi_\mathcal{R}}\left(\omega_{h-1} = \omega \wedge s_{h-1} = s \wedge a_{h-1} = a \wedge s_h = s'\right)$$

$$\overset{(2)}{=} \sum_{g^1 \in \mathcal{G}_{r^1, h-1}} \cdots \sum_{g^d \in \mathcal{G}_{r^d, h-1}} \sum_{\substack{\omega \in \Omega_{h-1}: \\ G(\omega; r^i) = g^i \ \forall i}} \sum_{\substack{(s,a) \in \mathcal{S} \times \mathcal{A}: \\ r^i_{h-1}(s,a) = \overline{g}^i - g^i \ \forall i}}$$

$$\mathbb{P}^{\pi_\mathcal{R}}\left(\omega_{h-1} = \omega \wedge s_{h-1} = s \wedge a_{h-1} = a \wedge s_h = s'\right)$$

$$\overset{(3)}{=} \sum_{g^1 \in \mathcal{G}_{r^1, h-1}} \cdots \sum_{g^d \in \mathcal{G}_{r^d, h-1}} \sum_{\substack{\omega \in \Omega_{h-1}: \\ G(\omega; r^i) = g^i \ \forall i}} \sum_{\substack{(s,a) \in \mathcal{S} \times \mathcal{A}: \\ r^i_{h-1}(s,a) = \overline{g}^i - g^i \ \forall i}} \mathbb{P}^{\pi_\mathcal{R}}\left(\omega_{h-1} = \omega \wedge s_{h-1} = s\right)$$

$$\cdot \mathbb{P}^{\pi_\mathcal{R}}\left(a_{h-1} = a | \omega, s\right) \mathbb{P}^{\pi_\mathcal{R}}\left(s_h = s' | \omega, s, a\right)$$

$$\overset{(4)}{=} \sum_{g^1 \in \mathcal{G}_{r^1, h-1}} \cdots \sum_{g^d \in \mathcal{G}_{r^d, h-1}} \sum_{\substack{\omega \in \Omega_{h-1}: \\ G(\omega; r^i) = g^i \ \forall i}} \sum_{\substack{(s,a) \in \mathcal{S} \times \mathcal{A}: \\ r^i_{h-1}(s,a) = \overline{g}^i - g^i \ \forall i}} \mathbb{P}^{\pi_\mathcal{R}}\left(\omega_{h-1} = \omega \wedge s_{h-1} = s\right)$$

$$\cdot \mathbb{P}^{\pi_\mathcal{R}}\left(a_{h-1} = a | \omega, s\right) p_{h-1}(s' | s, a)$$

$$\overset{(5)}{=} \sum_{g^1 \in \mathcal{G}_{r^1, h-1}} \cdots \sum_{g^d \in \mathcal{G}_{r^d, h-1}} \sum_{\substack{\omega \in \Omega_{h-1}: \\ G(\omega; r^i) = g^i \ \forall i}} \sum_{\substack{(s,a) \in \mathcal{S} \times \mathcal{A}: \\ r^i_{h-1}(s,a) = \overline{g}^i - g^i \ \forall i}} \mathbb{P}^{\pi_\mathcal{R}}\left(\omega_{h-1} = \omega \wedge s_{h-1} = s\right)$$

$$\cdot \pi_{\mathcal{R}}(a|\omega, s) p_{h-1}(s'|s,a)$$

$$\overset{(6)}{=} \sum_{g^1 \in \mathcal{G}_{r^1, h-1}} \cdots \sum_{g^d \in \mathcal{G}_{r^d, h-1}} \sum_{\substack{(s,a) \in \mathcal{S} \times \mathcal{A}: \\ r_{h-1}^i(s,a) = \overline{g}^i - g^i \ \forall i}} \sum_{\substack{\omega \in \Omega_{h-1}: \\ G(\omega; r^i) = g^i \ \forall i}} \mathbb{P}^{\pi_{\mathcal{R}}}(\omega_{h-1} = \omega \wedge s_{h-1} = s)$$

$$\cdot \pi_{\mathcal{R}}(a|\omega, s) p_{h-1}(s'|s,a)$$

$$\overset{(7)}{=} \sum_{g^1 \in \mathcal{G}_{r^1, h-1}} \cdots \sum_{g^d \in \mathcal{G}_{r^d, h-1}} \sum_{\substack{(s,a) \in \mathcal{S} \times \mathcal{A}: \\ r_{h-1}^i(s,a) = \overline{g}^i - g^i \ \forall i}}$$

$$\pi_{\mathcal{R}}(a|g^1, \ldots, g^d, s) p_{h-1}(s'|s,a) \sum_{\substack{\omega \in \Omega_{h-1}: \\ G(\omega; r^i) = g^i \ \forall i}} \mathbb{P}^{\pi_{\mathcal{R}}}(\omega_{h-1} = \omega \wedge s_{h-1} = s)$$

$$= \sum_{g^1 \in \mathcal{G}_{r^1, h-1}} \cdots \sum_{g^d \in \mathcal{G}_{r^d, h-1}} \sum_{\substack{(s,a) \in \mathcal{S} \times \mathcal{A}: \\ r_{h-1}^i(s,a) = \overline{g}^i - g^i \ \forall i}} \pi_{\mathcal{R}}(a|g^1, \ldots, g^d, s) p_{h-1}(s'|s,a)$$

$$\mathbb{P}^{\pi_{\mathcal{R}}}(G_{h-1}^1 = g^1 \wedge \ldots \wedge G_{h-1}^d = g^d \wedge s_{h-1} = s)$$

$$\overset{(8)}{=} \sum_{g^1 \in \mathcal{G}_{r^1, h-1}} \cdots \sum_{g^d \in \mathcal{G}_{r^d, h-1}} \sum_{\substack{(s,a) \in \mathcal{S} \times \mathcal{A}: \\ r_{h-1}^i(s,a) = \overline{g}^i - g^i \ \forall i}} \pi_{\mathcal{R}}(a|g^1, \ldots, g^d, s) p_{h-1}(s'|s,a)$$

$$\mathbb{P}^{\pi}(G_{h-1}^1 = g^1 \wedge \ldots \wedge G_{h-1}^d = g^d \wedge s_{h-1} = s)$$

$$\overset{(9)}{=} \sum_{g^1 \in \mathcal{G}_{r^1, h-1}} \cdots \sum_{g^d \in \mathcal{G}_{r^d, h-1}} \sum_{\substack{(s,a) \in \mathcal{S} \times \mathcal{A}: \\ r_{h-1}^i(s,a) = \overline{g}^i - g^i \ \forall i}}$$

$$\frac{\mathbb{P}^{\pi}(G_{h-1}^1 = g^1 \wedge \ldots \wedge G_{h-1}^d = g^d \wedge s_{h-1} = s \wedge a_{h-1} = a)}{\mathbb{P}^{\pi}(G_{h-1}^1 = g^1 \wedge \ldots \wedge G_{h-1}^d = g^d \wedge s_{h-1} = s)}$$

$$\cdot p_{h-1}(s'|s,a) \mathbb{P}^{\pi}(G_{h-1}^1 = g^1 \wedge \ldots \wedge G_{h-1}^d = g^d \wedge s_{h-1} = s)$$

$$= \sum_{g^1 \in \mathcal{G}_{r^1, h-1}} \cdots \sum_{g^d \in \mathcal{G}_{r^d, h-1}} \sum_{\substack{(s,a) \in \mathcal{S} \times \mathcal{A}: \\ r_{h-1}^i(s,a) = \overline{g}^i - g^i \ \forall i}}$$

$$\mathbb{P}^{\pi}(G_{h-1}^1 = g^1 \wedge \ldots \wedge G_{h-1}^d = g^d \wedge s_{h-1} = s \wedge a_{h-1} = a) p_{h-1}(s'|s,a)$$

$$= \sum_{g^1 \in \mathcal{G}_{r^1, h-1}} \cdots \sum_{g^d \in \mathcal{G}_{r^d, h-1}} \sum_{\substack{(s,a) \in \mathcal{S} \times \mathcal{A}: \\ r_{h-1}^i(s,a) = \overline{g}^i - g^i \ \forall i}}$$

$$\mathbb{P}^{\pi}(G_{h-1}^1 = g^1 \wedge \ldots \wedge G_{h-1}^d = g^d \wedge s_{h-1} = s \wedge a_{h-1} = a \wedge s_h = s')$$

$$= \mathbb{P}^{\pi}(G_h^1 = \overline{g}^1 \wedge \ldots \wedge G_h^d = \overline{g}^d \wedge s_h = s'),$$

where at (1) we use symbol $\omega_{h''}$ to denote the random trajectory long $h''$ stages, i.e., whose realizations belong to $\Omega_{h''}$, for any $h'' \in \llbracket H \rrbracket$. At (2) we define symbols $\mathcal{G}_{r,h} := \{g \in [0, h-1] \,|\, \exists \omega \in \Omega_h : G(\omega; r) = g\}$ for any $r$, at (3) we use the chain rule of conditional probabilities, at (4) we use the Markovianity of the environment, at (5) we note that $\mathbb{P}^{\pi_{\mathcal{R}}}(a_{h-1} = a|\omega, s)$ actually is $\pi_{\mathcal{R}}(a|\omega, s)$, at (6) we exchange the two summations, at (7) we recognize that, by definition, $\pi_{\mathcal{R}}(a|\omega, s)$ takes on the same value for all the trajectories $\omega$ with the same value of return for all rewards $r^i \in \mathcal{R}$, and thus we can bring this quantity outside the summation over the $\omega$. We use symbol $\pi_{\mathcal{R}}(a|g^1, \ldots, g^d, s)$ to denote this fact for brevity. We do the same also for $p_{h-1}(s'|s,a)$. At (8) we use the induction hypothesis, at (9) we replace $\pi_{\mathcal{R}}(a|g^1, \ldots, g^d, s)$ with its definition when $\mathbb{P}^{\pi}(G_{h-1}^1 = g^1 \wedge G_{h-1}^d = g^d \wedge s_{h-1} = s) > 0$ as in the opposite case the entire formula takes on value zero. $\qquad \square$

### C.4.2   PROOF OF THEOREM C.7

**Theorem C.7.** *Let $\epsilon \in (0, H]$ and $\delta \in (0, 1)$. Let $\mathcal{M}$ be any MDP\R, $\mathcal{R} = \{r^1, \ldots, r^d\}$ any set of $d \geqslant 1$ rewards containing the unknown $r^E$, and $\pi^E \in \Pi^{NM}$ be any policy. Then, choosing $\theta = \epsilon/(4H)$,*

*with probability at least* $1 - \delta$, *the policy* $\widehat{\pi}$ *output by Algorithm 3 satisfies* $\mathcal{W}(\eta_{r^E}^{\pi^E}, \eta_{r^E}^{\widehat{\pi}}) \leqslant \epsilon$, *with a number of samples:*

$$N \leqslant \widetilde{\mathcal{O}}\left( \frac{SH^{4+2d}d^2 \ln \frac{1}{\delta}}{\epsilon^{2+d}} \left( A + \ln \frac{1}{\delta} \right) \right). \tag{22}$$

*Proof.* We can write:

$$\mathcal{W}\left( \eta_{r^E}^{\pi^E}, \eta_{r^E}^{\widehat{\pi}} \right)$$

$$\overset{(1)}{\leqslant} \mathcal{W}\left( \eta_{r^E}^{\pi^E}, \eta_{r^E}^{\pi_{\mathcal{R}^\theta}} \right) + \mathcal{W}\left( \eta_{r^E}^{\pi_{\mathcal{R}^\theta}}, \eta_{r_\theta^E}^{\pi_{\mathcal{R}^\theta}} \right) + \mathcal{W}\left( \eta_{r_\theta^E}^{\pi_{\mathcal{R}^\theta}}, \eta_{r_\theta^E}^{\widehat{\pi}} \right) + \mathcal{W}\left( \eta_{r_\theta^E}^{\widehat{\pi}}, \eta_{r^E}^{\widehat{\pi}} \right)$$

$$\overset{(2)}{\leqslant} 2H\theta + \mathcal{W}\left( \eta_{r_\theta^E}^{\pi_{\mathcal{R}^\theta}}, \eta_{r_\theta^E}^{\widehat{\pi}} \right)$$

$$\overset{(3)}{\leqslant} 2H\theta + H \left\| \eta_{r_\theta^E}^{\pi_{\mathcal{R}^\theta}} - \eta_{r_\theta^E}^{\widehat{\pi}} \right\|_1$$

$$\overset{(4)}{\leqslant} 2H\theta + H \sum_{h \in [\![H]\!]} \sum_{g^1 \in \mathcal{G}_{r_\theta^1, h}} \cdots \sum_{g^d \in \mathcal{G}_{r_\theta^d, h}} \sum_{s \in \mathcal{S}} \mathbb{P}^{\pi_{\mathcal{R}^\theta}}(G_h^1 = g^1 \wedge \ldots \wedge G_h^d = g^d \wedge s_h = s)$$

$$\left\| \pi_{\mathcal{R}^\theta, h}(\cdot | s, g^1, \ldots, g^d) - \widehat{\pi}_h(\cdot | s, g^1, \ldots, g^d) \right\|_1$$

$$\overset{(5)}{=} 2H\theta + H \sum_{h \in [\![H]\!]} \sum_{g^1 \in \mathcal{G}_{r_\theta^1, h}} \cdots \sum_{g^d \in \mathcal{G}_{r_\theta^d, h}} \sum_{s \in \mathcal{S}} \mathbb{P}^{\pi^E}(G_h^1 = g^1 \wedge \ldots \wedge G_h^d = g^d \wedge s_h = s)$$

$$\left\| \pi_{\mathcal{R}^\theta, h}(\cdot | s, g^1, \ldots, g^d) - \widehat{\pi}_h(\cdot | s, g^1, \ldots, g^d) \right\|_1$$

$$\overset{(6)}{\leqslant} 2H\theta + H\epsilon',$$

where at (1) we apply triangle's inequality, at (2) we apply Lemma C.6 and Lemma C.1 twice, at (3) we use Particular Case 6.13 of Villani (2008), which tells us that we can upper bound the Wasserstein distance between two distributions supported on set $\mathcal{X}$ by the diameter of $\mathcal{X}$ ($\max_{x,x' \in \mathcal{X}} |x - x'|$) times the one norm between the two distributions. Since $\mathcal{X} = [0, H]$ in our case, we get the expression written above. At (4) we apply Lemma C.10 with the notation defined in that lemma with set $\mathcal{R}_\theta$ and policies $\pi_{\mathcal{R}^\theta}$ and $\widehat{\pi}$. Moreover, this holds recalling that $r^E \in \mathcal{R}$ and so $r_\theta^E \in \mathcal{R}^\theta$. At (5) we use Lemma C.9. Lastly, at (6) we apply Lemma C.11 with accuracy $\epsilon'$.

The result follows by imposing that $\epsilon' \leqslant \frac{\epsilon}{2H}$ and $2H\theta \leqslant \frac{\epsilon}{2}$, which can be achieved by taking $\epsilon' = \frac{\epsilon}{2H}$ and $\theta = \frac{\epsilon}{4H}$, and by observing that:

$$\overline{\mathcal{G}} := \sum_{h \in [\![H]\!]} \prod_{i \in [\![d]\!]} |\mathcal{G}_{r_\theta^i, h}|$$

$$\leqslant \sum_{h \in [\![H]\!]} \prod_{i \in [\![d]\!]} |\mathcal{Y}_h^\theta|$$

$$= \sum_{h \in [\![H]\!]} |\mathcal{Y}_h^\theta|^d$$

$$\leqslant \sum_{h \in [\![H]\!]} (1 + (h-1)/\theta)^d$$

$$\leqslant \mathcal{O}\left( H(H/\theta)^d \right)$$

$$\overset{(10)}{=} \mathcal{O}\left( H(H^2/\epsilon)^d \right),$$

where at (11) we used the previous choice $\theta = \frac{\epsilon}{4H}$.

Replacing into the number of samples in Lemma C.11 (and also $\epsilon' = \frac{\epsilon}{2H}$) we get the result:

$$N \leqslant \widetilde{\mathcal{O}}\left( \frac{SH^{4+2d}d^2 \ln \frac{1}{\delta}}{\epsilon^{2+d}} \left( A + \ln \frac{1}{\delta} \right) \right).$$

By using $\widetilde{\mathcal{O}}$ notation to hide logarithmic terms in $S, A, H, \frac{1}{\epsilon}, \ln \frac{1}{\delta}, d$, we get the result. $\qquad \square$

**Lemma C.10** (Error Propagation). *Let $\mathcal{M}$ be any MDP\R and $\mathcal{R} = \{r^1, \ldots, r^d\}$ any set of $d \geqslant 1$ rewards. For any pair of policies $\pi, \pi' \in \Pi^{NM}$ such that, for all $h \in [\![H]\!]$, $a \in \mathcal{A}$, $s \in \mathcal{S}$ and $\omega, \omega' \in \Omega_h$ with $G(\omega; r^i) = G(\omega'; r^i) \; \forall i \in [\![d]\!]$:*

$$\pi(a|\omega, s) = \pi(a|\omega', s) \qquad \wedge \qquad \pi'(a|\omega, s) = \pi'(a|\omega', s),$$

*it holds that, for any $r^i \in \mathcal{R}$:*

$$\left\| \eta_{r^i}^\pi - \eta_{r^i}^{\pi'} \right\|_1 \leqslant \sum_{h \in [\![H]\!]} \sum_{g^1 \in \mathcal{G}_{r^1, h}} \cdots \sum_{g^d \in \mathcal{G}_{r^d, h}} \sum_{s \in \mathcal{S}} \mathbb{P}^\pi (G_h^1 = g^1 \wedge \ldots \wedge G_h^d = g^d \wedge s_h = s)$$

$$\left\| \pi_h(\cdot|s, g^1, \ldots, g^d) - \pi'_h(\cdot|s, g^1, \ldots, g^d) \right\|_1,$$

*where $\mathcal{G}_{r,h} := \{g \in [0, h-1] \,|\, \exists \omega \in \Omega_h : G(\omega; r) = g\}$ for any reward $r$, $G_h^i := \sum_{h'=1}^{h-1} r_{h'}^i(s_{h'}, a_{h'})$ denotes the* random *return at stage $h$ under reward $r^i$, and $\pi_h(\cdot|s, g^1, \ldots, g^d)$ and $\pi'_h(\cdot|s, g^1, \ldots, g^d)$ denote the unique probability with which the policies $\pi$ and $\pi'$ prescribe actions in $s$ at $h$ under any trajectory $\omega \in \Omega_h$ with $G(\omega; r^i) = g^i \; \forall i \in [\![d]\!]$.*

*Proof.* To prove the result, we first demonstrate by induction that, for all $h \in [\![2, H]\!]$, it holds that:

$$\sum_{g^1 \in \mathcal{G}_{r^1, h}} \cdots \sum_{g^d \in \mathcal{G}_{r^d, h}} \sum_{s \in \mathcal{S}} \left| \mathbb{P}^\pi (G_h^1 = g^1 \wedge \ldots \wedge G_h^d = g^d \wedge s_h = s) \right.$$

$$\left. - \mathbb{P}^{\pi'} (G_h^1 = g^1 \wedge \ldots \wedge G_h^d = g^d \wedge s_h = s) \right|$$

$$\leqslant \sum_{h' \in [\![h-1]\!]} \sum_{g^1 \in \mathcal{G}_{r^1, h'}} \cdots \sum_{g^d \in \mathcal{G}_{r^d, h'}} \sum_{s \in \mathcal{S}} \mathbb{P}^\pi (G_{h'}^1 = g^1 \wedge \ldots \wedge G_{h'}^d = g^d \wedge s_{h'} = s)$$

$$\left\| \pi_{h'}(\cdot|s, g^1, \ldots, g^d) - \pi'_{h'}(\cdot|s, g^1, \ldots, g^d) \right\|_1.$$

We begin with the base case $h = 2$. We can write:

$$\sum_{g^1 \in \mathcal{G}_{r^1, 2}} \cdots \sum_{g^d \in \mathcal{G}_{r^d, 2}} \sum_{s \in \mathcal{S}} \left| \mathbb{P}^\pi (G_2^1 = g^1 \wedge \ldots \wedge G_2^d = g^d \wedge s_2 = s) \right.$$

$$\left. - \mathbb{P}^{\pi'} (G_2^1 = g^1 \wedge \ldots \wedge G_2^d = g^d \wedge s_2 = s) \right|$$

$$\overset{(1)}{=} \sum_{g^1 \in \mathcal{G}_{r^1, 2}} \cdots \sum_{g^d \in \mathcal{G}_{r^d, 2}} \sum_{s' \in \mathcal{S}} \left| \sum_{\substack{(s,a) \in \mathcal{S} \times \mathcal{A}: \\ r_1^i(s,a) = g^i \forall i}} \left( p(s'|s, a) \mathbb{1}\{s = s_0\} \pi(a|s) \right. \right.$$

$$\left. \left. - p(s'|s, a) \mathbb{1}\{s = s_0\} \pi'(a|s) \right) \right|$$

$$\overset{(2)}{\leqslant} \sum_{g^1 \in \mathcal{G}_{r^1, 2}} \cdots \sum_{g^d \in \mathcal{G}_{r^d, 2}} \sum_{\substack{a \in \mathcal{A}: \\ r_1^i(s_0, a) = g^i \forall i}} \sum_{s' \in \mathcal{S}} p(s'|s_0, a) \left| \pi(a|s_0) - \pi'(a|s_0) \right|$$

$$= \sum_{g^1 \in \mathcal{G}_{r^1, 2}} \cdots \sum_{g^d \in \mathcal{G}_{r^d, 2}} \sum_{\substack{a \in \mathcal{A}: \\ r_1^i(s_0, a) = g^i \forall i}} \left| \pi(a|s_0) - \pi'(a|s_0) \right|$$

$$= \sum_{a \in \mathcal{A}} \left| \pi(a|s_0) - \pi'(a|s_0) \right|$$

$$= \left\| \pi(\cdot|s_0) - \pi'(\cdot|s_0) \right\|_1$$

$$\overset{(3)}{=} \sum_{h' \in [\![1]\!]} \sum_{g^1 \in \mathcal{G}_{r^1, h'}} \cdots \sum_{g^d \in \mathcal{G}_{r^d, h'}} \sum_{s \in \mathcal{S}} \mathbb{P}^\pi (G_{h'}^1 = g^1 \wedge \ldots \wedge G_{h'}^d = g^d \wedge s_{h'} = s)$$

$$\left\| \pi_{h'}(\cdot|s, g^1, \ldots, g^d) - \pi'_{h'}(\cdot|s, g^1, \ldots, g^d) \right\|_1,$$

where at (1) we realize that in $\mathcal{M}$ the initial state is always $s_0$, and that the transition model is Markovian and independent of the policy, at (2) we apply triangle's inequality and keep only $s_0$

because of the indicator, and at (3) we have simply rewritten the expression in a more convenient way for proving the result.

Now, let us consider any stage $h \in [\![3, H]\!]$. Let us make the inductive hypothesis that:

$$
\sum_{g^1 \in \mathcal{G}_{r^1, h-1}} \cdots \sum_{g^d \in \mathcal{G}_{r^d, h-1}} \sum_{s \in \mathcal{S}} \left| \mathbb{P}^\pi(G^1_{h-1} = g^1 \wedge \ldots \wedge G^d_{h-1} = g^d \wedge s_{h-1} = s) \right.
$$

$$
\left. - \mathbb{P}^{\pi'}(G^1_{h-1} = g^1 \wedge \ldots \wedge G^d_{h-1} = g^d \wedge s_{h-1} = s) \right|
$$

$$
\leqslant \sum_{h' \in [\![h-2]\!]} \sum_{g^1 \in \mathcal{G}_{r^1, h'}} \cdots \sum_{g^d \in \mathcal{G}_{r^d, h'}} \sum_{s \in \mathcal{S}} \mathbb{P}^\pi(G^1_{h'} = g^1 \wedge \ldots \wedge G^d_{h'} = g^d \wedge s_{h'} = s)
$$

$$
\left\| \pi_{h'}(\cdot | s, g^1, \ldots, g^d) - \pi'_{h'}(\cdot | s, g^1, \ldots, g^d) \right\|_1.
$$

Then, we can write (we use symbol $\omega_h$ to denote the random trajectory $(s_1, a_1, \ldots, s_h, a_h)$ up to stage $h$):

$$
\sum_{g^1 \in \mathcal{G}_{r^1, h}} \cdots \sum_{g^d \in \mathcal{G}_{r^d, h}} \sum_{s \in \mathcal{S}} \left| \mathbb{P}^\pi(G^1_h = g^1 \wedge \ldots \wedge G^d_h = g^d \wedge s_h = s) \right.
$$

$$
\left. - \mathbb{P}^{\pi'}(G^1_h = g^1 \wedge \ldots \wedge G^d_h = g^d \wedge s_h = s) \right|
$$

$$
\overset{(4)}{=} \sum_{g^1 \in \mathcal{G}_{r^1, h}} \cdots \sum_{g^d \in \mathcal{G}_{r^d, h}} \sum_{s' \in \mathcal{S}} \left| \sum_{\overline{g}^1 \in \mathcal{G}_{r^1, h-1}} \cdots \sum_{\overline{g}^d \in \mathcal{G}_{r^d, h-1}} \sum_{\substack{\omega \in \Omega_{h-1}: \\ G(\omega; r^i) = \overline{g}^i \forall i}} \sum_{\substack{(s,a) \in \mathcal{S} \times \mathcal{A}: \\ r^i_{h-1}(s,a) = g^i - \overline{g}^i \forall i}} \right( 
$$

$$
\mathbb{P}^\pi(\omega_{h-2} = \omega \wedge s_{h-1} = s) \mathbb{P}^\pi(a_{h-1} = a \wedge s_h = s' | \omega, s)
$$

$$
\left. - \mathbb{P}^{\pi'}(\omega_{h-2} = \omega \wedge s_{h-1} = s) \mathbb{P}^{\pi'}(a_{h-1} = a \wedge s_h = s' | \omega, s) \right) \right|
$$

$$
\overset{(5)}{=} \sum_{g^1 \in \mathcal{G}_{r^1, h}} \cdots \sum_{g^d \in \mathcal{G}_{r^d, h}} \sum_{s' \in \mathcal{S}} \left| \sum_{\overline{g}^1 \in \mathcal{G}_{r^1, h-1}} \cdots \sum_{\overline{g}^d \in \mathcal{G}_{r^d, h-1}} \sum_{\substack{\omega \in \Omega_{h-1}: \\ G(\omega; r^i) = \overline{g}^i \forall i}} \sum_{\substack{(s,a) \in \mathcal{S} \times \mathcal{A}: \\ r^i_{h-1}(s,a) = g^i - \overline{g}^i \forall i}} \right(
$$

$$
\mathbb{P}^\pi(\omega_{h-2} = \omega \wedge s_{h-1} = s) \pi(a | \omega, s) p_{h-1}(s' | s, a)
$$

$$
\left. - \mathbb{P}^{\pi'}(\omega_{h-2} = \omega \wedge s_{h-1} = s) \pi'(a | \omega, s) p_{h-1}(s' | s, a) \right) \right|
$$

$$
= \sum_{g^1 \in \mathcal{G}_{r^1, h}} \cdots \sum_{g^d \in \mathcal{G}_{r^d, h}} \sum_{s' \in \mathcal{S}} \left| \sum_{\overline{g}^1 \in \mathcal{G}_{r^1, h-1}} \cdots \sum_{\overline{g}^d \in \mathcal{G}_{r^d, h-1}} \sum_{\substack{\omega \in \Omega_{h-1}: \\ G(\omega; r^i) = \overline{g}^i \forall i}} \sum_{\substack{(s,a) \in \mathcal{S} \times \mathcal{A}: \\ r^i_{h-1}(s,a) = g^i - \overline{g}^i \forall i}} \right.
$$

$$
p_{h-1}(s' | s, a) \Big( \mathbb{P}^\pi(\omega_{h-2} = \omega \wedge s_{h-1} = s) \pi(a | \omega, s)
$$

$$
- \mathbb{P}^{\pi'}(\omega_{h-2} = \omega \wedge s_{h-1} = s) \pi'(a | \omega, s)
$$

$$
\left. \pm \mathbb{P}^\pi(\omega_{h-2} = \omega \wedge s_{h-1} = s) \pi'(a | \omega, s) \Big) \right|
$$

$$
\overset{(6)}{\leqslant} \sum_{g^1 \in \mathcal{G}_{r^1, h}} \cdots \sum_{g^d \in \mathcal{G}_{r^d, h}} \sum_{s' \in \mathcal{S}} \sum_{\overline{g}^1 \in \mathcal{G}_{r^1, h-1}} \cdots \sum_{\overline{g}^d \in \mathcal{G}_{r^d, h-1}} \sum_{\substack{\omega \in \Omega_{h-1}: \\ G(\omega; r^i) = \overline{g}^i \forall i}} \sum_{\substack{(s,a) \in \mathcal{S} \times \mathcal{A}: \\ r^i_{h-1}(s,a) = g^i - \overline{g}^i \forall i}}
$$

$$
p_{h-1}(s' | s, a) \cdot \mathbb{P}^\pi(\omega_{h-2} = \omega \wedge s_{h-1} = s) \left| \pi(a | \omega, s) - \pi'(a | \omega, s) \right|
$$

$$
+ \sum_{g^1 \in \mathcal{G}_{r^1, h}} \cdots \sum_{g^d \in \mathcal{G}_{r^d, h}} \sum_{s' \in \mathcal{S}} \sum_{\overline{g}^1 \in \mathcal{G}_{r^1, h-1}} \cdots \sum_{\overline{g}^d \in \mathcal{G}_{r^d, h-1}}
$$

$$
\left| \sum_{\substack{\omega \in \Omega_{h-1}: \\ G(\omega; r^i) = \overline{g}^i \forall i}} \sum_{\substack{(s,a) \in \mathcal{S} \times \mathcal{A}: \\ r^i_{h-1}(s,a) = g^i - \overline{g}^i \forall i}} p_{h-1}(s' | s, a) \pi'(a | \omega, s) \right.
$$

$$
\left. \cdot \Big( \mathbb{P}^\pi(\omega_{h-2} = \omega \wedge s_{h-1} = s) - \mathbb{P}^{\pi'}(\omega_{h-2} = \omega \wedge s_{h-1} = s) \Big) \right|
$$

$$\overset{(7)}{\leqslant} \sum_{g^1 \in \mathcal{G}_{r^1,h}} \cdots \sum_{g^d \in \mathcal{G}_{r^d,h}} \sum_{\overline{g}^1 \in \mathcal{G}_{r^1,h-1}} \cdots \sum_{\overline{g}^d \in \mathcal{G}_{r^d,h-1}} \sum_{\substack{\omega \in \Omega_{h-1}: \\ G(\omega;r^i)=\overline{g}^i \forall i}} \sum_{\substack{(s,a) \in \mathcal{S} \times \mathcal{A}: \\ r^i_{h-1}(s,a)=g^i-\overline{g}^i \forall i}}$$

$$\mathbb{P}^\pi\big(\omega_{h-2}=\omega \wedge s_{h-1}=s\big) \Big| \pi(a|\omega,s) - \pi'(a|\omega,s)\Big|$$

$$+ \sum_{g^1 \in \mathcal{G}_{r^1,h}} \cdots \sum_{g^d \in \mathcal{G}_{r^d,h}} \sum_{s' \in \mathcal{S}} \sum_{\overline{g}^1 \in \mathcal{G}_{r^1,h-1}} \cdots \sum_{\overline{g}^d \in \mathcal{G}_{r^d,h-1}} \sum_{\substack{(s,a) \in \mathcal{S} \times \mathcal{A}: \\ r^i_{h-1}(s,a)=g^i-\overline{g}^i \forall i}}$$

$$p_{h-1}(s'|s,a) \Big| \sum_{\substack{\omega \in \Omega_{h-1}: \\ G(\omega;r^i)=\overline{g}^i \forall i}} \pi'(a|\omega,s)$$

$$\cdot \Big( \mathbb{P}^\pi\big(\omega_{h-2}=\omega \wedge s_{h-1}=s\big) - \mathbb{P}^{\pi'}\big(\omega_{h-2}=\omega \wedge s_{h-1}=s\big)\Big)\Big|$$

$$\overset{(8)}{=} \sum_{\overline{g}^1 \in \mathcal{G}_{r^1,h-1}} \cdots \sum_{\overline{g}^d \in \mathcal{G}_{r^d,h-1}} \sum_{(s,a) \in \mathcal{S} \times \mathcal{A}} \Big| \pi_{h-1}(a|s,\overline{g}^1,\ldots,\overline{g}^d) - \pi'_{h-1}(a|s,\overline{g}^1,\ldots,\overline{g}^d)\Big|$$

$$\cdot \sum_{\substack{\omega \in \Omega_{h-1}: \\ G(\omega;r^i)=\overline{g}^i \forall i}} \mathbb{P}^\pi\big(\omega_{h-2}=\omega \wedge s_{h-1}=s\big)$$

$$+ \sum_{g^1 \in \mathcal{G}_{r^1,h}} \cdots \sum_{g^d \in \mathcal{G}_{r^d,h}} \sum_{\overline{g}^1 \in \mathcal{G}_{r^1,h-1}} \cdots \sum_{\overline{g}^d \in \mathcal{G}_{r^d,h-1}} \sum_{\substack{(s,a) \in \mathcal{S} \times \mathcal{A}: \\ r^i_{h-1}(s,a)=g^i-\overline{g}^i \forall i}}$$

$$\pi'_{h-1}(a|s,\overline{g}^1,\ldots,\overline{g}^d) \Big| \sum_{\substack{\omega \in \Omega_{h-1}: \\ G(\omega;r^i)=\overline{g}^i \forall i}}$$

$$\cdot \Big( \mathbb{P}^\pi\big(\omega_{h-2}=\omega \wedge s_{h-1}=s\big) - \mathbb{P}^{\pi'}\big(\omega_{h-2}=\omega \wedge s_{h-1}=s\big)\Big)\Big|$$

$$= \sum_{\overline{g}^1 \in \mathcal{G}_{r^1,h-1}} \cdots \sum_{\overline{g}^d \in \mathcal{G}_{r^d,h-1}} \sum_{(s,a) \in \mathcal{S} \times \mathcal{A}} \Big| \pi_{h-1}(a|s,\overline{g}^1,\ldots,\overline{g}^d) - \pi'_{h-1}(a|s,\overline{g}^1,\ldots,\overline{g}^d)\Big|$$

$$\mathbb{P}^\pi\big(G^1_{h-1}=\overline{g}^1 \wedge \ldots, \wedge G^d_{h-1}=\overline{g}^d \wedge s_{h-1}=s\big)$$

$$+ \sum_{g^1 \in \mathcal{G}_{r^1,h}} \cdots \sum_{g^d \in \mathcal{G}_{r^d,h}} \sum_{\overline{g}^1 \in \mathcal{G}_{r^1,h-1}} \cdots \sum_{\overline{g}^d \in \mathcal{G}_{r^d,h-1}} \sum_{\substack{(s,a) \in \mathcal{S} \times \mathcal{A}: \\ r^i_{h-1}(s,a)=g^i-\overline{g}^i \forall i}}$$

$$\pi'_{h-1}(a|s,\overline{g}^1,\ldots,\overline{g}^d)$$

$$\Big| \mathbb{P}^\pi\big(G^1_{h-1}=\overline{g}^1 \wedge \ldots, \wedge G^d_{h-1}=\overline{g}^d \wedge s_{h-1}=s\big)$$

$$- \mathbb{P}^{\pi'}\big(G^1_{h-1}=\overline{g}^1 \wedge \ldots, \wedge G^d_{h-1}=\overline{g}^d \wedge s_{h-1}=s\big)\Big|$$

$$\overset{(9)}{=} \sum_{\overline{g}^1 \in \mathcal{G}_{r^1,h-1}} \cdots \sum_{\overline{g}^d \in \mathcal{G}_{r^d,h-1}} \sum_{s \in \mathcal{S}} \mathbb{P}^\pi\big(G^1_{h-1}=\overline{g}^1 \wedge \ldots, \wedge G^d_{h-1}=\overline{g}^d \wedge s_{h-1}=s\big)$$

$$\Big\| \pi_{h-1}(\cdot|s,\overline{g}^1,\ldots,\overline{g}^d) - \pi'_{h-1}(\cdot|s,\overline{g}^1,\ldots,\overline{g}^d)\Big\|_1$$

$$+ \sum_{\overline{g}^1 \in \mathcal{G}_{r^1,h-1}} \cdots \sum_{\overline{g}^d \in \mathcal{G}_{r^d,h-1}} \sum_{s \in \mathcal{S}}$$

$$\Big| \mathbb{P}^\pi\big(G^1_{h-1}=\overline{g}^1 \wedge \ldots, \wedge G^d_{h-1}=\overline{g}^d \wedge s_{h-1}=s\big)$$

$$- \mathbb{P}^{\pi'}\big(G^1_{h-1}=\overline{g}^1 \wedge \ldots, \wedge G^d_{h-1}=\overline{g}^d \wedge s_{h-1}=s\big)\Big|$$

$$\overset{(10)}{\leqslant} \sum_{\overline{g}^1 \in \mathcal{G}_{r^1,h-1}} \cdots \sum_{\overline{g}^d \in \mathcal{G}_{r^d,h-1}} \sum_{s \in \mathcal{S}} \mathbb{P}^\pi\big(G^1_{h-1}=\overline{g}^1 \wedge \ldots, \wedge G^d_{h-1}=\overline{g}^d \wedge s_{h-1}=s\big)$$

$$\left\|\pi_{h-1}(\cdot|s,\overline{g}^1,\dots,\overline{g}^d) - \pi'_{h-1}(\cdot|s,\overline{g}^1,\dots,\overline{g}^d)\right\|_1$$

$$+ \sum_{h'\in[\![h-2]\!]} \sum_{g^1\in\mathcal{G}_{r^1,h'}} \cdots \sum_{g^d\in\mathcal{G}_{r^d,h'}} \sum_{s\in\mathcal{S}}$$

$$\mathbb{P}^\pi(G^1_{h'} = g^1 \wedge \dots \wedge G^d_{h'} = g^d \wedge s_{h'} = s)$$

$$\left\|\pi_{h'}(\cdot|s,g^1,\dots,g^d) - \pi'_{h'}(\cdot|s,g^1,\dots,g^d)\right\|_1$$

$$= \sum_{h'\in[\![h-1]\!]} \sum_{g^1\in\mathcal{G}_{r^1,h'}} \cdots \sum_{g^d\in\mathcal{G}_{r^d,h'}} \sum_{s\in\mathcal{S}}$$

$$\mathbb{P}^\pi(G^1_{h'} = g^1 \wedge \dots \wedge G^d_{h'} = g^d \wedge s_{h'} = s)$$

$$\left\|\pi_{h'}(\cdot|s,g^1,\dots,g^d) - \pi'_{h'}(\cdot|s,g^1,\dots,g^d)\right\|_1,$$

where at (4) we use the chain rule of conditional probabilities, at (5) we do it again, and we recognize the policies $\pi$ and $\pi'$, and also that the transition model is Markovian, at (6) we use triangle's inequality to split the summations and bring the absolute value inside, at (7), in the first term, we note that $p_{h-1}(s'|s,a)$ is the only term that depends on $s'$ and that it sums to 1, while in the second term we exchange the order of two summations and apply triangle's inequality to bring one inside, at (8), in the first term, we first remove the summation on $g^i$ along with the indicator function that forces us to consider a subset of state-action pairs, and then we exchange two other summations and note that the policies do not depend by hypothesis on the entire past trajectory, but just on the return so far for any $r^i \in \mathcal{R}$. Instead, in the second term, we use that $\sum_{s'\in\mathcal{S}} p_{h-1}(s'|s,a) = 1$, and also that, by hypothesis, $\pi'$ does not depend on the entire past trajectory, but just on $g^1,\dots,g^d$. At (9), i.a., we use that $\sum_{g^1\in\mathcal{G}_{r^1,h}} \cdots \sum_{g^d\in\mathcal{G}_{r^d,h}} \mathbb{1}\{r^i_{h-1}(s,a) = g^i - \overline{g}^i \forall i\} = 1$ and that $\sum_{a\in\mathcal{A}} \pi'_{h-1}(a|s,\overline{g}^1,\dots,\overline{g}^d) = 1$. Finally, at (10), we apply the inductive hypothesis.

Thanks to this result, we can finally prove the claim in the lemma, using passages analogous to those above, with the difference that we do not have the summation over the states at the current stage (i.e., $H + 1$). For any $r^j \in \mathcal{R}$:

$$\left\|\eta^\pi_{r^j} - \eta^{\pi'}_{r^j}\right\|_1 = \sum_{g\in\mathcal{G}_{r^j,H+1}} \left|\eta^\pi_{r^j}(g) - \eta^{\pi'}_{r^j}(g)\right|$$

$$= \sum_{g\in\mathcal{G}_{r^j,H+1}} \left|\mathbb{P}^\pi(G^j_{H+1} = g) - \mathbb{P}^{\pi'}(G^j_{H+1} = g)\right|$$

$$= \sum_{g\in\mathcal{G}_{r^j,H+1}} \Bigg| \sum_{g^1\in\mathcal{G}_{r^1,H}} \cdots \sum_{g^d\in\mathcal{G}_{r^d,H}} \sum_{\substack{\omega\in\Omega_H:\\ G(\omega;r^i)=g^i\forall i}} \sum_{\substack{(s,a)\in\mathcal{S}\times\mathcal{A}:\\ r^j_H(s,a)=g-g^j}}$$

$$\left(\mathbb{P}^\pi(\omega_{H-1} = \omega \wedge s_H = s)\pi(a|\omega,s)\right.$$

$$- \mathbb{P}^{\pi'}(\omega_{H-1} = \omega \wedge s_H = s)\pi'(a|\omega,s)$$

$$\left.\pm \mathbb{P}^\pi(\omega_{H-1} = \omega \wedge s_H = s)\pi'(a|\omega,s)\right)\Bigg|$$

$$\overset{(11)}{\leqslant} \sum_{g^1\in\mathcal{G}_{r^1,H}} \cdots \sum_{g^d\in\mathcal{G}_{r^d,H}} \sum_{s\in\mathcal{S}} \mathbb{P}^\pi(G^1_H = g^1 \wedge \dots, \wedge G^d_H = g^d \wedge s_H = s)$$

$$\left\|\pi_H(\cdot|s,g^1,\dots,g^d) - \pi'_H(\cdot|s,g^1,\dots,g^d)\right\|_1$$

$$+ \sum_{h'\in[\![H-1]\!]} \sum_{g^1\in\mathcal{G}_{r^1,h'}} \cdots \sum_{g^d\in\mathcal{G}_{r^d,h'}} \sum_{s\in\mathcal{S}}$$

$$\mathbb{P}^\pi(G^1_{h'} = g^1 \wedge \dots \wedge G^d_{h'} = g^d \wedge s_{h'} = s)$$

$$\left\|\pi_{h'}(\cdot|s,g^1,\dots,g^d) - \pi'_{h'}(\cdot|s,g^1,\dots,g^d)\right\|_1,$$

where at (11) we made the same passages as above from (6) on, all in one, with the only differences that we sum only over $g \in \mathcal{G}_{r^j,H+1}$ instead of doing so for any reward of $\mathcal{R}$, and also that we do

not have the sum over the next state $s'$. The result follows by summing the two terms in the last expression written.

This concludes the proof. $\qquad\square$

**Lemma C.11** (Concentration). *Let $\epsilon \in (0, H]$ and $\delta \in (0, 1)$. Let $\mathcal{M}$ be any MDP\\$R$ and $\mathcal{R} = \{r^1, \ldots, r^d\}$ any set of $d \geqslant 1$ rewards, $\pi^E \in \Pi^{NM}$ be any expert's policy, and $\widehat{\pi}$ be the output of Algorithm 3. Then, with probability $1 - \delta$, we have that (we use the notation in Lemma C.10):*

$$\sum_{h\in\llbracket H\rrbracket}\sum_{g^1\in\mathcal{G}_{r^1_\theta,h}}\cdots\sum_{g^d\in\mathcal{G}_{r^d_\theta,h}}\sum_{s\in\mathcal{S}}\mathbb{P}^{\pi^E}(G^1_h = g^1 \wedge \ldots \wedge G^d_h = g^d \wedge s_h = s)$$

$$\left\|\pi_{\mathcal{R}^\theta,h}(\cdot|s, g^1, \ldots, g^d) - \widehat{\pi}_h(\cdot|s, g^1, \ldots, g^d)\right\|_1 \leqslant \epsilon,$$

*with a number of samples:*

$$N \leqslant \frac{193 SH\overline{\mathcal{G}}\ln\frac{2S\overline{\mathcal{G}}}{\delta}}{\epsilon^2}\left(\ln\frac{2S\overline{\mathcal{G}}}{\delta} + (A-1)\ln\left(\frac{128 eSH\overline{\mathcal{G}}\ln\frac{2S\overline{\mathcal{G}}}{\delta}}{\epsilon^2}\right)\right),$$

*where $\overline{\mathcal{G}} := \sum_{h\in\llbracket H\rrbracket}\prod_{i\in\llbracket d\rrbracket}|\mathcal{G}_{r^i_\theta,h}|$.*

*Proof.* We can write:

$$\sum_{h\in\llbracket H\rrbracket}\sum_{g^1\in\mathcal{G}_{r^1_\theta,h}}\cdots\sum_{g^d\in\mathcal{G}_{r^d_\theta,h}}\sum_{s\in\mathcal{S}}\mathbb{P}^{\pi^E}(G^1_h = g^1 \wedge \ldots \wedge G^d_h = g^d \wedge s_h = s)$$

$$\left\|\pi_{\mathcal{R}^\theta,h}(\cdot|s, g^1, \ldots, g^d) - \widehat{\pi}_h(\cdot|s, g^1, \ldots, g^d)\right\|_1$$

$$= \sum_{h\in\llbracket H\rrbracket}\sum_{g^1\in\mathcal{G}_{r^1_\theta,h}}\cdots\sum_{g^d\in\mathcal{G}_{r^d_\theta,h}}\sum_{s\in\mathcal{S}}\mathbb{P}^{\pi^E}(G^1_h = g^1 \wedge \ldots \wedge G^d_h = g^d \wedge s_h = s)$$

$$\sum_{a\in\mathcal{A}}\left|\pi_{\mathcal{R}^\theta,h}(a|s, g^1, \ldots, g^d) - \left(\frac{M_h(s, g^1, \ldots, g^d, a)}{\sum_{a'} M_h(s, g^1, \ldots, g^d, a')}\right.\right.$$

$$\left.\left.\cdot \mathbb{1}\{\sum_{a'} M_h(s, g^1, \ldots, g^d, a') > 0\} + \frac{1}{A}\mathbb{1}\{\sum_{a'} M_h(s, g^1, \ldots, g^d, a') = 0\}\right)\right|$$

$$\overset{(1)}{\leqslant} \sum_{h\in\llbracket H\rrbracket}\sum_{g^1\in\mathcal{G}_{r^1_\theta,h}}\cdots\sum_{g^d\in\mathcal{G}_{r^d_\theta,h}}\sum_{s\in\mathcal{S}}\mathbb{P}^{\pi^E}(G^1_h = g^1 \wedge \ldots \wedge G^d_h = g^d \wedge s_h = s)$$

$$\cdot 2\sqrt{2}\sqrt{\frac{\ln\frac{2S\overline{\mathcal{G}}}{\delta} + (A-1)\ln\left(e\left(1 + \frac{\sum_{a'} M_h(s,g^1,\ldots,g^d,a')}{A-1}\right)\right)}{\sum_{a'} M_h(s, g^1, \ldots, g^d, a')}}$$

$$\overset{(2)}{\leqslant} 2\sqrt{2}\sqrt{\ln\frac{2S\overline{\mathcal{G}}}{\delta} + (A-1)\ln\left(e\left(1 + \frac{N}{A-1}\right)\right)}$$

$$\cdot \sum_{h\in\llbracket H\rrbracket}\sum_{g^1\in\mathcal{G}_{r^1_\theta,h}}\cdots\sum_{g^d\in\mathcal{G}_{r^d_\theta,h}}\sum_{s\in\mathcal{S}}\mathbb{P}^{\pi^E}(G^1_h = g^1 \wedge \ldots \wedge G^d_h = g^d \wedge s_h = s)$$

$$\sqrt{\frac{1}{\sum_{a'} M_h(s, g^1, \ldots, g^d, a')}}$$

$$\overset{(3)}{\leqslant} 2\sqrt{2}\sqrt{\ln\frac{2S\overline{\mathcal{G}}}{\delta} + (A-1)\ln\left(e\left(1 + \frac{N}{A-1}\right)\right)}$$

$$\cdot \sum_{h\in\llbracket H\rrbracket}\sum_{g^1\in\mathcal{G}_{r^1_\theta,h}}\cdots\sum_{g^d\in\mathcal{G}_{r^d_\theta,h}}\sum_{s\in\mathcal{S}}$$

$$\sqrt{\frac{8 \ln \frac{2S\overline{\mathcal{G}}}{\delta} \mathbb{P}^{\pi^E}(G_h^1 = g^1 \wedge \ldots \wedge G_h^d = g^d \wedge s_h = s)}{N}}$$

$$= 8 \sqrt{\frac{\ln \frac{2S\overline{\mathcal{G}}}{\delta}}{N}} \sqrt{\ln \frac{2S\overline{\mathcal{G}}}{\delta} + (A-1)\ln\left(e\left(1 + \frac{N}{A-1}\right)\right)}$$

$$\cdot \sum_{h \in \llbracket H \rrbracket} \sum_{g^1 \in \mathcal{G}_{r_\theta^1,h}} \cdots \sum_{g^d \in \mathcal{G}_{r_\theta^d,h}} \sum_{s \in \mathcal{S}}$$

$$\sqrt{\mathbb{P}^{\pi^E}(G_h^1 = g^1 \wedge \ldots \wedge G_h^d = g^d \wedge s_h = s)}$$

$$\overset{(4)}{\leqslant} 8\sqrt{\frac{\ln \frac{2S\overline{\mathcal{G}}}{\delta}}{N}} \sqrt{\ln \frac{2S\overline{\mathcal{G}}}{\delta} + (A-1)\ln\left(e\left(1 + \frac{N}{A-1}\right)\right)} \sqrt{S\overline{\mathcal{G}}}$$

$$\sqrt{\sum_{h \in \llbracket H \rrbracket} \sum_{g^1 \in \mathcal{G}_{r_\theta^1,h}} \cdots \sum_{g^d \in \mathcal{G}_{r_\theta^d,h}} \sum_{s \in \mathcal{S}} \mathbb{P}^{\pi^E}(G_h^1 = g^1 \wedge \ldots \wedge G_h^d = g^d \wedge s_h = s)}$$

$$= 8\sqrt{\frac{S\overline{\mathcal{G}} \ln \frac{2S\overline{\mathcal{G}}}{\delta}}{N}} \sqrt{\ln \frac{2S\overline{\mathcal{G}}}{\delta} + (A-1)\ln\left(e\left(1 + \frac{N}{A-1}\right)\right)} \sqrt{\sum_{h \in \llbracket H \rrbracket} 1}$$

$$\overset{(5)}{\leqslant} 8\sqrt{\frac{SH\overline{\mathcal{G}} \ln \frac{2S\overline{\mathcal{G}}}{\delta}}{N}} \sqrt{\ln \frac{2S\overline{\mathcal{G}}}{\delta} + (A-1)\ln\left(e\left(1 + \frac{N}{A-1}\right)\right)},$$

where at (1) we use that, if $\sum_{a'} M_h(s, g^1, \ldots, g^d, a') = 0$, then:

$$\sum_{a \in \mathcal{A}} \left| \pi_{\mathcal{R}^\theta,h}(a|s, g^1, \ldots, g^d) - \left( \frac{M_h(s, g, a)}{\sum_{a'} M_h(s, g^1, \ldots, g^d, a')} \mathbb{1}\{\sum_{a'} M_h(s, g^1, \ldots, g^d, a') > 0\} \right. \right.$$

$$\left. \left. + \frac{1}{A} \mathbb{1}\{\sum_{a'} M_h(s, g^1, \ldots, g^d, a') = 0\}\right) \right|$$

$$= \sum_{a \in \mathcal{A}} \left| \pi_{\mathcal{R}^\theta,h}(a|s, g^1, \ldots, g^d) - \frac{1}{A} \right|$$

$$\leqslant 2,$$

as we the total variation distance between two probability distributions cannot exceed 1. Instead, if $\sum_{a'} M_h(s, g^1, \ldots, g^d, a') > 0$, *conditioning* on $\sum_{a'} M_h(s, g^1, \ldots, g^d, a')$, at all $s, g^1, \ldots, g^d$ where $\mathbb{P}^{\pi^E}(G_h = g \wedge s_h = s) > 0$, we note that $M_h(s, g^1, \ldots, g^d, a)/\sum_{a'} M_h(s, g^1, \ldots, g^d, a')$ is the empirical vector of probabilities of $\pi_{\mathcal{R}^\theta,h}(a|s, g^1, \ldots, g^d)$ (recall its definition from Eq. 21), thus we can apply Lemma 8 of Kaufmann et al. (2021) to get that, for any $\delta \in (0, 1)$:

$$\mathbb{P}^{\pi^E}\left( KL\left( \frac{M_h(s, g^1, \ldots, g^d, \cdot)}{\sum_{a'} M_h(s, g^1, \ldots, g^d, a')} \Big\| \pi_{\mathcal{R}^\theta,h}(\cdot|s, g^1, \ldots, g^d) \right) \right.$$

$$\left. \leqslant \frac{\ln \frac{1}{\delta} + (A-1)\ln\left(e\left(1 + \frac{\sum_{a'} M_h(s, g^1, \ldots, g^d, a')}{A-1}\right)\right)}{\sum_{a'} M_h(s, g^1, \ldots, g^d, a')} \right) \geqslant 1 - \delta.$$

Combining this result with the Pinsker's inequality, that tells us that $\|x - y\|_1 \leqslant \sqrt{2KL(x\|y)}$, and with a union bound over all $h \in \llbracket H \rrbracket$, $s \in \mathcal{S}$, $g^1 \in \mathcal{G}_{r_\theta^1,h}, \ldots, g^d \in \mathcal{G}_{r_\theta^d,h}$, we get the passage in (1) w.p. $1 - \delta/2$. Note that we add an additional 2 for the case $\sum_{a'} M_h(s, g^1, \ldots, g^d, a') = 0$, and we define $\overline{\mathcal{G}} := \sum_{h \in \llbracket H \rrbracket} \prod_{i \in \llbracket d \rrbracket} |\mathcal{G}_{r_\theta^i,h}|$. At (2) we bound $\sum_{a'} M_h(s, g^1, \ldots, g^d, a') \leqslant N$, and bring that quantity outside, at (3) we apply Lemma A.1 of Xie et al. (2021), after having noticed that $\sum_{a'} M_h(s, g^1, \ldots, g^d, a') \sim \text{Bin}\left(N, \mathbb{P}^{\pi^E}(G_h^1 = g^1 \wedge \ldots \wedge G_h^d = g^d \wedge s_h = s)\right)$, and make it hold for all $s, g^1, \ldots, g^d, h$ w.p. $1 - \delta/2$. At (4) and (5) we apply the Cauchy-Schwarz's inequality.

Now, we impose that this quantity is smaller than $\epsilon$:

$$8\sqrt{\frac{SH\overline{\mathcal{G}}\ln\frac{2S\overline{\mathcal{G}}}{\delta}}{N}}\sqrt{\ln\frac{2S\overline{\mathcal{G}}}{\delta}+(A-1)\ln\left(e\left(1+\frac{N}{A-1}\right)\right)}\leqslant\epsilon$$

$$\iff N\geqslant\frac{64SH\overline{\mathcal{G}}\ln^2\frac{2S\overline{\mathcal{G}}}{\delta}}{\epsilon^2}+\frac{64SH\overline{\mathcal{G}}(A-1)\ln\frac{2S\overline{\mathcal{G}}}{\delta}}{\epsilon^2}\ln\left(\frac{eN}{A-1}+e\right).$$

Thanks to Lemma J.3 of Lazzati et al. (2024), we know that this inequality is satisfied with:

$$N\leqslant\frac{128SH\overline{\mathcal{G}}\ln^2\frac{2S\overline{\mathcal{G}}}{\delta}}{\epsilon^2}+\frac{192SH\overline{\mathcal{G}}(A-1)\ln\frac{2S\overline{\mathcal{G}}}{\delta}}{\epsilon^2}\ln\left(\frac{128eSH\overline{\mathcal{G}}\ln\frac{2S\overline{\mathcal{G}}}{\delta}}{\epsilon^2}\right)+A-1.$$

Rearranging and applying a final union bound concludes the proof. $\qquad\square$

### C.4.3 Proof of Theorem C.8

**Theorem C.8.** *Let $\epsilon\in(0,H]$ and $\delta\in(0,1)$. Let $\mathcal{M}$ be any MDP\R, $\mathcal{R}=\{r^1,\ldots,r^d\}$ any set of $d\geqslant 1$ rewards containing the unknown $r^E$, and $\pi^E\in\Pi^{NM}$ be any policy. Assume that the optimization problem in Line 2 is solved exactly. Then, choosing $\theta=\epsilon/(4H)$, with probability $1-\delta$, the policy $\widehat{\pi}$ output by Algorithm 4 satisfies $\mathcal{W}(\eta_{r^E}^{\pi^E},\eta_{r^E}^{\widehat{\pi}})\leqslant\epsilon$, with:*

$$N\leqslant\mathcal{O}\left(\frac{H^2}{\epsilon^2}\ln\frac{d}{\delta}\right).\tag{23}$$

*Proof.* Observe that, in the same way as in the proof of Theorem 4.4 or 5.1, it is simple to see that:

$$\max_{r\in\mathcal{R}}\mathcal{W}\left(\eta_r^{\pi^E},\widehat{\eta}_r\right)\leqslant H\theta/2+H\epsilon',$$

with probability $1-\delta$, by using:

$$N\leqslant\frac{1}{2(\epsilon')^2}\ln\frac{2d}{\delta},$$

data, where we made a union bound over the $d$ rewards in $\mathcal{R}^\theta$. Then, conditioning on this event and proceeding as in the proof of Theorem 5.2, we have:

$$\begin{aligned}\mathcal{W}\left(\eta_{r^E}^{\pi^E},\eta_{r^E}^{\widehat{\pi}}\right)&\leqslant\max_{r\in\mathcal{R}}\mathcal{W}\left(\eta_r^{\pi^E},\eta_r^{\widehat{\pi}}\right)\\&\overset{(1)}{\leqslant}\max_{r\in\mathcal{R}}\mathcal{W}\left(\eta_r^{\pi^E},\widehat{\eta}_r\right)+\max_{r\in\mathcal{R}}\mathcal{W}\left(\widehat{\eta}_r,\eta_{r_\theta}^{\widehat{\pi}}\right)+\max_{r\in\mathcal{R}}\mathcal{W}\left(\eta_{r_\theta}^{\widehat{\pi}},\eta_r^{\widehat{\pi}}\right)\\&\overset{(2)}{\leqslant}H\epsilon'+H\theta+\max_{r\in\mathcal{R}}\mathcal{W}\left(\widehat{\eta}_r,\eta_{r_\theta}^{\widehat{\pi}}\right)\\&\overset{(3)}{=}H\epsilon'+H\theta+\min_{\pi\in\Pi(\mathcal{R}^\theta)}\max_{r\in\mathcal{R}}\mathcal{W}\left(\widehat{\eta}_r,\eta_{r_\theta}^{\pi}\right)\\&\overset{(4)}{\leqslant}H\epsilon'+H\theta+\max_{r\in\mathcal{R}}\mathcal{W}\left(\widehat{\eta}_r,\eta_{r_\theta}^{\pi^E}\right)\\&\overset{(5)}{\leqslant}2H\epsilon'+2H\theta,\end{aligned}$$

where at (1) we use triangle's inequality, at (2) we use that event above holds and Lemma C.1, at (3) we use the definition of $\widehat{\pi}$, at (4) we upper bound the minimum with a specific choice of reward in $\Pi^{NM}$, i.e., $\pi^E$, and finally, at (5), we apply again that event holds and Lemma C.1.

If we now choose $\theta=\epsilon/(4H)$, and $\epsilon'=\epsilon/(4H)$, we get that, with probability $1-\delta$, the claim of the theorem holds with data:

$$N\leqslant\frac{8H^2}{\epsilon^2}\ln\frac{2d}{\delta}.$$

$\qquad\square$

## C.5 WHEN $r^E$ IS LINEAR IN A KNOWN FEATURE MAP

In this appendix, we consider a variant of the known-reward setting, in which $r^E$ is unknown, but we have knowledge of a $d-$dimensional feature map $\phi : \mathcal{S} \times \mathcal{A} \to [-1, +1]^d$ such that the expert reward $r^E$ can be written as:

$$r_h^E(s, a) = \langle \phi(s, a), w_h \rangle = \sum_{i \in [\![d]\!]} \phi_i(s, a) w_{h,i},$$

for some unknown vectors $w_h \in [-1, +1]^d$. Note that the linear reward setting is common in the IL literature (Abbeel & Ng, 2004). We consider the following robust variant of RDM for this setting:

$$\widehat{\pi} \in \arg\min_{\pi \in \Pi^{\mathrm{NM}}} \max_{w:[\![H]\!] \to [-1, +1]^d} \mathcal{W}\left(\eta_{\phi w}^\pi, \eta_{\phi w}^{\pi^E}\right), \qquad (27)$$

where notation $\phi w$ denotes the reward obtained through the dot product between $\phi$ and $w$.

We now sketch how this setting can be easily addressed through the technique presented in Appendix C.4.

First, consider each *known* feature $\phi^i : \mathcal{S} \times \mathcal{A} \to [-1, +1]$ as a reward function, and define the set of "rewards" $\Phi := \{\phi^1, \dots, \phi^d\}$. Then, consider the set of policies $\Pi(\Phi)$ using definition in Appendix C.4, and let $\pi_\Phi$ be the policy defined as in Eq. (21). Then, from Lemma C.5,[8] we have the guarantee that:

$$\mathbb{P}^{\pi_\Phi}\left(\sum_{h=1}^H \phi^i(s_h, a_h) = g\right) = \mathbb{P}^{\pi^E}\left(\sum_{h=1}^H \phi^i(s_h, a_h) = g\right) \qquad \forall g \in [-1, +1], \forall i \in [\![d]\!].$$

As a consequence, we have that, for any $w_h \in [-1, +1]^d$, it holds that:

$$\eta_{\phi w}^{\pi_\Phi}(g) = \eta_{\phi w}^{\pi^E}(g) \qquad \forall g,$$

since the cumulative feature map collected is the same. Simply put, this means that set $\Pi(\Phi)$ (and also policy $\pi_\Phi$) suffice for this "linear" variant of the RDM problem. However, as the feature map is arbitrary, $\Pi(\Phi)$ might be too large. Thus, we may want to discretize.

Extend the discretization approach of Section 4.1 to "rewards" in $[-1, +1]$ and define set $\Phi^\theta := \{\phi_\theta^1, \dots, \phi_\theta^d\}$. Then, observe that the policy $\pi_{\Phi^\theta}$ satisfies a variant of Lemma C.6:

$$\max_{w:[\![H]\!] \to [-1, +1]^d} \mathcal{W}\left(\eta_{\phi w}^{\pi_{\Phi^\theta}}, \eta_{\phi w}^{\pi^E}\right)$$

$$\overset{(1)}{\leqslant} \max_{w:[\![H]\!] \to [-1, +1]^d} \left(\mathcal{W}\left(\eta_{\phi w}^{\pi_{\Phi^\theta}}, \eta_{\phi_\theta w}^{\pi_{\Phi^\theta}}\right) + \mathcal{W}\left(\eta_{\phi_\theta w}^{\pi_{\Phi^\theta}}, \eta_{\phi_\theta w}^{\pi^E}\right) + \mathcal{W}\left(\eta_{\phi_\theta w}^{\pi^E}, \eta_{\phi w}^{\pi^E}\right)\right)$$

$$\overset{(2)}{\leqslant} 2dH^2\theta + \max_{w:[\![H]\!] \to [-1, +1]^d} \mathcal{W}\left(\eta_{\phi_\theta w}^{\pi_{\Phi^\theta}}, \eta_{\phi_\theta w}^{\pi^E}\right)$$

$$\overset{(3)}{=} 2dH^2\theta,$$

where at (1) we use triangle's inequality and denote $\phi_\theta$ the feature map which is discretized in each dimension, at (2) we apply Lemma C.1 twice[9] and observe that, using Holder's inequality and the definition of discretization: $\max_w \|\phi w - \phi_\theta w\|_\infty \leqslant \max_\phi \|\phi - \phi_\theta\|_1 \leqslant d \max_\phi \|\phi - \phi_\theta\|_\infty \leqslant dH\theta/2$, and at (3) we use the aforementioned property (i.e., Lemma C.5).

Therefore, the policy $\pi_{\Phi^\theta}$, and so the set $\Pi(\Phi^\theta)$, suffice for this new robust RDM problem. It is immediate to extend **RS-BC** and **RS-KT** to this setting by simply extending Algorithms 3 and 4 using input rewards taking values in $[-1, +1]$, and we are done. Then, by adjusting Theorems C.7 and C.8 to keep track of this small variation, we can have also theoretical guarantees. Specifically, for the number of samples in Eqs. (22) and (23), we can guarantee that the policy output by the newly constructed algorithms has a return distribution close to that of $\eta_{\phi w}^{\pi_{\Phi^\theta}}$ for any $w$, which in turn has a return distribution close to that of the expert for any $w$ as shown above. To do this for the variant of **RS-BC**, note that we just need to extend the proof of Lemma C.10. The crucial insight in doing so is that both $\pi_{\Phi^\theta}$ and our estimate $\widehat{\pi}$ play actions with same probability at all trajectories with the same cumulative discretized features. Regarding the variant of **RS-KT**, the proof is immediate as we just need a union bound over all the features.

---

[8]Modulo some slight difference as rewards are in $[0, 1]$ but features are in $[-1, +1]$.

[9]We upper bound with an additional factor of 2 to keep into account that now rewards are in $[-1, +1]$.

## C.6 GENERALIZATION TO ARBITRARY PROBLEMS

In this appendix, we sketch how to extend **RS-BC** and its analysis to arbitrary IL problems of the following kind, in which we aim to find a policy $\hat{\pi}$ that minimizes:

$$\hat{\pi} \in \arg\min_{\pi \in \Pi^{\mathrm{NM}}} \sum_{h \in [\![H]\!]} \sum_{x \in \mathcal{X}_h} \left| \mathbb{P}^{\pi}(\omega_h \in x) - \mathbb{P}^{\pi^E}(\omega_h \in x) \right|,$$

where $\mathcal{X} = \{\mathcal{X}_h\}_h$ is any partition of the set of trajectories satisfying a certain property, specifically:

$$\bigcup_{x \in \mathcal{X}_h} x = \Omega_{h+1} \qquad \forall h \in [\![H]\!],$$

$$x \cap x' = \{\} \qquad \forall x, x' \in \mathcal{X}_h, \forall h \in [\![H]\!],$$

$$\forall x \in \mathcal{X}_h, \ \forall (s,a) \in \mathcal{S} \times \mathcal{A}, \ \exists x' \in \mathcal{X}_{h+1}, \ \forall \omega \in x : \ \omega \cdot s \cdot a \in x' \ \forall h \in [\![H-1]\!],$$

where $\omega \cdot s \cdot a$ denotes the trajectory obtained by concatenating $\omega$ with $s, a$.

First, define:

$$\pi_{\mathcal{X}}(a|s,\omega) := \frac{\mathbb{P}^{\pi^E}\left(s_h = s, \ a_h = a, \ \omega_{h-1} \in \mathcal{X}_{h-1}(\omega)\right)}{\mathbb{P}^{\pi^E}\left(s_h = s, \ \omega_{h-1} \in \mathcal{X}_{h-1}(\omega)\right)}, \tag{28}$$

when the denominator is not 0, and $1/A$ otherwise, and $\mathcal{X}_{h'}(\cdot)$ denotes the set of trajectories to which $\cdot$ belongs. Then, this policy satisfies:

**Theorem C.12.** *It holds that:*

$$\mathbb{P}^{\pi_{\mathcal{X}}}\left(s_h = s \wedge \omega_{h-1} \in x\right) = \mathbb{P}^{\pi^E}\left(s_h = s \wedge \omega_{h-1} \in x\right) \qquad \forall h \in [\![H+1]\!], s \in \mathcal{S}, x \in \mathcal{X}_{h-1}.$$

*Proof.* The proof follows that of Lemma C.2 and C.9. Specifically, we prove it by induction. At $h = 1$, note that it trivially holds as $\mathcal{X}_0 = \varnothing$. Now, make the induction hypothesis that at any $h' < h$, we have:

$$\mathbb{P}^{\pi_{\mathcal{X}}}\left(s_{h'} = s \wedge \omega_{h'-1} \in x\right) = \mathbb{P}^{\pi^E}\left(s_{h'} = s \wedge \omega_{h'-1} \in x\right) \qquad s \in \mathcal{S}, x \in \mathcal{X}_{h'-1}.$$

Then, at $h$, for any $s' \in \mathcal{S}$ and $x' \in \mathcal{X}_{h-1}$, we can write:

$$\mathbb{P}^{\pi_{\mathcal{X}}}\left(s_h = s' \wedge \omega_{h-1} \in x'\right) = \sum_{s,a,\omega} \mathbb{1}\{\omega \cdot s \cdot a \in x'\}$$

$$\mathbb{P}^{\pi_{\mathcal{X}}}\left(s_h = s' \wedge s_{h-1} = s \wedge a_{h-1} = a \wedge \omega_{h-2} = \omega\right)$$

$$= \sum_{x \in \mathcal{X}_{h-1}} \sum_{\omega \in x} \sum_{s,a} \mathbb{1}\{\omega \cdot s \cdot a \in x'\}$$

$$p_h(s'|s,a)\pi_{\mathcal{X}}(a|s,\omega)\mathbb{P}^{\pi_{\mathcal{X}}}\left(s_{h-1} = s \wedge \omega_{h-2} = \omega\right)$$

$$\overset{(1)}{=} \sum_{x \in \mathcal{X}_{h-1}} \sum_{s,a} \mathbb{1}\{x \cdot s \cdot a \in x'\} p_h(s'|s,a)\pi_{\mathcal{X}}(a|s,x)$$

$$\sum_{\omega \in x} \mathbb{P}^{\pi_{\mathcal{X}}}\left(s_{h-1} = s \wedge \omega_{h-2} = \omega\right)$$

$$= \sum_{x \in \mathcal{X}_{h-1}} \sum_{s,a} \mathbb{1}\{x \cdot s \cdot a \in x'\} p_h(s'|s,a)\pi_{\mathcal{X}}(a|s,x)$$

$$\mathbb{P}^{\pi_{\mathcal{X}}}\left(s_{h-1} = s \wedge \omega_{h-2} \in x\right)$$

$$\overset{(2)}{=} \sum_{x \in \mathcal{X}_{h-1}} \sum_{s,a} \mathbb{1}\{x \cdot s \cdot a \in x'\} p_h(s'|s,a)\pi_{\mathcal{X}}(a|s,x)$$

$$\mathbb{P}^{\pi^E}\left(s_{h-1} = s \wedge \omega_{h-2} \in x\right)$$

$$\overset{(3)}{=} \sum_{x \in \mathcal{X}_{h-1}} \sum_{s,a} \mathbb{1}\{x \cdot s \cdot a \in x'\} p_h(s'|s,a)$$

$$\mathbb{P}^{\pi^E}\left(s_{h-1} = s \wedge a_{h-1} = a \wedge \omega_{h-2} \in x\right)$$

$$= \mathbb{P}^{\pi^E}(s_h = s' \wedge \omega_{h-1} \in x'),$$

where at (1) we use notation $x \cdot s \cdot a$ to denote that all the trajectories in $x$ when combined to a given $s, a$ give birth to the same $x'$ by hypothesis, at (2) we use the induction hypothesis, and at (3) the definition of $\pi_\mathcal{X}$. $\qquad\square$

Then, **RS-BC** can be easily extended by counting the number of occurrences in each set of trajectories, and also the theoretical guarantees can be easily extended to this setting.

# D    ADDITIONAL RESULTS AND PROOFS FOR SECTION 5

**Theorem 5.1.** *Let $\epsilon \in (0, H]$ and $\delta \in (0, 1)$. Let $\mathcal{M}_{r^E}$ be any MDP and $\pi^E \in \Pi^{NM}$ any policy. Then, choosing $\theta = \epsilon/(2H)$, a number of samples*

$$N \leqslant \widetilde{\mathcal{O}}\left( \frac{SAH^3}{\epsilon^2} \ln \frac{1}{\delta} \right), \tag{11}$$

*suffices to guarantee that, with probability at least $1 - \delta$, for the estimator $\widehat{\eta}_r(g) := \frac{1}{N} \sum_{\omega \in \mathcal{D}^E} \mathbb{1}\{G(\omega; r_\theta) = g\} \, \forall g, r$, we have:*

$$\max_{r:\mathcal{S} \times \mathcal{A} \times [\![H]\!] \to [0,1]} \mathcal{W}\left( \eta_r^{\pi^E}, \widehat{\eta}_r \right) \leqslant \epsilon.$$

*Proof.* For any reward $r : \mathcal{S} \times \mathcal{A} \times [\![H]\!] \to [0, 1]$, we can write:

$$\mathcal{W}\left( \eta_r^{\pi^E}, \widehat{\eta}_r \right) \overset{(1)}{\leqslant} \mathcal{W}\left( \eta_r^{\pi^E}, \eta_{r_\theta}^{\pi^E} \right) + \mathcal{W}\left( \eta_{r_\theta}^{\pi^E}, \widehat{\eta}_r \right)$$

$$\overset{(2)}{\leqslant} \frac{H\theta}{2} + \mathcal{W}\left( \eta_{r_\theta}^{\pi^E}, \widehat{\eta}_r \right)$$

$$\overset{(3)}{\leqslant} \frac{H\theta}{2} + H\epsilon',$$

where at (1) we use triangle's inequality, at (2) we apply Lemma C.1, and at (3) we use the same derivation as in the proof of Theorem 4.4, after having noticed that the estimate $\widehat{\eta}_r$ is the same estimate used in Line 1 of **RS-KT**. So, by the DKW inequality, we have that the last passage holds with probability $1 - \delta$ using a number of samples:

$$N \leqslant \frac{1}{2(\epsilon')^2} \ln \frac{2}{\delta}.$$

This holds for a single reward $r : \mathcal{S} \times \mathcal{A} \times [\![H]\!] \to [0, 1]$. To make this hold for any possible reward, observe that it suffices to guarantee that it holds for all the rewards in the set $\mathcal{R}$, defined as the set of all reward functions taking on discretized values:

$$\mathcal{R} := \left\{ r : \mathcal{S} \times \mathcal{A} \times [\![H]\!] \to \mathcal{Y}_2^\theta \right\}.$$

Indeed, $\mathcal{R}$ represents an $H\theta/2$-covering of the set of all the real-valued reward functions. Therefore, the result follows through the application of a union bound over all the rewards in $\mathcal{R}$. Since they are $|\mathcal{Y}_2^\theta|^{SAH}$, then we obtain a number of samples:

$$N \leqslant \frac{2SAH^3}{\epsilon^2} \ln \frac{2|\mathcal{Y}_2^\theta|}{\delta} \leqslant \widetilde{\mathcal{O}}\left( \frac{SAH^3}{\epsilon^2} \ln \frac{1}{\delta} \right),$$

to guarantee that $\mathcal{W}\left( \eta_r^{\pi^E}, \widehat{\eta}_r \right) \leqslant \epsilon$ for all $r$, where we set $\epsilon' = \epsilon/(2H)$, $\theta = \epsilon/H$, and $|\mathcal{Y}_2^\theta| \leqslant 1 + 1/\theta = 1 + H/\epsilon$. $\qquad\square$

**Theorem 5.2.** *Under the conditions of Theorem 5.1, assume access to a computational oracle that takes as input the dataset $\mathcal{D}^E$ and the transition model $p$, and outputs a solution to*

$$\widehat{\pi} \in \arg\min_{\pi \in \Pi^{NM}} \max_{r:\mathcal{S} \times \mathcal{A} \times [\![H]\!] \to [0,1]} \mathcal{W}\left( \eta_r^\pi, \widehat{\eta}_r \right). \tag{12}$$

*Then, with probability at least $1 - \delta$, using the number of samples in Eq. (11), it holds that*

$$\max_{r:\mathcal{S} \times \mathcal{A} \times [\![H]\!] \to [0,1]} \mathcal{W}\left( \eta_r^{\pi^E}, \eta_r^{\widehat{\pi}} \right) \leqslant 2\epsilon.$$

*Proof.* Define the good event $\mathcal{E}$ as:

$$\mathcal{E} := \left\{ \max_{r:\mathcal{S}\times\mathcal{A}\times[\![H]\!]\to[0,1]} \mathcal{W}\left(\eta_r^{\pi^E}, \widehat{\eta}_r\right) \leqslant \epsilon \right\}.$$

Then, under $\mathcal{E}$, it holds that:

$$
\max_{r:\mathcal{S}\times\mathcal{A}\times[\![H]\!]\to[0,1]} \mathcal{W}\left(\eta_r^{\pi^E}, \eta_r^{\widehat{\pi}}\right) \overset{(1)}{\leqslant} \max_{r:\mathcal{S}\times\mathcal{A}\times[\![H]\!]\to[0,1]} \mathcal{W}\left(\eta_r^{\pi^E}, \widehat{\eta}_r\right) + \max_{r:\mathcal{S}\times\mathcal{A}\times[\![H]\!]\to[0,1]} \mathcal{W}\left(\widehat{\eta}_r, \eta_r^{\widehat{\pi}}\right)
$$

$$
\overset{(2)}{\leqslant} \epsilon + \max_{r:\mathcal{S}\times\mathcal{A}\times[\![H]\!]\to[0,1]} \mathcal{W}\left(\widehat{\eta}_r, \eta_r^{\widehat{\pi}}\right)
$$

$$
\overset{(3)}{\leqslant} \epsilon + \min_{\pi\in\Pi^{\mathrm{NM}}} \max_{r:\mathcal{S}\times\mathcal{A}\times[\![H]\!]\to[0,1]} \mathcal{W}\left(\widehat{\eta}_r, \eta_r^{\pi}\right)
$$

$$
\overset{(4)}{\leqslant} \epsilon + \max_{r:\mathcal{S}\times\mathcal{A}\times[\![H]\!]\to[0,1]} \mathcal{W}\left(\widehat{\eta}_r, \eta_r^{\pi^E}\right)
$$

$$
\overset{(5)}{\leqslant} 2\epsilon,
$$

where at (1) we use triangle's inequality, at (2) we use that event $\mathcal{E}$ holds, at (3) we use the definition of $\widehat{\pi}$, at (4) we upper bound the minimum with a specific choice of reward in $\Pi^{\mathrm{NM}}$, i.e., $\pi^E$, and finally, at (5), we apply again that event $\mathcal{E}$ holds. The proof is concluded by applying Theorem 5.1, which shows that, with the samples in Eq. (11), the event $\mathcal{E}$ holds w.p. $1 - \delta$. $\qquad\square$

# E ADDITIONAL DETAILS ON NUMERICAL SIMULATIONS

We describe how we sample an MDP and an expert's policy in the experiments in Appendix E.1. In Appendix E.2, we discuss the (im)possibility of implementing a variant of **RS-KT** that works with Markovian policies, in Appendix E.3 we provide implementation details on MIMIC-MD, in Appendix E.4 we provide a graphical example of expert's return distribution and its estimates from the four algorithms considered in Section 6, and in Appendix E.5 we provide a simulation comparing **RS-BC** with the W-RS-GAIL (Lacotte et al., 2019). Finally, in Appendices E.6 and E.7 we provide additional details and results on the simulations conducted to address, respectively, questions 1 and 2. We mention that all simulations took place in some hours on a AMD Ryzen 5 5500U processor.

## E.1 DETAILS ON SAMPLING THE MDPS AND THE EXPERT'S POLICIES

All the MDPs are sampled as follows. The initial state $s_0 \in \mathcal{S}$ is sampled uniformly at random from $\mathcal{S}$. The reward function $r^E : \mathcal{S} \times \mathcal{A} \times [\![H]\!] \to [0,1]$ is sampled, in each $s, a, h$, uniformly at random from a set $\{0, \rho, 2\rho, \dots, \lfloor 1/\rho \rfloor \rho\}$, whose values are controlled by a parameter $\rho \in (0, 1]$ (intuitively, the difference $\theta - \rho$ gives insights into the approximation error). The transition model $p$ is obtained in two steps. First, we sample it uniformly at random in each $s, a, h$ from the simplex $\Delta^{\mathcal{S}}$ (by sampling from the Dirichlet distribution), but then we make the transition of each $s, a, h$ deterministic with probability 0.7 to increase the variety of the MDP.

Regarding the expert's policy, when we sample Markovian expert's policies, we simply sample them uniformly from the simplex $\Delta^{\mathcal{A}}$ in every $\mathcal{S} \times [\![H]\!]$. Instead, when we mention "non-Markovian" expert's policies, then, of course, we cannot sample them uniformly at random from $\Pi^{\mathrm{NM}}$ due to the curse of dimensionality. Instead, what we do is sample randomly from a parameteric subset of $\Pi^{\mathrm{NM}}$ defined as follows. We map states $\{s_1, \dots, s_S\} = \mathcal{S}$ to $\{0, \dots, S-1\}$, and actions $\{a_1, \dots, a_A\} = \mathcal{A}$ to $\{0, \dots, A-1\}$. Then, each policy projects each past history $(s_1, a_1, s_2, a_2, \dots, s_h, a_h)$ to $\mathbb{R}^{16}$ after having mapped it to integers and padded with zeros to reach size $2H$, by using a projection matrix of size $(2H, 16)$ randomly generated. Then, we obtain the probabilities of playing each action by multiplying such 16-dimensional vector by a matrix of weigths of size $(16, A)$ corresponding to the current state $s_h$ (we randomly generated one of these weight matrices for every state).

## E.2 ON ADDRESSING RETURN DISTRIBUTION MATCHING WITH MARKOVIAN POLICIES

We mention that finding the best Markovian policy $\pi' \in \Pi^{\mathrm{M}}$ with return distribution closest to a given array $\eta$, i.e., addressing $\min_{\pi\in\Pi^{\mathrm{M}}} \mathcal{W}(\eta_{r_\theta}^{\pi^E}, \eta)$, seems a problem that cannot be formulated as

an LP and not even as a more general convex optimization problem, as it seems to require bilinear constraints. If so, generating random MDPs and (non-Markovian) expert policies to understand how much Markovian policies are outperformed by non-Markovian policies for return distribution matching may be difficult. We believe that understanding more in-depth this fact will be interesting for future works.

### E.3 IMPLEMENTATION DETAILS OF MIMIC-MD

Algorithm MIMIC-MD has been devised by Rajaraman et al. (2020) but an efficient LP formulation is provided in Rajaraman et al. (2021) (see its proof of Theorem 2). Although it concerns only deterministic expert's policies, we extend it to stochastic policies as in the full algorithm as follows. Simply, in addition to the constraints for $d$ being a valid occupancy measure, we replace constraint $d_h(s,a) = \mathbb{1}\{a = \pi_h^E(s)\}$ (valid for deterministic policies) with $d_h(s,a) = \frac{N_h(s,a)}{N_h(s)} \sum_{a'} d_h(s,a')$ where $N_h(s) \neq 0$ to extend to stochastic experts. Note that it is linear in $d$.

### E.4 AN EXAMPLE OF EXPERT'S RETURN DISTRIBUTION AND ITS ESTIMATES

In Figure 3, we plot the (discretized) return distribution of the expert's policy $\eta_{r^E}^{\pi^E}$ (in green, $\eta^E$) obtained after having sampled at random an MDP with size $S, A, H = (3, 2, 8)$ and the policy as well. We discretized it with bins at a distance of $0.5$ for computation and plotting.

Then, we have sampled a dataset $\mathcal{D}^E$ of $N = 300$ trajectories from $\pi^E$, and given it (and potentially the transition model) in input to the four algorithms considered in Section 6, i.e., **RS-BC**, **RS-KT**, BC and MIMIC-MD, obtaining the return distributions $\eta^{\text{RS-BC}}$, $\eta^{\text{RS-KT}}$, $\eta^{\text{BC}}$ and $\eta^{\text{MIMIC-MD}}$.

Observe that the return distributions of both **RS-BC** and **RS-KT** are close to $\eta^E$. Instead, BC and MIMIC-MD, by working with Markovian policies, are biased, and their return distributions are not that close to $\eta^E$.

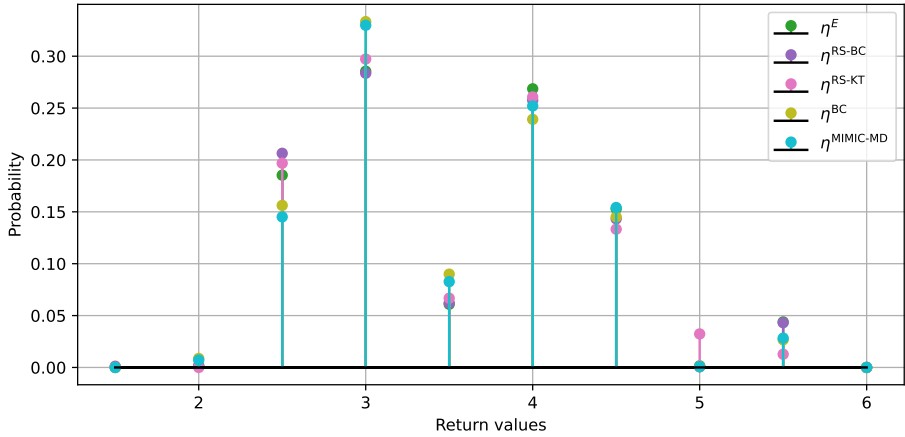

Figure 3: An example of expert's return distribution and various estimates computed with **RS-BC**, **RS-KT**, BC and MIMIC-MD.

### E.5 COMPARISON WITH LACOTTE ET AL. (2019)

The algorithm W-RS-GAIL (Lacotte et al., 2019) aims to match both the expectation and the CVaR at some given level $\alpha$ of the return distribution. Crucially, it addresses this problem using Markovian and stationary policies, and function approximation. To make the comparison with **RS-BC** fairer, we implement W-RS-GAIL without function approximation, by using $SA$ "parameters". In addition, note that **RS-BC** does not use dynamics information, i.e., it is purely offline, while W-RS-GAIL can interact with the environment. At the same time, however, **RS-BC** has access to the reward of the expert, while W-RS-GAIL does not.

|  | $N = 100$ | $N = 1000$ |
|---|---|---|
| **RS-BC** | **0.045±0.022** | **0.025 ± 0.017** |
| `W-RS-GAIL, `$\alpha = 0.3$ | 0.226 ± 0.143 | 0.22 ± 0.147 |
| `W-RS-GAIL, `$\alpha = 0.7$ | 0.202 ± 0.122 | 0.197 ± 0.123 |

Table 2: Results of simulation in Appendix E.5.

|  | $N = 20$ | $N = 80$ | $N = 300$ | $N = 1000$ | $N = 10000$ |
|---|---|---|---|---|---|
| **RS-BC** | **0.101±0.041** | **0.059±0.017** | **0.032±0.011** | **0.016±0.006** | **0.005±0.002** |
| **RS-KT** | 0.164±0.04 | 0.084±0.021 | 0.052±0.013 | 0.033±0.01 | 0.02±0.006 |
| BC | **0.104±0.041** | **0.058±0.018** | **0.035±0.012** | 0.024±0.01 | 0.018±0.009 |
| MIMIC-MD | 0.139±0.058 | 0.079±0.028 | 0.045±0.016 | 0.029±0.012 | 0.019±0.009 |

Table 3: Results of simulation with $S, A, H = (50, 5, 5)$ for Q1.

|  | $N = 20$ | $N = 80$ | $N = 300$ | $N = 1000$ | $N = 10000$ |
|---|---|---|---|---|---|
| **RS-BC** | **0.193±0.086** | **0.087±0.035** | **0.046±0.019** | **0.027±0.01** | **0.012±0.005** |
| **RS-KT** | 0.223±0.066 | 0.115±0.034 | 0.072±0.023 | 0.053±0.017 | 0.041±0.018 |
| BC | 0.208±0.08 | 0.162±0.083 | 0.156±0.086 | 0.151±0.086 | 0.151±0.085 |
| MIMIC-MD | 0.265±0.106 | 0.18±0.086 | 0.159±0.084 | 0.153±0.087 | 0.15±0.085 |

Table 4: Results of simulation with $S, A, H = (2, 2, 20)$ for Q1.

Specifically, we model both the cost function and the policy to learn $\pi$ by using a parameter $w_{s,a}$ and $\theta_{s,a}$ for every possible $(s, a) \in \mathcal{S} \times \mathcal{A}$. For the policy, we compute the probability using a softmax: $\pi_\theta(a|s) = \frac{e^{\theta_{s,a}}}{\sum_{a'} e^{\theta_{s,a'}}}$, so as to avoid the need for normalization after every gradient step. Note that the gradient in $\theta_{\overline{s},\overline{a}}$ of the log policy is:[10]

$$\frac{d\pi_\theta(a|s)}{d\theta_{\overline{s},\overline{a}}} = \frac{d\frac{e^{\theta_{s,a}}}{\sum_{a'} e^{\theta_{s,a'}}}}{d\theta_{\overline{s},\overline{a}}} = \mathbb{1}\{s = \overline{s}\}\left(\mathbb{1}\{a = \overline{a}\} - \frac{e^{\theta_{\overline{s},\overline{a}}}}{\sum_{a'} e^{\theta_{s,a'}}}\right).$$

For the sake of simplicity (and for avoiding extending results from the discounted infinite-horizon setting to ours), we drop the causal entropy term. Moreover, we implement the updates of both the cost function and the policy parameters through gradient ascent/descent, avoiding the usage of Adam and a KL-constrained gradient descent step w.r.t. a linear approximation of the objective.

We generate at random 50 MDPs with $S = 2, A = 2, H = 5$ and 50 non-Markovian stochastic policies. For each environment, for each number of trajectories in $N \in \{100, 1000\}$, we generate 2 datasets of $N$ expert's trajectories. We then run **RS-BC** and two different versions of `W-RS-GAIL` on the datasets; estimate the return distributions, compute the Wasserstein distance and finally compute the average distance among the environments and the 2 datasets. The results are reported in Table 2.

The two versions of `W-RS-GAIL` considered differ for the level $\alpha$ of the CVaR desired, where one uses $\alpha = 0.3$, and the other $\alpha = 0.7$. Both share the values of other hyperparameters: we set $\lambda = 2$ (to balance the CVaR and the mean objectives), learning rate 0.0005 for both the cost and the policy parameters, 3000 number of iterations, and 500 trajectories collected at each training loop.

By looking at Table 2, we realize that **RS-BC** outpeforms both versions of `W-RS-GAIL`. Moreover, since by increasing the number of expert trajectories from $N = 100$ to 1000 the error of `W-RS-GAIL` does not decrease, we observe that it is not a matter of sample efficiency, but of limited expressivity of the policy class adopted by this algorithm (Markovian and stationary policies), as well as the fact that it does not aim to match the whole return distribution, but just the CVaR at a specific level $\alpha$. This limited expressivity is expected (e.g., see Proposition 3.2.)

### E.6 Additional Details on Q1

For simulations, we set $\rho = 0.03 < \theta$, meaning that there is some approximation error.

---

[10]This is used for the update of the policy parameters.

|          | $N = 20$ | $N = 80$ | $N = 300$ | $N = 1000$ | $N = 10000$ |
|----------|----------|----------|-----------|------------|-------------|
| **RS-BC** | **0.081±0.036** | **0.035±0.016** | **0.019±0.011** | **0.011±0.005** | **0.004±0.002** |
| **RS-KT** | 0.095±0.042 | 0.043±0.019 | 0.024±0.012 | **0.013±0.005** | **0.004±0.002** |
| BC | 0.104±0.053 | 0.076±0.048 | 0.068±0.048 | 0.066±0.049 | 0.065±0.049 |
| MIMIC-MD | 0.13±0.063 | 0.085±0.046 | 0.071±0.046 | 0.068±0.049 | 0.065±0.049 |

Table 5: Results of simulation with $S, A, H = (2, 2, 5)$ and $\theta = \rho$ for Q2.

|          | $N = 20$ | $N = 80$ | $N = 300$ | $N = 1000$ | $N = 10000$ |
|----------|----------|----------|-----------|------------|-------------|
| **RS-BC** | **0.088±0.026** | **0.048±0.022** | **0.025±0.011** | **0.012±0.005** | **0.004±0.002** |
| **RS-KT** | 0.135±0.037 | 0.068±0.02 | 0.034±0.01 | 0.019±0.005 | **0.006±0.002** |
| BC | **0.092±0.032** | 0.054±0.029 | 0.038±0.023 | 0.031±0.022 | 0.028±0.024 |
| MIMIC-MD | 0.118±0.042 | 0.066±0.028 | 0.044±0.023 | 0.033±0.021 | 0.029±0.024 |

Table 6: Results of simulation with $S, A, H = (20, 3, 5)$ for Q2.

|          | $N = 20$ | $N = 80$ | $N = 300$ | $N = 1000$ | $N = 10000$ |
|----------|----------|----------|-----------|------------|-------------|
| **RS-BC** | **0.177±0.067** | **0.091±0.038** | **0.047±0.018** | **0.023±0.008** | **0.008±0.003** |
| **RS-KT** | 0.224±0.083 | 0.108±0.039 | 0.057±0.018 | **0.031±0.01** | **0.011±0.004** |
| BC | 0.196±0.104 | 0.159±0.102 | 0.148±0.103 | 0.145±0.104 | 0.144±0.106 |
| MIMIC-MD | 0.246±0.115 | 0.174±0.103 | 0.151±0.103 | 0.145±0.104 | 0.144±0.106 |

Table 7: Results of simulation with $S, A, H = (2, 2, 20)$ for Q2.

Regarding Table 3, we would like to discuss some points. First, we mention that the increase in size of $S$ and $A$ is not sufficiently big to permit to **RS-KT** to outperform the sample complexity of **RS-BC**, as discussed in Q4. Second, increasing $S, A, H$ makes **RS-KT** much more time-consuming, as it requires solving an LP with much more variables and constraints. Third, **RS-KT** performs comparably to BC and MIMIC-MD, but this is due to an increment of approximation error due to the increase of $S, A$, as clear from Table 6, where in absence of approximation error **RS-KT** outperforms BC and MIMIC-MD. Note that this is not the fact for **RS-BC**, which seems more robust to approximation error for this problem size (intuitively, the reason is that it is strictly more expressive than BC for any choice of $\theta$).

Regarding Table 4, we mention that a larger $H$ implies a larger approximation error in particular for **RS-KT**, as clear from Table 7, where in absence of approximation error **RS-KT** outperforms BC and MIMIC-MD.

### E.7 ADDITIONAL DETAILS ON Q2

The three additional simulations have all a non-Markovian expert and $\rho = \theta$ (to enforce no approximation error) with parameters $S, A, H, \theta \in \{(2, 2, 5, 5e-2), (20, 3, 5, 5e-2), (2, 2, 20, 1e-1)\}$, and the results are reported respectively in Tables 5, 6 and 7.

By comparing these tables respectively with Table 1 (top), 3 and 4, where there is approximation error due to $\theta = 0.05 > 0.03 = \rho$, we realize that the approximation error mostly concerns **RS-KT** and with larger horizons $H$ (as expected from Lemma 4.2, and from knowledge of how **RS-BC** works).

