# OpenReview forum: "Imitation Learning as Return Distribution Matching"
_ICLR.cc/2026/Conference — ICLR 2026 Poster_

### Official Review · Reviewer_XjG5 · 2025-10-29

**Soundness:** 3
**Presentation:** 3
**Contribution:** 3
**Rating:** 6
**Confidence:** 3

**Summary:**

The paper addresses the problem of risk-sensitive imitation learning (IL) in tabular finite-horizon Markov Decision Processes (MDPs). Traditional IL focuses on matching the expert's occupancy measure, which only imitates the expected return (average performance) and is inherently risk-neutral, ignoring aspects like variance or tail behavior in the return distribution. The authors reformulate IL as "Return Distribution Matching" (RDM), where the goal is to train an agent that matches the expert's entire return distribution (encoding risk attitude) in Wasserstein distance, rather than just the expectation or limited risk measures like Conditional Value at Risk (CVaR). They consider settings where the expert's reward is known (primary focus) or unknown, and the transition model is either unknown (no-interaction/offline) or known. The aim is to develop provably sample-efficient algorithms that handle non-Markovian experts, who may exhibit complex risk-sensitive behaviors not capturable by Markovian policies.

**Strengths:**

1. The analysis carried out is sound and well-motivated.
2. The paper is polished and easy to read.
3. The problem setting is interesting.
4. The empirical results are encouraging.

**Weaknesses:**

1. Primary focus assumes known r^E, which may not hold in real IL (e.g., human demos); unknown-reward variant relies on oracles, potentially impractical. If the expert’s rewards were known, we could have solved the forward RL, which allows us for the full access to the rich literature: function approximation, generalization, etc. Assumptions: Requires known rewards (or oracle for unknown); no online interaction for learning transitions.
2. There is no compelling empirical evidence presented in the main body that validates whether the correct distribution is being learned (the results presented in the main body only show the overall performance).
3. There is a figure in the appendix that seems to show the learned distribution, however this figure is hard to understand, and can likely be significantly improved. I strongly encourage the authors to improve this figure and include it in the main body.

**Questions:**

1.It is not clear what about Algorithm 1 is ‘risk-sensitive’. It seems to just be learning a policy, without a clear link to the return distribution.
2.The paper could benefit from a broader discussion as to how the proposed methods connect to the distributional RL literature. For example, how does the proposed method relate to a categorical or quantile representation.
3.Since we know the expert's rewards, wouldn't it be better to solve the forward RL with the known reward to completely restore the expert's policy? What are the benefits of the proposed methods over the forward RL?

---

> ### Author Response · Authors · 2025-11-17
>
> We are glad that the Reviewer appreciated the novel problem setting we introduced, as well as found our analysis sound and well-motivated. Below, we respond to the Reviewer's comments and questions.
>
> > Primary focus assumes known r^E, which may not hold in real IL (e.g., human demos); unknown-reward variant relies on oracles, potentially impractical. If the expert’s rewards were known, we could have solved the forward RL, which allows us for the full access to the rich literature: function approximation, generalization, etc. Assumptions: Requires known rewards (or oracle for unknown); no online interaction for learning transitions.
>
> We remark that, even if the expert’s reward is known, we *cannot* solve the problem with forward RL. Indeed, we are considering the *risk-sensitive* problem setting, where maximizing the *expected* return, i.e., constructing risk-neutral agents, is undesirable (e.g., for reasons such as safety; see the recent book [1]). In addition, in the considered problem setting, we cannot even solve the problem with forward *risk-sensitive* RL, because the specific risk objective to optimize is unknown. Existing risk-sensitive RL algorithms know in advance the risk-sensitive objective to optimize (e.g., CVaR, entropic risk measure, etc.), whereas we do not; instead, we aim to "mimic" the risk attitude contained in the expert's demonstrations. In other words, we observe the expert's behavior, and want to imitate its risk attitude. In contrast, in risk-sensitive RL, we do not observe any expert's behavior, but aim to optimize a *given* risk measure. Intuitively, our approach brings a human into the learning loop to avoid the difficulty of designing/choosing the correct risk measure (and its parameters) for the task, which can be quite difficult.
>
> While we recognize that dropping the knowledge of the expert's reward represents an important direction for future work, as we acknowledged also in Section 7, we would like to additionally point out that:
>
> - The "known $r^E$" setting is of independent interest. Indeed, various works in the literature consider problem settings with a known expert reward function, including inverse constrained RL [2] and utility learning [3]. Examples of applications with a known reward mentioned in these works include, e.g., autonomous driving [2].
> - A very simple and important real-world known-reward example is finance [4], where the reward function is represented by money, and the amount of money received after an action is usually known or observed. In this setting, merely training an agent to maximize the *expected* return may be undesirable. Thus, we might prefer to use human expert guidance on which risk-sensitive behavior should be applied. In particular, since the reward function is known, matching the expert's risk-sensitivity can be done through our algorithms (i.e., RS-BC or RS-KT), which successfully exploit the reward information.
>
> Regarding the assumption of a known transition model in RS-KT instead of considering online interaction for learning the transitions, as mentioned in lines 311-313, we note that it might be possible to construct a variant of RS-KT that requires online access to the environment instead of exact knowledge of the transition model, based on the insights of [5]. Specifically, we would first run a reward-free exploration algorithm (e.g., [6]) to construct an estimate of the transition model, and then use this estimate in place of the true $p$ in the LP of RS-KT. We are happy to provide additional details if the Reviewer desires.
>
> > There is no compelling empirical evidence presented in the main body that validates whether the correct distribution is being learned (the results presented in the main body only show the overall performance).
>
> We remark that Table 1 does *not* show the overall performance; rather, it presents the average *distance* between the expert's return distribution and the return distribution of the learned policy. In other words, all the simulations in Section 6 provide empirical evidence validating that the correct distribution is being learned.
>
> > There is a figure in the appendix that seems to show the learned distribution, however this figure is hard to understand, and can likely be significantly improved. I strongly encourage the authors to improve this figure and include it in the main body.
>
> We have updated the submission by improving the figure by removing $\hat{\eta}$, restricted the plot to only points in the support with positive probabilities, used a coarser grid, and adopted a stem plot for better clarity. Moreover, we have added this new figure to the main paper.

---

> > ### Author Response · Authors · 2025-11-17
> >
> > > It is not clear what about Algorithm 1 is ‘risk-sensitive’. It seems to just be learning a policy, without a clear link to the return distribution
> >
> > From a high-level perspective, the key point is that we work with risk-sensitive experts (i.e., they do not maximize the *expected* cumulative reward). As a result, learning standard Markovian policies is not sufficient to match the expert’s risk attitude (as illustrated, for example, in Proposition 3.2). Instead, by learning policies within our novel policy class (which includes more expressive *non-Markovian* policies) our algorithms can accurately capture the expert's risk attitude.
> >
> > More formally, note that any policy induces a return distribution. The standard behavioral cloning (BC) algorithm (e.g., see [7]), outputs a policy whose return distribution has an *expected value* close to the expectation of the return distribution of the expert's policy. Note that matching the expectation is a risk-neutral objective [1]. However, as shown, for instance, in our Proposition 3.2, the "shape" (i.e., aspects beyond the mean) of the return distribution of the policy learned through BC might be quite different from that of the expert. This means that BC (and, similarly, other standard imitation learning algorithms), does *not* match the risk-sensitivity of the expert (informally encoded in the "shape" of its return distribution). Instead, as demonstrated by Theorem 4.3, by leveraging a more expressive policy class, our RS-BC algorithm learns a policy whose induced return distribution matches that of the expert (intuitively, it matches both the expected value and the "shape"), thus properly capturing its risk attitude.
> >
> > > The paper could benefit from a broader discussion as to how the proposed methods connect to the distributional RL literature. For example, how does the proposed method relate to a categorical or quantile representation
> >
> > Based on the notation in the standard reference for distributional RL (book [1], Section 5.5), our RS-KT algorithm at line 1 adopts a fixed-size $|\mathcal{Y}^\theta|$-*categorical* representation for storing its estimate of the expert's return distribution, where the $|\mathcal{Y}^\theta|$ particles lie at the evenly spaced locations $\mathcal{Y}^\theta=\\{0,\theta,2\theta,...,\lfloor H/\theta\rfloor\theta\\}$. Although the objective is different from that of approximating a probability distribution, a discretization based on the set $\mathcal{Y}^\theta$ is also adopted in RS-BC, as explained in Sections 4.1 and 4.2.
> >
> > We have updated the submission by explicitly mentioning this fact in Section 4.3, after Theorem 4.4.
> >
> > > Since we know the expert's rewards, wouldn't it be better to solve the forward RL with the known reward to completely restore the expert's policy? What are the benefits of the proposed methods over the forward RL?
> >
> > The crucial point is that performing forward RL does *not* yield the expert's policy, because the expert is risk-sensitive rather than risk-neutral; that is, the expert devised its own policy based on some *unknown* risk objective, instead of maximizing the expected return. Please, refer to our first comment for additional details.

---

> > > ### Author Response · Authors · 2025-11-17
> > >
> > > [1] Marc G. Bellemare, Will Dabney, and Mark Rowland. Distributional Reinforcement Learning. MIT Press, 2023.
> > >
> > > [2] Shehryar Malik, Usman Anwar, Alireza Aghasi, and Ali Ahmed. Inverse constrained reinforcement learning. In International Conference on Machine Learning 38 (ICML), 2021.
> > >
> > > [3] Filippo Lazzati and Alberto Maria Metelli. Learning utilities from demonstrations in markov decision processes. In International Conference on Machine Learning 42 (ICML), 2025.
> > >
> > > [4] Hans Follmer and Alexander Schied. Stochastic Finance: An Introduction in Discrete Time. De Gruyter, 2016.
> > >
> > > [5] Tian Xu, Ziniu Li, Yang Yu, and Zhi-Quan Luo. Provably efficient adversarial imitation learning with unknown transitions. In Conference on Uncertainty in Artificial Intelligence 39 (UAI), 2023.
> > >
> > > [6] Emilie Kaufmann, Pierre Menard, Omar Darwiche Domingues, Anders Jonsson, Edouard Leurent, and Michal Valko. Adaptive reward-free exploration. In International Conference on Algorithmic Learning Theory 32 (ALT 2021), pp. 865–891, 2021.
> > >
> > > [7] Dylan J. Foster, Adam Block, and Dipendra Misra. Is behavior cloning all you need? understanding horizon in imitation learning. In Advances in Neural Information Processing Systems 37 (NeurIPS), 2024.

---

### Official Review · Reviewer_9FFJ · 2025-10-31

**Soundness:** 3
**Presentation:** 3
**Contribution:** 3
**Rating:** 6
**Confidence:** 3

**Summary:**

The paper presents the risk-sensitive imitation learning problem as matching the expert's return distribution in Wasserstein distance, and designs two types of algorithms, RS-BC (without model, based on behavior cloning non-Markov strategy class) and RS-KT (known transition, constructing LP search occupancy measure to match distribution), for the cases with/without known transition models. It provides theoretical sample complexity bounds (Theorem 4.3/4.4/5.1/5.2) and conducts a large number of simulation experiments.

**Strengths:**

1. Directly promoting risk-sensitive imitation learning to return distribution matching is more comprehensive in depicting "risk attitude" than merely matching expected value or several quantiles, which has both theoretical and intuitive appeal.
2. This paper proposed a  subclass of non-Markovian policies $\Pi(r_\theta)$ that can be stored and computed. The sample complexity bounds for RS-BC (no interaction) and RS-KT (known transition) are provided, with clear analysis and comparison. The theorem gives upper bounds for various scenarios.
3. The experimental design is clear, aiming to answer four specific research questions (Q1-Q4).

**Weaknesses:**

1. This paper proposes the main algorithms (especially RS-BC and RS-KT), which all assume that the expert's reward function $r^E(s,a)$ is known, or at least the reward for each step can be accurately calculated from the expert's trajectory. Section 5 of the paper explicitly points out that this is an important limitation, and it specifically discusses the "unknown $r^E$" case, but this part only stays at the theoretical analysis level (oracle-based) and does not propose an actual runnable algorithm.
2. The theoretical guarantees depend on strong assumptions (e.g., known transitions, exact LP solution), yet the experiments use approximations without quantifying the induced errors, leaving the practical scalability and validity of the theoretical bounds unclear.

**Questions:**

1. The RS-KT algorithm needs to solve an LP with a scale of $\mathcal{O}(SAH|\mathcal{Y}^{\theta}|)$, which seems difficult to scale in practice. Have the authors considered extending it to an approximate method in the continuous domain (for example, using function approximation to replace tabular occupancy measures)?
2. Can the algorithm be extended to an online interactive version, which can learn unknown transitions and unknown expert rewards from the environment. Maybe the author can add more discussion on this approach and the comparison with risk-sensitive RL.

---

> ### Author Response · Authors · 2025-11-17
>
> We thank the Reviewer for recognizing the theoretical appeal of our novel *return distribution matching* setting, as well as the clarity of our analysis. Below, we provide responses to the Reviewer's comments and questions.
>
> > This paper proposes the main algorithms (especially RS-BC and RS-KT), which all assume that the expert's reward function $r^E(s,a)$  is known, or at least the reward for each step can be accurately calculated from the expert's trajectory. Section 5 of the paper explicitly points out that this is an important limitation, and it specifically discusses the "unknown $r^E$" case, but this part only stays at the theoretical analysis level (oracle-based) and does not propose an actual runnable algorithm.
>
> Although we agree with the Reviewer that the development of practical algorithms for the "unknown $r^E$" case represents an interesting direction for future works, as we also mentioned in Section 7, we would like to remark that:
>
> - The "known $r^E$" setting is of independent interest. Indeed, various works in the literature consider problem settings with a known expert reward function, including inverse constrained RL [1] and utility learning [2]. Examples of applications with a known reward mentioned in these works include, e.g., autonomous driving [1].
> - A very simple and important real-world known-reward example is finance [3], where the reward function is represented by money, and the amount of money received after executing an action is usually known or observed. In this setting, merely training an agent to maximize the *expected* return may be undesirable. Thus, we might prefer to use human expert guidance on which risk-sensitive behavior should be applied. In particular, since the reward function is known, matching the expert's risk sensitivity can be achieved through our algorithms (i.e., RS-BC or RS-KT), which successfully exploit the reward information.
> - Designing provably-efficient algorithms for the "unknown $r^E$" setting is certainly not straightforward, and there is the chance that it may not even be possible. Intuitively, the lack of knowledge about $r^E$ suggests that general non-Markovian policies might be necessary for addressing the robust RDM problem, which would introduce a *computational complexity* challenge. On the other hand, our interesting result in Theorem 5.2 shows that the problem can be solved in a *sample efficient* manner using an oracle (when the dynamics are known). Taken together, these observations suggest that the problem may be solvable sample efficiently only at the cost of computational inefficiency, although it does not rule out that a computational and sample efficient algorithm could be developed. We believe that our RS-BC and RS-KT algorithms provide a meanigful starting point for future research on this task.
>
> > The theoretical guarantees depend on strong assumptions (e.g., known transitions, exact LP solution), yet the experiments use approximations without quantifying the induced errors, leaving the practical scalability and validity of the theoretical bounds unclear.
>
> We emphasize that *none* of these assumptions is necessary, as also mentioned in the main paper (see the paragraph after Theorem 4.4). In particular:
>
> - Regarding the knowledge of the transition model in RS-KT: For the unknown-transition setting, we already provide a powerful provably efficient algorithm, i.e., RS-BC. The goal of RS-KT is precisely to demonstrate that, if there is additional knowledge of the transition model, then it is possible to exploit it to achieve a much lower sample complexity compared to the case where the transition model is unknown (and to also show how to exploit this additional information). In addition, as mentioned in lines 311-313, we would like to note that it might be possible to construct a variant of RS-KT that requires online access to the environment instead of exact knowledge of the transition model, based on the insights of [4]. Specifically, we would first run a reward-free exploration algorithm (e.g., [5]) to construct an estimate of the transition model, and then use this estimate in place of the true $p$ in the LP of RS-KT.
> - Regarding the exactness of the LP solution: We remark that this assumption is made only to simplify the presentation of the results in Theorem 4.4, but it is *not necessary*, as explained in lines 313-314. Indeed, if our LP solver outputs a solution with error $\epsilon\_{\text{apx}}$, then the guarantee of Theorem 4.4 can be extended to: $\mathcal{W}(\eta^{\pi^E}\_{r^E}, \eta^{\widehat{\pi}}_{r^E}) \le \epsilon +
> \epsilon\_{\text{apx}}$, where the approximation error due to the LP solver $\epsilon\_{\text{apx}}$ is added to the total error $\epsilon$, which in turn represents the sum of the estimation error and the discretization error.

---

> > ### Author Response · Authors · 2025-11-17
> >
> > > The RS-KT algorithm needs to solve an LP with a scale of $\mathcal{O}(SAH|\mathcal{Y}^\theta|)$, which seems difficult to scale in practice. Have the authors considered extending it to an approximate method in the continuous domain (for example, using function approximation to replace tabular occupancy measures)?
> >
> > RS-KT is an extremely sample-efficient algorithm designed ad-hoc for tabular environments, where it achieves a sample complexity *independent of the cardinality of the state-action space* (see Theorem 4.4). As correctly observed by the Reviewer, this sample efficiency comes to the price of increased computational complexity, specifically that of solving an LP with $\mathcal{O}(SAH|\mathcal{Y}^\theta|)$ optimization variables and constraints. Although there exist various efficient polynomial-time algorithms for solving LPs (e.g., see [6]), we agree that this can become infeasible when the quantities of interest $S,A,H,1/\theta$ are too large or the state space is continuous. However, we conjecture that this (polynomial) "LP complexity" is unavoidable if one wants to achieve the extremely favorable sample complexity of RS-KT.
> >
> > Regarding an extension of RS-KT to the continuous domain, we note that a direct extension is not immediate and would necessarily require various approximations, as we cannot enforce a flow constraint for every possible state, since there are infinitely many. Moreover, as the Reviewer correctly notes, introducing function approximation for modeling the tabular occupancy measures is necessary here, and we additionally note that assuming access to the dynamics $p$ would also be quite unrealistic, as $p$ may in principle be an arbitrary measure. For these reasons, we believe that, in continuous environments, different techniques should be adopted to improve sample efficiency using information about the dynamics of the environment. If this information consists of the ability to collect trajectories by playing arbitrary policies (i.e., online access to the environment), then we believe that a meanigful extension of GAIL [2] might do the trick. Indeed, GAIL aims to find a policy that matches the occupancy measure of the expert's policy by exploiting additional rollouts in the environment. If we augment the state space to keep track of the cumulative reward of trajectories, then an "augmented" version of GAIL would match the distribution over states and returns, which implies matching the return distribution. We would be happy to provide additional details on this idea if the Reviewer desires.
> >
> > > Can the algorithm be extended to an online interactive version, which can learn unknown transitions and unknown expert rewards from the environment. Maybe the author can add more discussion on this approach and the comparison with risk-sensitive RL.
> >
> > The core difference between standard risk-sensitive RL and our setting is, roughly speaking, the knowledge of the risk measure to optimize. Existing risk-sensitive RL algorithms know in advance the risk-sensitive objective to optimize (e.g., CVaR, entropic risk measure, etc.), whereas we do not; instead, we aim to mimic the risk attitude contained in the expert's demonstrations. In other words, we observe the expert's behavior, and want to imitate its risk attitude. In risk-sensitive RL, by contrast, we do not observe any expert behavior, but aim to optimize a *given* risk measure. Intuitively, our approach brings a human into the learning loop to avoid the difficulty of designing/choosing the correct risk measure (and its parameters) for the task.
> >
> > Regarding the extension of RS-KT to an online interactive version: Yes, it is possible. In particular, as mentioned also in a previous comment, one can exploit the insights of [4] to first estimate the transition model by running a reward-free algorithm (e.g., [5]), and then solve the LP using this estimate.

---

> > > ### Author Response · Authors · 2025-11-17
> > >
> > > [1] Shehryar Malik, Usman Anwar, Alireza Aghasi, and Ali Ahmed. Inverse constrained reinforcement learning. In International Conference on Machine Learning 38 (ICML), 2021.
> > >
> > > [2] Filippo Lazzati and Alberto Maria Metelli. Learning utilities from demonstrations in markov decision processes. In International Conference on Machine Learning 42 (ICML), 2025.
> > >
> > > [3] Hans Follmer and Alexander Schied. Stochastic Finance: An Introduction in Discrete Time. De Gruyter, 2016.
> > >
> > > [4] Tian Xu, Ziniu Li, Yang Yu, and Zhi-Quan Luo. Provably efficient adversarial imitation learning with unknown transitions. In Conference on Uncertainty in Artificial Intelligence 39 (UAI), 2023.
> > >
> > > [5] Emilie Kaufmann, Pierre Menard, Omar Darwiche Domingues, Anders Jonsson, Edouard Leurent, and Michal Valko. Adaptive reward-free exploration. In International Conference on Algorithmic Learning Theory 32 (ALT 2021), pp. 865–891, 2021.
> > >
> > > [6] Bertsimas, D., & Tsitsiklis, J.N. Introduction to linear optimization. Athena scientific optimization and computation series. 1997.
> > >
> > > [7] Jonathan Ho and Stefano Ermon. Generative adversarial imitation learning. In Advances in Neural Information Processing Systems 29 (NeurIPS), 2016.

---

### Official Review · Reviewer_sUz6 · 2025-10-31

**Soundness:** 3
**Presentation:** 3
**Contribution:** 3
**Rating:** 8
**Confidence:** 3

**Summary:**

The paper studies the problem of risk-sensitive imitation learning. The goal is to train an risk-sensitive agent that matches the expert’s return rather than merely the expected return. They show that Markovian policies are in general insufficient for this task,  and introduce an efficiently storable subclass of non-Markovian policies based on discretized cumulative reward. They propose  two provably efficient algorithms, RS-BC and RS-KT, when the transition model is unknown and known, respectively for the tabular setting. They prove sample complexity guarantees for both algorithms, show a polynomial-sample result for the unknown-reward robust variant. They also provide simulation evidence that RS-BC and RS-KT better match expert return distributions than standard baselines in a range of synthetic MDPs.

**Strengths:**

- The problem reformulation of risk-sensitive IL is clear and well-motivated. Since return contains for relavant distribution  for risk-sensitive decision-making, matching return distributions can generalize expectation/CVaR matching to general risk-attitude imitation.
- The paper is well organized and the main ideas are presented clearly. The transitions are fluent and follow a logical flow.
- All three algorithms are conceptually simple and intuitive. Both RS-BC and RS-KT are computationally efficient and  come with  sample complexity guarantees. For the known reward setting, the RS-KT algorithm attains a strong sample bound independent of S,A. These sample complexity results with different dependence regimes, e.g., with and without transition knowledge are novel and interesting.
- The paper provides empirical validations across many random MDPs, which shows the proposed methods reduce Wasserstein error relative to standard baselines.

**Weaknesses:**

- One major limitation is the computational cost of RS-KT. It requires solving an LP with size scaling as $O(S A H |Y_\theta|)$. For moderate/large tabular MDPs this can become expensive.
- Unknown-reward setting lacks practical algorithm. Theoretical sample-efficiency for the unknown-reward case is established by using an oracle-based argument, but a concrete practical algorithm for this setting is deferred. This is an important practical case since expert reward unknown.

**Questions:**

questions/comments
-  Could the authors provide an experiment comparing RDM to a strong CVaR-matching IL method across a range of \alpha levels, to show where full distribution matching yields material improvement?
- how to choose \theta in practice is still not very clear
- Lower bounds is missing (the authors have noted this as a limitation)
- There are some small typos and places where exposition could be tightened
- Missing related works, to name a few,  sample complexity/regret guarantees on risk-sensitive RL [1][2], CVaR RL [3][4], risk-sensitive distributional RL [5][6]

[1]Fei, Yingjie, et al. "Risk-sensitive reinforcement learning: Near-optimal risk-sample tradeoff in regret." Advances in Neural Information Processing Systems 33 (2020): 22384-22395.

[2]Fei, Yingjie, et al. "Exponential bellman equation and improved regret bounds for risk-sensitive reinforcement learning." Advances in neural information processing systems 34 (2021): 20436-20446.

[3]Bastani, Osbert, et al. "Regret bounds for risk-sensitive reinforcement learning." Advances in Neural Information Processing Systems 35 (2022): 36259-36269.

[4]Wang, Kaiwen, Nathan Kallus, and Wen Sun. "Near-minimax-optimal risk-sensitive reinforcement learning with cvar." International Conference on Machine Learning. PMLR, 2023

[5]Liang, Hao, and Zhi-Quan Luo. "Bridging distributional and risk-sensitive reinforcement learning with provable regret bounds." Journal of Machine Learning Research 25.221 (2024): 1-56.

[6]Chen, Yu, et al. "Provable risk-sensitive distributional reinforcement learning with general function approximation." arXiv preprint arXiv:2402.18159 (2024).

---

> ### Author Response · Authors · 2025-11-17
>
> We thank the Reviewer for recognizing the importance of the problem formulation, as well as the clarity of the presentation and the novelty and relevance of our algorithms and the corresponding sample complexity bounds. Below, we provide responses to the Reviewer's comments and questions.
>
> > One major limitation is the computational cost of RS-KT. It requires solving an LP with size scaling as $\mathcal{O}(SAH|\mathcal{Y}^\theta|)$. For moderate/large tabular MDPs this can become expensive.
>
> We agree with the Reviewer that the computational complexity of solving the LP in RS-KT can become quite expensive when the quantities of interest $S,A,H,1/\theta$ become large. However, we conjecture that this (polynomial) "LP complexity" is unavoidable if one wants to achieve the small sample complexity of RS-KT stated in Theorem 4.4. We detail our intuition below and leave to future works the task of formally proving a lower bound.
>
> **Intuition**: Our goal is to find a policy whose return distribution is close to that of the expert. Intuitively, it is reasonable that this problem requires at least enough demonstrations to properly estimate the expert's return distribution. Since this task requires precisely the amount of data required by RS-KT (see Eq. 9), and since computing a policy whose return distribution is close to the estimated one amounts to solving the LP in RS-KT, we conjecture that no algorithm can achieve the same sample complexity as RS-KT with lower computational complexity.
>
> > Unknown-reward setting lacks practical algorithm. Theoretical sample-efficiency for the unknown-reward case is established by using an oracle-based argument, but a concrete practical algorithm for this setting is deferred. This is an important practical case since expert reward unknown.
>
> We agree with the Reviewer and would like to note that we already highlighted this point as an interesting direction for future work in Section 7.
>
> That said, we would like to mention that:
>
> - The known-reward setting is of independent interest. Indeed, various works in the literature consider problem settings with a known expert reward function, including inverse constrained RL [7] and utility learning [8]. Examples of applications with a known reward mentioned in these works include, e.g., autonomous driving [7].
> - A very simple and important real-world known-reward example is finance [9], where the reward function is represented by money, and the amount of money received after executing an action is usually known or observed. In this setting, merely training an agent to maximize the *expected* return may be undesirable. Thus, we might prefer to use human expert guidance on which risk-sensitive behavior should be applied. In particular, since the reward function is known, matching the expert's risk sensitivity can be achieved through our algorithms (i.e., RS-BC or RS-KT), which successfully exploit the reward information.

---

> > ### Author Response · Authors · 2025-11-17
> >
> > > Could the authors provide an experiment comparing RDM to a strong CVaR-matching IL method across a range of \alpha levels, to show where full distribution matching yields material improvement?
> >
> > We have implemented the W-RS-GAIL algorithm proposed in [10], which aims to match both the expected return and the CVaR at a given level $\alpha$. We ran it with $\alpha=0.3$ and $\alpha=0.7$, and compared it with RS-BC across 50 environments and two different amounts of expert trajectories ($N\in\\{100,1000\\}$), following the evaluation methodology adopted in Section 6. A detailed description of the implementation is provided in a new Appendix E.5, and we have updated the submission on OpenReview.
> >
> > It should be remarked that, to make the comparison fairer, we implemented W-RS-GAIL without function approximation. We mention also that W-RS-GAIL *can* interact with the environment, while RS-BC cannot, which makes W-RS-GAIL potentially more sample efficient than RS-BC. However, RS-BC has access to the expert's reward information, while W-RS-GAIL does not. The results are as follow:
> >
> > | Method                         | N = 100        | N = 1000        |
> > |-------------------------------|----------------|-----------------|
> > | **RS-BC**                       | **0.045 ± 0.022** | **0.025 ± 0.017** |
> > | W-RS-GAIL (α = 0.3)            | 0.226 ± 0.143  | 0.220 ± 0.147   |
> > | W-RS-GAIL (α = 0.7)            | 0.202 ± 0.122  | 0.197 ± 0.123   |
> >
> > All algorithms output policies whose return distribution is close to that of the expert (the Wasserstein distance ranges from 0 to 5), since the average values of Wasserstein distance are quite small (about 0.04 for RS-BC, and about 0.2 for both versions of W-RS-GAIL). However, RS-BC outperforms both versions of W-RS-GAIL, which perform similarly. This is expected, as matching the CVaR at a specific level does not guarantee to match the entire return distribution. In particular, note that the difference is substantial, and does not decrease when the number of expert trajectories increases. This indicates that the issue is not one of sample efficiency, but of expressivity: the reliance of W-RS-GAIL on Markovian stationary policies, which are not expressive enough to capture the return distribution of arbitrary non-Markovian experts, leads to poorer performance (see also our Proposition 3.2 for a simple example of this phenomenon). In contrast, RS-BC, thanks to its more expressive policy class, can get much closer. Moreover, another reason why W-RS-GAIL performs significantly worse than RS-BC is that this is an algorithm thought to scale to large/continuous problems, and therefore introduces various approximations in its optimization procedure.
> >
> > > how to choose \theta in practice is still not very clear
> >
> > Intuitively, $\theta$ governs the trade-off between the accuracy of our algorithms and their complexity (computational complexity for RS-KT, and both computational and sample complexity for RS-BC, since $\theta$ determines the cardinality of the "augmented" state space on which we estimate the expert's policy). Ideally, we would like to set $\theta=0$ so that there is no approximation error, but this is unfeasible, as it would require an infinite amount of resources.
> >
> > In practice, the choice of $\theta$ depends on the amount of available resources (computational time and data) and on our tolerance to approximation error. Recall that the approximation error is bounded by $H\theta$ (see Lemma 4.2). Therefore, if we have a tolerance $\Delta$, we will try to set $\theta\le\Delta/H$. However, this might be impossible if we cannot afford to solve an LP with $n=(SAH+1)(H/\theta+1)$ optimization variables and $m=2+((H-1)S+3)(H/\theta+1)$ constraints (for RS-KT), or if we have too few demonstrations for estimating the expert's policy at all $SAH(H/\theta+1)$ augmented states (for RS-BC). In such cases, $\theta$ needs to be increased, accepting a larger approximation error $>\Delta$ while reducing the computational time and/or the estimation error.
> >
> > > Missing related works, to name a few, sample complexity/regret guarantees on risk-sensitive RL [1][2], CVaR RL [3][4], risk-sensitive distributional RL [5][6]
> >
> > We thank the Reviewer for the suggestion. We have updated the submission by including a discussion of these works in Appendix A.

---

> > > ### Author Response · Authors · 2025-11-17
> > >
> > > [7] Shehryar Malik, Usman Anwar, Alireza Aghasi, and Ali Ahmed. Inverse constrained reinforcement learning. In International Conference on Machine Learning 38 (ICML), 2021.
> > >
> > > [8] Filippo Lazzati and Alberto Maria Metelli. Learning utilities from demonstrations in markov decision processes. In International Conference on Machine Learning 42 (ICML), 2025.
> > >
> > > [9] Hans Follmer and Alexander Schied. Stochastic Finance: An Introduction in Discrete Time. De Gruyter, 2016.
> > >
> > > [10] Jonathan Lacotte, Mohammad Ghavamzadeh, Yinlam Chow, and Marco Pavone. Risk-sensitive
> > > generative adversarial imitation learning. In International Conference on Artificial Intelligence and Statistics 22 (AISTATS), 2019.

---

> ### Comment · Reviewer_sUz6 · 2025-11-24
>
> Thank you for your detailed repsonse and for incorporating the revisions, which address most of my concern. Thus, I maintain my positive assessment.

---

### Official Review · Reviewer_P3jS · 2025-11-12

**Soundness:** 4
**Presentation:** 3
**Contribution:** 3
**Rating:** 6
**Confidence:** 3

**Summary:**

The paper studies training a risk-sensitive reinforcement learning agent through imitation learning. The classical imitation learning methods only focus on matching the expert's expected return (average performance), which may not capture risk attitudes. To overcome this, the authors propose matching the entire expert return distribution, such as variance and tail risks. The approach formulates a return distribution matching (RDM) problem using the Wasserstein distance between distributions.

The authors propose two provably efficient algorithms, RS-BC (Risk-Sensitive Behavior Cloning) for unknown transition models and RS-KT for known transition models that learn a class of non-Markovian policies that efficiently express risk-sensitive behaviors. These algorithms improve over standard imitation learning by matching the full return distribution, accommodating risk attitudes. They provide theoretical guarantees on sample complexity and policy approximation, and then empirically validated that RS-BC and RS-KT outperform baseline methods.

**Strengths:**

**The following are the strengths of the paper:**
1. This paper considers training a risk-sensitive reinforcement learning agent through imitation learning while matching the entire expert return distribution to capture the risk attitudes.

2. The authors propose two provably efficient algorithms, RS-BC and RS-KT, that learn a class of non-Markovian policies that efficiently express risk-sensitive behaviors, while having theoretical guarantees on sample complexity and policy approximation.

3. Finally, the authors empirically validated that RS-BC and RS-KT outperform baseline methods.

**Weaknesses:**

**The following are the weaknesses of the paper:**
1. The proposed algorithms are restricted to finite tabular RL environments, limiting their scalability to RL problems with large or continuous state spaces.

2. It is unclear how expensive Step 2 of RS-KT is, especially for large LP. How practical it is to assume the expert's reward function (also the transition model for RS-KT) is known.

**Questions:**

Please address the weaknesses of the paper. I am open to changing my score based on the authors' responses.

**Details Of Ethics Concerns:**

Since this work is a theoretical paper, I do not find any ethical concerns.

---

> ### Author Response · Authors · 2025-11-17
>
> We are glad that the Reviewer appreciated the problem formulation of training a risk-sensitive RL agent through imitation learning, as well as our two novel, provably efficient algorithms. Below, we respond to the Reviewer's comments and questions.
>
> > The proposed algorithms are restricted to finite tabular RL environments, limiting their scalability to RL problems with large or continuous state spaces.
>
> We agree with the Reviewer that the proposed algorithms, RS-BC and RS-KT, cannot be applied as is to RL problems with large or continuous state spaces, since, by design, they are suited for tabular MDPs, as their sample and/or computational complexities scale linearly with the size of the state space. *However*, we would like to note that it is possible to devise a quite immediate extension of RS-BC that can be applied to larger or continuous environments.
>
> Specifically, observe that RS-BC can be extended to an algorithm that learns a policy through supervised learning, similar to standard behavioral cloning. Formally, we model the policy as taking as input the features $x$, consisting of the current state and the amount of reward collected so far in the trajectory, $x=(s_h^i,\sum_{h'=1}^{h-1}r_{h'}^E(s_{h'}^i,a_{h'}^i))$, and use as target $t$ the action taken by the expert, $t=a_h^i$. In the case where the number of actions $A$ is finite and relatively small, this can be done, for instance, by training a neural network where the last layer is a softmax, using the categorical cross-entropy loss; while in the case where $A$ is large or continuous, we can learn a generative model for actions conditioned on the current state and cumulative reward. In this case, an interesting option would be to use diffusion models, in an analogous way to how they are used for learning risk-neutral diffusion policies through behavioral cloning in recent works (e.g., see [1]).
>
> Regarding RS-KT, we note that extending it to continuous state spaces is not immediate, and it would necessarily require many approximations, as we cannot enforce a flow constraint for every possible state, since there are infinitely many. Moreover, also assuming access to the dynamics $p$ would be quite unrealistic in continuous state spaces, as $p$ may in principle be an arbitrary measure. For these reasons, we believe that, in such environments, a different approach from that adopted by RS-KT (potentially less sample efficient) should be considered for exploiting the available dynamics information. In particular, in the case where we have access to information on the dynamics of the environment in terms of the possibility of collecting trajectories by playing arbitrary policies (i.e., we can interact with the environment), we believe that a meanigful extension of GAIL [2] might do the trick. Indeed, GAIL aims to find a policy that matches the occupancy measure of the expert's policy by exploiting additional rollouts in the environment. If we augment the state space to keep track of the cumulative reward of trajectories, then an "augmented" version of GAIL would match the distribution over states and returns, which implies matching the return distribution. We would be happy to provide additional details if the Reviewer desires. However, given the difficulties and the substantial changes required, as detailed above, we do not consider it appropriate to include this extension in the present work.

---

> > ### Author Response · Authors · 2025-11-17
> >
> > > It is unclear how expensive Step 2 of RS-KT is, especially for large LP.
> >
> > As explained in Appendix C.3.2, the LP of RS-KT has $n=(SAH+1)|\mathcal{Y}^\theta|$ optimization variables and $m=2+((H-1)S+3)|\mathcal{Y}^\theta|$ constraints (note that $|\mathcal{Y}^\theta|\le 1+H/\theta$). In other words, the number of optimization variables $n$ and constraints $m$ of the LP is polynomial in the quantities of interest $S,A,H,1/\theta$ (and the constants are small). Given that in the literature there exist various efficient algorithms in $n$ and $m$ for solving linear programs (i.e., polynomial-time, such as interior-point methods and the ellipsoid method, or more practical, such as the simplex method; see [3]), our LP can then be efficiently solved for "reasonable" (i.e., not too large) values of $S,A,H$ and $1/\theta$.
> >
> > We recognize that this computational complexity can, of course, become quite expensive when the quantities of interest $S,A,H,1/\theta$ become large. However, we conjecture that this (polynomial) "LP complexity" is unavoidable if one wants to achieve the small sample complexity of RS-KT  (see Theorem 4.4). We detail our intuition below, and leave to future work the task of proving a lower bound formally.
> >
> > **Intuition**: Our goal is to find a policy whose return distribution is close to that of the expert. Intuitively, it is reasonable that this problem requires at least enough demonstrations to properly estimate the expert's return distribution. Since this task requires precisely the amount of data required by RS-KT (see Eq. 9), and since computing a policy whose return distribution is close to the estimated one amounts to solving the LP in RS-KT, we conjecture that no algorithm can achieve the same sample complexity as RS-KT with lower computational complexity.
> >
> > > How practical it is to assume the expert's reward function (also the transition model for RS-KT) is known.
> >
> > **Regarding the knowledge of the expert's reward function**: We would like to mention that:
> >
> > - There are other works in the literature that consider problem settings with a known reward function, such as inverse constrained RL [4] and utility learning [5]. Examples of applications mentioned in these works include autonomous driving [4].
> > - The simplest real-world example we can give is finance [6], where the reward function is represented by money, and the amount of money received after executing an action is usually known or observed. In this setting, observe that merely training an agent to maximize the *expected* return may be undesirable. Thus, we might prefer to use human expert guidance on which risk-sensitive behavior should be applied. In particular, matching the risk sensitivity of the expert can be done through our algorithms (i.e., RS-BC or RS-KT), which can meaningfully exploit the reward information.
> >
> > **Regarding the knowledge of the transition model in RS-KT**: We remark that we already provided an algorithm for risk-sensitive imitation learning when the transition model is unknown, i.e., RS-BC. The goal of RS-KT is precisely to demonstrate that, with the additional knowledge of the transition model, it is possible to achieve a much lower sample complexity compared to when the transition model is unknown (and also to show how to exploit this additional information). Moreover, as mentioned at lines 311-313, we would like to note that it might be possible to construct a variant of RS-KT that requires online access to the environment instead of exact knowledge of the transition model, based on the insights of [7]. Specifically, we would first run a reward-free exploration algorithm (e.g., [8]) to construct an estimate of the transition model, and then use this estimate in place of the true $p$ in the LP of RS-KT. We would be happy to provide additional details on this idea.

---

> > > ### Author Response · Authors · 2025-11-17
> > >
> > > [1] Chi, C., Feng, S., Du, Y., Xu, Z., Cousineau, E., Burchfiel, B., & Song, S. Diffusion policy: Visuomotor policy learning via action diffusion. The International Journal of Robotics Research, 2023.
> > >
> > > [2] Jonathan Ho and Stefano Ermon. Generative adversarial imitation learning. In Advances in Neural Information Processing Systems 29 (NeurIPS), 2016.
> > >
> > > [3] Bertsimas, D., & Tsitsiklis, J.N. Introduction to linear optimization. Athena scientific optimization and computation series. 1997.
> > >
> > > [4] Shehryar Malik, Usman Anwar, Alireza Aghasi, and Ali Ahmed. Inverse constrained reinforcement learning. In International Conference on Machine Learning 38 (ICML), 2021.
> > >
> > > [5] Filippo Lazzati and Alberto Maria Metelli. Learning utilities from demonstrations in markov decision processes. In International Conference on Machine Learning 42 (ICML), 2025.
> > >
> > > [6] Hans Follmer and Alexander Schied. Stochastic Finance: An Introduction in Discrete Time. De Gruyter, 2016.
> > >
> > > [7] Tian Xu, Ziniu Li, Yang Yu, and Zhi-Quan Luo. Provably efficient adversarial imitation learning with unknown transitions. In Conference on Uncertainty in Artificial Intelligence 39 (UAI), 2023.
> > >
> > > [8] Emilie Kaufmann, Pierre Menard, Omar Darwiche Domingues, Anders Jonsson, Edouard Leurent, and Michal Valko. Adaptive reward-free exploration. In International Conference on Algorithmic Learning Theory 32 (ALT 2021), pp. 865–891, 2021.

---

### Meta-Review · Area_Chair_bJ8d · 2025-12-24

**Summary:**

This paper considers the risk-sensitive imitation learning in RL that aims to train an agent to match not only an expert’s average performance but also its risk attitude. Under the tabular RL setting, the authors propose and analyze two algorithms called RS-BC and RS-KT, depending on the availability of the knowledge on transition dynamics. Preliminary numerical evaluation of the methods are also performed.

Overall, the authors have properly addressed many of the major issues. For part of major concerns that are not resolved, they also provide appropriate justification. There are a few questions not well addressed, but the reviewers are still giving overall positive ratings. Combining the rebuttal discussion and the initial rating of the paper, the AC suggest an accept to this paper.

**Reviewer Concerns:**

Addressed (or properly justified):
(1) P3jS: It is unclear how expensive Step 2 of RS-KT is, especially for large LP. (addressed)
(2) P3jS: The proposed algorithms are restricted to finite tabular RL environments, limiting their scalability to RL problems with large or continuous state spaces. (justified).
(3) sUz6: The computational cost of RS-KT. (justified).

Not Addressed:
(1) sUz6 & 9FFJ & XjG5: Unknown-reward setting lacks practical algorithm. Theoretical sample-efficiency for the unknown-reward case is established by using an oracle-based argument, but a concrete practical algorithm for this setting is deferred. This is an important practical case since expert reward unknown.
(2) 9FFJ: practical scalability and validity issue due to strong assumptions.

**Reviewer Scores:**

The reviewers are all giving positive scores. However, their concerns on known reward are only justified, but not addressed. This indicate that they may not want to increase the score.

---

### Decision · Program_Chairs · 2026-01-26

Accept (Poster)